

# The global form of flavor symmetries and 2-group symmetries in 5d SCFTs

**Fabio Apruzzi, Sakura Schäfer-Nameki, Lakshya Bhardwaj and Jihwan Oh**

Mathematical Institute, University of Oxford,
Andrew-Wiles Building, Woodstock Road, Oxford, OX2 6GG, UK

## Abstract

2-group symmetries arise when 1-form symmetries and 0-form symmetries of a theory mix with each other under group multiplication. We discover the existence of 2-group symmetries in 5d $\mathcal{N} = 1$ abelian gauge theories arising on the (non-extended) Coulomb branch of 5d superconformal field theories (SCFTs), leading us to argue that the UV 5d SCFT itself admits a 2-group symmetry. Furthermore, our analysis determines the global forms of the 0-form flavor symmetry groups of 5d SCFTs, irrespective of whether or not the 5d SCFT admits a 1-form symmetry. As a concrete application of our method, we analyze 2-group symmetries of all 5d SCFTs, which reduce in the IR, after performing mass deformations, to 5d $\mathcal{N} = 1$ non-abelian gauge theories with simple, simply connected gauge groups. For rank-1 Seiberg theories, we check that our predictions for the flavor symmetry groups match with the superconformal and ray indices available in the literature. We also comment on the mixed 't Hooft anomaly between 1-form and 0-form symmetries arising in 5d $\mathcal{N} = 1$ non-abelian gauge theories and its relation to the 2-groupS.


doi:10.21468/SciPostPhys.13.2.024

# 1   Introduction and Overview

## 1.1   Motivation

Generalized $p$-form global symmetries [1] have emerged as a fruitful tool for understanding new and fundamental properties of quantum field theories (QFTs). The charged objects under such $p$-form symmetries are $p$-dimensional, and symmetry generators are topological operators of co-dimension $p + 1$. The case $p = 0$ is the standard global symmetry case.

In this paper we will study higher-form symmetries, in particular 0- and 1-form symmetries in 5d theories. In 5d, 0-form flavor symmetries can enhance at the strongly-coupled superconformal field theory (SCFT) point [2], and have been studied from various perspectives. The most comprehensive analysis is based on geometric considerations i.e. the realization of 5d SCFTs as M-theory on singular Calabi-Yau threefolds, as obtained in [3–10][1]. In these geometric studies, however, only the flavor symmetry algebra has been determined, not the global

---

[1]Alternative points of view using field theory arguments as well as brane-webs have appeared in e.g. [11–16].

form of the flavor symmetry group [2]. A possible way to constrain the global form was proposed in [18], in terms of field theoretic constraints coming from t' Hooft anomaly matching between the 0-form and the 1-form symmetry anomaly present in 5d gauge theories. One of the results in the paper is the derivation, from the geometry, of the *global* form, i.e. the 0-form flavor symmetry *group*.

Central to this endeavour is the concept of the structure group, which is the group that acts faithfully on all the states in the theory. We derive the structure group from the geometry, which encodes all the generators of the symmetries (gauge and global symmetries), and the charged states, arising from wrapped M2-branes. Quotienting the gauge and global symmetry groups by the maximal group that acts trivially on the matter, results in the structure group. The 0-form part of this is the global 0-form flavor symmetry group. We determine this in many examples, including the rank 1 $E_{N_f+1}$ Seiberg theories, for which we show that the global symmetry group is the maximally center reduced group except for $N_f = 1$. For example, for $E_1 = SU(2)_0$ the flavor algebra is $\mathfrak{su}(2)$, and the global flavor symmetry group is $SO(3)$. This is consistent with the superconformal index [12] and with the proposal in [18]. More generally, for all the rank 1 $E_{N_F+1}$ theories, the flavor group is in agreement with the results of the superconformal index, and also the ray index of [19]. Note that we determine the continuous, non-abelian part of the flavor symmetries, i.e. there could be additional discrete factors and/or abelian factors.

In addition to 0-form symmetries, 5d SCFTs and gauge theories can also enjoy higher-form symmetries. From the M-theory realization of 5d theories on singular Calabi-Yau three-folds, one can identify both theories with 1-form (or magnetic dual 2-form) symmetries [20–22] as well as 3-form symmetries [23]. Our focus here will be on 1-form symmetries $\mathcal{O}$, which characterize the spectrum of line operators of the 5d theory. If the theory has an IR gauge theory description, the 1-form symmetry of the geometric construction and thereby the SCFT is in agreement with that of the gauge theory (plus instanton particles). Again, these can be computed purely from the geometry, and do not rely necessarily on an IR gauge theory description. Global symmetries can have (mixed) 't Hooft anomalies. In 5d such anomalies were proposed in [18, 24].

To fully determine the global symmetries of a theory, however, it is not enough to compute the *p*-form symmetry groups: generalized global symmetries can form higher-group structures. In the continuous case, this has a description in terms of gauge transformations of the background gauge fields, which mix with each other. In the case of discrete symmetries, such as the 1-form symmetries in 5d, the higher-group structure is more subtle. It is the interconnection between all these aspects, and the exploration of 2-groups in 5d theories that are another focus of this paper.

Higher-group symmetries arise when *p*-form and *q*-form generalized symmetry groups do not form a direct product, but rather mix with each other [25]. In this paper, we study 2-group symmetries, formed by 0- and 1-form symmetries, in a class of strongly coupled QFTs, the aforementioned 5d SCFTs. Various works have studied higher-groups in QFTs in various dimensions, see e.g. [25–40] where they are discussed from various points of view. Note that since 5d and 6d SCFTs do not admit a conserved 2-form current multiplet [41], these theories therefore do not admit continuous 1-form symmetries [38]. However, this does not preclude the existence of *discrete* 1-form symmetries, and 2-group structures in which these discrete 1-form symmetries participate.

The 5d $SU(2)_0$ theory, despite its deceptive simplicity, incorporates most of these effects: it has a non-simply connected global flavor symmetry $\mathcal{F} = SO(3)$, a 1-form symmetry $\mathcal{O} = \mathbb{Z}_2$, and we show it has a 2-group symmetry, which becomes clearly visible when we deform the

---

[2]See a recent work [17] which determines the global form of flavor symmetry groups of 4d $\mathcal{N} = 2$ Class S theories.

conformal vacuum to a non-conformal vacuum lying on the Coulomb branch of vacua[3] of this SCFT. The low-energy theory after choosing such a non-conformal vacuum is a $U(1)$ gauge theory with $\mathfrak{su}(2)$ flavor symmetry algebra and $\mathbb{Z}_2$ 1-form symmetry, such that the associated flavor symmetry group is $SO(3)$ which forms a non-trivial 2-group with the $\mathbb{Z}_2$ 1-form symmetry. We observe that this 2-group symmetry continues to hold as we include states of arbitrarily large energies, leading us to propose that the 2-group symmetry must be a property of the UV SCFT itself, rather than an emergent property of the low-energy abelian gauge theory description.

Given this rich structure, we devote an entire section of the paper to this theory, which provides a detailed exposition of the structures in this paper in the simplest possible setting – and the readers wanting to understand the main conceptual points, may well focus their attention on section 3. In this context we also discuss the theory obtained after gauging the 1-form symmetry: the $SO(3)_0$ theory. This has a 2-form symmetry $\mathbb{Z}_2$ and we provide evidence for its global flavor symmetry to be $SO(3)$ as well. If the 2-group symmetry of the $SU(2)_0$ theory is non-anomalous, the $SO(3)_0$ theory has a mixed $0-2$-form symmetry anomaly – a general effect when gauging 1-form symmetries in non-anomalous 2-groups.

## 1.2 Overview of Results

In this more technical part of the introduction, we provide a road map of the results in the paper. In particular, we give a summary of the relevant concepts, such as the structure group, the 2-group symmetry and provide an overview of the results.

**2-Groups.** The class of 2-group symmetries studied in this paper can be understood as an "extension" of the 0-form symmetry group $\mathcal{F}$ (which is taken to be a compact Lie group) of a QFT $\mathfrak{T}$ by the 1-form symmetry group $\mathcal{O}$ (which is taken to be a finite group) of $\mathfrak{T}$. The "extension class" $[\mathcal{P}_3]$, often referred to as the Postnikov class, is an element of $H^3(B\mathcal{F}, \mathcal{O})$, which controls the background fields for $\mathcal{F}$ and $\mathcal{O}$ as follows

$$\delta B_2 = B_1^* \mathcal{P}_3, \tag{1}$$

where $\mathcal{P}_3 \in C^3(B\mathcal{F}, \mathcal{O})$ is an $\mathcal{O}$ valued 3-cochain on the classifying space $B\mathcal{F}$ of $\mathcal{F}$ which is a representative of $[\mathcal{P}_3]$, $B_2 \in C^2(M, \mathcal{O})$ is an $\mathcal{O}$ valued 2-cochain describing the background field for 1-form symmetry on the spacetime manifold $M$, and $B_1^*$ is the pull-back associated to the map $B_1 : M \to B\mathcal{F}$ describing the background principal $\mathcal{F}$-bundle on $M$. The relation (1) has the following two important consequences:

- The 1-form symmetry background field $B_2$ is not closed in the presence of a non-trivial 0-form symmetry background $B_1$.

- The 0-form symmetry backgrounds $B_1$ are constrained such that the pullback $B_1^*[\mathcal{P}_3]$ of the Postnikov class $[\mathcal{P}_3]$ to spacetime $M$ is trivial[4].

In section 2.1 we discuss how such 2-groups arise naturally in gauge theories, with the 2-group structure being formed between electric[5] 1-form symmetry and flavor 0-form symmetry associated to matter content. It turns out in this analysis that, under some technical

---

[3]This is the non-extended Coulomb branch since we are not turning on mass parameters (i.e. the $SU(2)_0$ inverse gauge coupling).

[4]In other words, the Postnikov Class $[\mathcal{P}_3]$ can be thought of as the obstruction class for lifting a 0-form bundle to a 2-group bundle.

[5]We do not consider magnetic 1-form symmetries in our analysis as our natural application is to 5d gauge theories while magnetic 1-form symmetries arise only in 4d gauge theories. The magnetic 1-form symmetries can also form 2-groups as discussed in [37].

assumptions, the Postnikov class can be described as

$$[\mathcal{P}_3] = \text{Bock}\big([\nu_2]\big), \tag{2}$$

where Bock is the Bockstein homomorphism in the long exact sequence in cohomology associated to a short exact sequence

$$0 \to \mathcal{O} \to \mathcal{E} \to \mathcal{Z} \to 0, \tag{3}$$

where $\mathcal{O}$ is the electric 1-form symmetry group, $\mathcal{Z}$ appears in the 0-form flavor symmetry group as

$$\mathcal{F} = F/\mathcal{Z}, \tag{4}$$

where $F$ is the simply connected group associated to the flavor Lie algebra $\mathfrak{f}$, and thus $\mathcal{Z}$ is the subgroup of the center $Z_F$ of $F$ that is modded out to give rise to the 0-form symmetry group $\mathcal{F}$. $\mathcal{E}$ is an extension of $\mathcal{Z}$ by $\mathcal{O}$ determined by the matter content, and $[\nu_2] \in H^2(B\mathcal{F}, \mathcal{Z})$ is the characteristic class capturing the obstruction of lifting $\mathcal{F}$ bundles to $F$ bundles.

It should be noted that the 2-group structure is non-trivial only if the Postnikov class (2) is a non-trivial element of $H^3(B\mathcal{F}, \mathcal{O})$. However, the Bockstein homomorphism applied to $[\nu_2]$ may lead to trivial elements of $H^3(B\mathcal{F}, \mathcal{O})$. A trivial example of this situation occurs if the short exact sequence (3) splits, in which case the Bockstein homomorphism is the trivial homomorphism and so we obtain a trivial Postnikov class by applying Bock to $[\nu_2]$. In all the cases studied in this paper, we confirm that $H^3(B\mathcal{F}, \mathcal{O})$ is non-trivial, so that it is possible for (2) to be non-trivial if (3) does not split. However, we have not been able to confirm that (2) is indeed non-trivial for a few cases appearing in this paper. We highlight these cases in section 1.3.

**Flavor groups and 2-groups in 5d SCFTs and SQFTs.** We apply the general analysis of section 2.1 to 5d SCFTs. In particular in section 2.2.1, we apply this to the 5d $\mathcal{N} = 1$ abelian gauge theory arising in the IR when a 5d SCFT is studied with a choice of non-conformal vacuum lying on its Coulomb branch (CB) of vacua[6]. We use the M-theory construction of the 5d SCFTs to determine information about the matter content in this abelian gauge theory. This analysis determines a 0-form flavor symmetry group $\mathcal{F}$ and possibly a 2-group symmetry for the abelian theory. We propose that $\mathcal{F}$ should be identified as the 0-form flavor symmetry group of the UV 5d SCFT itself[7], and similarly the 2-group symmetry in the abelian theory should also be identified as 2-group symmetry of the UV 5d SCFT. This is because the 0-form flavor group $\mathcal{F}$ and the 2-group symmetry remain invariant as we include states of arbitrarily high energies as matter content into the above abelian gauge theory, reflecting that $\mathcal{F}$ and 2-group are properties of the UV theory itself, which is the 5d SCFT.

In section 2.2.2, we study 5d supersymmetric QFTs (SQFTs) obtained after mass deforming 5d SCFTs. The 0-form flavor symmetry group and possible 2-group symmetry of such a 5d SQFT can be obtained in a similar fashion by applying the analysis of section 2.1 to the 5d $\mathcal{N} = 1$ abelian gauge theory arising in the IR when we choose some generic non-conformal vacuum lying on the CB of vacua[8] of the 5d SQFT. Even though we use a generic vacuum in

---

[6]Here it should be noted that no mass parameters have been turned on, so the 5d SCFT has *not* been deformed. In other words, we are studying the theory on *non-extended* CB, which is a specific sub-locus of the extended CB, where the masses and tensions of various dynamical particles and strings have been tuned such that the resulting theory admits the full flavor symmetry of the 5d SCFT, and thus the theory can be coupled to backgrounds for this symmetry.

[7]It should be noted that we restrict our attention to global form of flavor groups associated to the non-abelian part of the flavor symmetry algebra only.

[8]Again, this should be understood as a particular sub-locus of the full extended CB. In fact the full extended CB is by definition the total space formed when the CB of vacua of the 5d SCFT and all the 5d SQFTs (obtained by mass deforming the 5d SCFT) are fibered over the space of mass deformations.

CB of the SQFT to derive results about 0-form groups and 2-groups, they continue to hold at non-generic vacua in CB of the SQFT, since these results are properties of the UV SQFT itself.

The prime example, where we illustrate this, is the $SU(2)_0$ SCFT, which we show to have a 2-group symmetry. For this SCFT, we determine the flavor symmetry group to be $SO(3)$. The flavor symmetry is broken to $U(1)$ after turning on the mass parameter (which can be identified as the inverse gauge coupling for the $SU(2)$ gauge group), and the resulting 5d SQFT *does not* admit a 2-group symmetry. This 5d SQFT admits a special point in its CB of vacua where the low-energy theory is the 5d $\mathcal{N} = 1$ non-abelian pure $SU(2)_0$ gauge theory.

**0-form/2-form mixed anomaly.**    As discussed in [25], if a theory has a 2-group without any 't Hooft anomaly, then one can gauge the 1-form symmetry to obtain another theory which has a dual $(d-3)$-form symmetry (where $d$ is the spacetime dimension) which does not mix with 0-form symmetry to form a higher-group structure, but there is a mixed 't Hooft anomaly between the $(d-3)$-form and 0-form symmetries. This can be concretely illustrated in the gauge theory context of section 2.1, which we discuss in section 2.3. If we consider a 5d SCFT/SQFT having a 2-group symmetry, we would expect to see a mixed 't Hooft anomaly between 2-form and 0-form symmetries in the 5d SCFT/SQFT obtained after gauging the 1-form symmetry of the starting 5d SCFT/SQFT, provided that the 2-group symmetry in the starting 5d SCFT/SQFT is non-anomalous.

**Perturbative 2-group symmetries.**    An interesting characterization of our results arises when a 5d SCFT can be mass deformed such that the resulting theory can be described in the IR by a 5d $\mathcal{N} = 1$ non-abelian gauge theory. As is well-known, in such a case, the non-abelian flavor symmetries of the 5d SCFT can be characterized as either perturbative or instantonic, with perturbative symmetries being the ones visible at the level of the 5d $\mathcal{N} = 1$ non-abelian gauge theory, while non-abelian instantonic symmetries are not visible at the level of gauge theory (only its abelian part is visible).

For perturbative symmetries, one would expect the global form of 0-form flavor symmetry group and 2-groups to be visible at the level of non-abelian gauge theory. Indeed, this turns out to be the case sometimes. An example is provided by $SU(4)$ gauge theory with Chern-Simons (CS) level 2 and a hyper in 2-index antisymmetric irreducible representation (irrep) of the $SU(4)$ gauge group. There is a perturbative $\mathfrak{su}(2)$ flavor symmetry algebra rotating the antisymmetric hyper. Using the general gauge theoretic analysis of section 2.1, one finds that at the level of gauge theory, the 0-form flavor symmetry group corresponding to the $\mathfrak{su}(2)$ flavor algebra should be $SO(3)$, which should further mix with the $\mathbb{Z}_2$ 1-form symmetry to give rise to a non-trivial 2-group structure. In this case, we find the same results by analyzing the CB of the corresponding 5d SCFT.

However, in some other cases, we find that the non-abelian gauge theory expectations are modified at the conformal point. An example is provided again by $SU(4)$ gauge theory with a hyper in 2-index antisymmetric irrep but now with CS level 0. The gauge theory expectation is insensitive to CS level, so we would expect same results as for the case with CS level 2. Instead, the CB of 5d SCFT tells us that the correct 0-form flavor symmetry group is $SU(2)$ instead of $SO(3)$ and furthermore the $SU(2)$ 0-form symmetry does *not* mix with the $\mathbb{Z}_2$ 1-form symmetry to form a 2-group. This mismatch between the non-abelian gauge theory expectation and the correct answer at the conformal point can be fixed by accounting for instanton particles as providing extra matter content in the non-abelian gauge theory while performing the computation of section 2.1. In section 2.2.3, we gather the required information about the instanton particles which when accounted in the computation of 2.1 leads to the correct prediction for global form of 0-form flavor symmetry groups and 2-group symmetries of the corresponding 5d SCFT.

**Instantonic 2-group symmetries.** On the other hand, we find that the instantonic symmetries can also form 2-groups, but such 2-groups cannot be understood straightforwardly in terms of non-abelian gauge theory plus instantons. An example of instantonic 2-group symmetry is provided by the 5d SCFT that describes UV completion of 5d $\mathcal{N} = 1$ pure $SU(2)$ gauge theory with vanishing discrete theta angle, which has an instantonic $\mathfrak{su}(2)$ flavor symmetry algebra. We find that the associated 0-form flavor symmetry group is $SO(3)$ which forms a non-trivial 2-group with the $\mathbb{Z}_2$ 1-form symmetry.

This analysis is generalized in section 4, where we study 2-group symmetries (and, in the process, global forms of 0-form flavor symmetry groups) of 5d SCFTs that admit a mass deformation, such that after the deformation, the low energy theory can be described by some 5d $\mathcal{N} = 1$ non-abelian gauge theory with a simple, simply connected gauge group [42, 43]. This is done by applying the analysis of section 2.2.1 to these 5d SCFTs. The M-theory constructions of these theories (including non-abelian flavor symmetries) are taken from [10], whose analysis was based on [9]. We collect the theories showing potentially non-trivial 2-group structures in section 1.3.

$SU(2)_0$ **Seiberg Theory.** We devote section 3 to a detailed study of the above-discussed 5d SCFT that describes UV completion of 5d $\mathcal{N} = 1$ pure $SU(2)$ gauge theory with vanishing discrete theta angle. In sections 3.1 and 3.2, we describe in detail how the M-theory construction of the theory leads one to conclude that the 0-form symmetry group of the theory is $SO(3)$ and that it mixes with $\mathbb{Z}_2$ 1-form symmetry to form a non-trivial 2-group symmetry. Key to this analysis is the structure group of the abelian gauge theory arising on the Coulomb branch which relies on non-perturbative input from geometry. In section 3.3, we provide an alternative way to derive the structure group for this theory that only utilizes the low-energy prepotential and does not require the details of the geometry and the corresponding non-perturbative input. In Section 3.4, we argue that the 2-group can be thought of as being induced due to a global anomaly cancellation for the worldsheet theory of strings ending on branes involved in a Type IIB construction of the 5$d$ theory.

In section 3.5, we provide additional evidence for the results of sections 3.1 and 3.2. We compactify the 5d SCFT on a circle of non-zero radius, and study the low-energy theory at a special point on the resulting 4d CB of vacua. This low-energy theory is a $U(1)$ gauge theory with 2 massless electrons of charge 2 [44]. The $\mathfrak{su}(2)$ flavor symmetry rotating the electrons is the 4d avatar of the instantonic flavor symmetry of the 5d SCFT, and the abelian theory has a $\mathbb{Z}_2$ 1-form symmetry because the charge of electrons is an even integer, which can be identified as the avatar of the $\mathbb{Z}_2$ 1-form symmetry of the 5d SCFT. Applying the general analysis of section 2.1, we find that the low-energy 4d abelian gauge theory should have an $SO(3)$ 0-form flavor symmetry group which forms a non-trivial 2-group with the $\mathbb{Z}_2$ 1-form symmetry. We propose that this 2-group is simply the 4d avatar of the proposed 2-group of the 5d SCFT.

**Mixed Anomaly and Higgs branch Matching.** In section 3.6, we discuss the relationship between the 2-group structure for this 5d SCFT proposed in this paper and the mixed 0-form/1-form anomaly in the low-energy $SU(2)_0$ gauge theory discussed in [18]. This anomaly is between the $\mathbb{Z}_2$ 1-form symmetry and $U(1)_I$ 0-form symmetry associated to instanton current. In [45] this anomaly with the $U(1)_I$ symmetry was derived from the M-theory reduction on the link (meaning boundary of the Calabi-Yau realization of the 5d SCFT) reduction. This mixed anomaly should lift to an anomaly of the 2-group symmetry at the conformal point, depending on the non-abelian flavor symmetry $SO(3)$. Such a lift, which is compatible with the 2-group should be possible and we will return to this question in the future. Another interesting question is how the mixed anomaly is matched on the Higgs branch (HB). We propose a mechanism to do so, by coupling the sigma-model on the Higgs branch to a 5d $\mathbb{Z}_2$

gauge theory, which realizes the 1-form symmetry.

Another possibility is that there are additional topological sectors that make the 2-group non-anomalous, in which case, as discussed in section 3.7, one can apply the analysis of section 2.3 to deduce that the global form of the 0-form symmetry group of the 5d SCFT UV completing the 5d pure $SO(3)_0$ gauge theory is $SO(3)$ which has a mixed anomaly with the magnetic $\mathbb{Z}_2$ 2-form symmetry. In section 3.8, we comment on the superconformal indices of $SU(2)_0$ and $SO(3)_0$ SCFTs and the relationship of the index to the global form of 0-form symmetry groups of these 5d SCFTs.

**Flavor Groups of $E_{N_f+1}$ Seiberg Theories.** In section 3.9, we use our method to deduce the global form of 0-form symmetry group of Seiberg theories, which do not admit 1-form symmetries. This is to illustrate that our method can be applied to deduce 0-form symmetry groups even when a 5d SCFT has no 1-form symmetry and hence cannot form any 2-group structure. We find that the flavor group is centerless for $N_f \neq 1$, which is in agreement with superconformal indices of these theories. We also explain how this result is consistent with the ray indices of these theories which exhibit representations charged non-trivially under the flavor center, whose presence would naively seem to suggest that the flavor group is the center-full simply connected one. In the process of the resolution of this apparent contradiction, we find that the geometric information needed for our computation is equivalently encoded in the ray indices.

**Appendices.** Appendix A.1 reviews the background material needed to understand the geometric computations performed in the paper. Appendix A.2 reviews how the charges under center symmetries are encoded in geometry, which form the backbone of the geometric computations performed in the paper. Appendix A.3 computes the flavor center charges of instantons which can be used to apply the analysis of section 2.2.3. Appendix B contains the details to determine the mixed anomaly between 0-form and 1-form symmetries for general 5d gauge theories, studied in [18] for $SU(N)$. We also review a cubic 't Hooft anomaly for 1-form symmetries in 5d $SU(N)$ gauge theories [24].

## 1.3 Single-Node Gauge-Theoretic 5d SCFTs with 2-Group Symmetry

In this subsection, we list all 5d SCFTs lying in a particular class which exhibit 2-group symmetries. This class of SCFTs is defined by demanding the existence of a mass deformation which reduces the theory in the IR to some 5d $\mathcal{N} = 1$ non-abelian gauge theory with a simple, simply connected gauge group and was studied extensively in [10, 42, 43]. We present our results in a tabular form in table 1 and we refer the reader to section 4 for more details. Different rows of the table denote different 5d SCFTs $\mathfrak{T}$. The corresponding 5d gauge theories are displayed in the first column as

$$G + \oplus_i n_i \boldsymbol{R}_i, \tag{5}$$

where $G$ denotes the gauge group, $\boldsymbol{R}_i$ denotes an irreducible representation of $G$ and $n_i$ denotes the number of full hypermultiplets present in the theory that transform in the representation $\boldsymbol{R}_i$. $\boldsymbol{R}_i = \boldsymbol{F}$ denotes fundamental representation for $G = SU(n)$ and vector representation for $G = \text{Spin}(n)$. $\boldsymbol{R}_i = \boldsymbol{\Lambda}^2$ denotes 2-index antisymmetric irrep for $G = SU(n), Sp(n)$. $\boldsymbol{R}_i = \boldsymbol{S}$ denotes a spinor irrep for $G = \text{Spin}(n)$. A subscript $k$ for $G = SU(n)$; $n \geq 3$ denotes that the CS level is $k$. A subscript 0 for $G = Sp(n), SU(2)$ denotes that the discrete theta angle is 0. An equality between two gauge theories denotes the fact that the 5d SCFT UV completing the two gauge theories is same. The second column lists the 1-form symmetry group $\mathcal{O}_{\mathfrak{T}}$ of the 5d SCFT $\mathfrak{T}$ which participates in the 2-group structure. The third column displays the 0-form flavor group $\mathcal{F}_{\mathfrak{T}}$ participating in the 2-group structure. The fourth column lists another

Table 1: Various 5d SCFTs exhibiting 2-group symmetries. $\mathfrak{T}$ labels the theory, $\mathcal{O}$ the 1-form symmetry, $\mathcal{F}$ the global flavor symmetry group. The last column indicates whether the Postnikov class is known or not. See text for details on how to read the table. $\mathbb{Z}_2^{\mathrm{diag}}$ is the diagonal $\mathbb{Z}_2$ subgroup of the center $\mathbb{Z}_2^{SU(2)} \times \mathbb{Z}_2^{Sp(4n-2)}$ of $SU(2) \times Sp(4n-2)$, where $\mathbb{Z}_2^{SU(2)}$ is the center of $SU(2)$ and $\mathbb{Z}_2^{Sp(4n-2)}$ is the center of $Sp(4n-2)$.

| $\mathfrak{T}$ | $\mathcal{O}_{\mathfrak{T}}$ | $\mathcal{F}_{\mathfrak{T}}$ | $F'$ | Confirmed? |
|---|---|---|---|---|
| $SU(2)_0$ | $\mathbb{Z}_2$ | $SO(3)$ | $SU(2)$ | ✓ |
| $SU(2n)_{2n}$ | $\mathbb{Z}_{2n}$ | $SO(3)$ | $SU(2)$ | |
| $Sp(2m-1)_0 + \mathbf{\Lambda}^2$ $= SU(2m)_{m+4} + \mathbf{\Lambda}^2$ | $\mathbb{Z}_2$ | $SO(3) \times SU(2)$ | $SU(2) \times SU(2)$ | ✓ |
| $Sp(2m)_0 + \mathbf{\Lambda}^2$ | $\mathbb{Z}_2$ | $SO(3) \times SU(2)$ | $SU(2) \times SU(2)$ | ✓ |
| $SU(2n)_4 + 2\mathbf{\Lambda}^2$ | $\mathbb{Z}_2$ | $SO(3) \times SO(3)$ | $SU(2) \times SO(3)$ | ✓ |
| $SU(2n)_0 + 2\mathbf{\Lambda}^2$ | $\mathbb{Z}_2$ | $SO(3) \times SU(2)$ | $SU(2) \times SU(2)$ | ✓ |
| $\mathrm{Spin}(4n) + (4n-3)\mathbf{F}$ | $\mathbb{Z}_2$ | $PSp(4n-2)$ | $Sp(4n-2)$ | ✓ |
| $\mathrm{Spin}(2n+1) + (2n-3)\mathbf{F}$ | $\mathbb{Z}_2$ | $SO(3) \times Sp(2n-3)$ | $SU(2) \times Sp(2n-3)$ | ✓ |
| $\mathrm{Spin}(4n+2) + (4n-2)\mathbf{F}$ | $\mathbb{Z}_2$ | $SO(3) \times PSp(4n-2)$ | $\frac{SU(2) \times Sp(4n-2)}{\mathbb{Z}_2^{\mathrm{diag}}}$ | ✓ |
| $\mathrm{Spin}(4n) + (4n-4)\mathbf{F}$ | $\mathbb{Z}_2$ | $SO(3) \times PSp(4n-4)$ | $SU(2) \times PSp(4n-4)$ | ✓ |
| $\mathrm{Spin}(4n+2) + 4m\mathbf{F}$; $0 \le m \le n-1$ | $\mathbb{Z}_2$ | $PSp(4m)$ | $Sp(4m)$ | |
| $\mathrm{Spin}(4n+2) + (4m+2)\mathbf{F}$; $0 \le m \le n-2$ | $\mathbb{Z}_2$ | $PSp(4m+2)$ | $Sp(4m+2)$ | ✓ |
| $SU(4)_2 + \mathbf{\Lambda}^2$ | $\mathbb{Z}_2$ | $SO(3)$ | $SU(2)$ | ✓ |
| $SU(4)_2 + 3\mathbf{\Lambda}^2$ | $\mathbb{Z}_2$ | $PSp(3) \times SU(2)$ | $Sp(3) \times SU(2)$ | |
| $\mathrm{Spin}(7) + 3\mathbf{F}$ | $\mathbb{Z}_2$ | $SO(3) \times Sp(3)$ | $SU(2) \times Sp(3)$ | ✓ |
| $\mathrm{Spin}(12) + 2\mathbf{S}$ | $\mathbb{Z}_2$ | $SO(3)^3$ | $SU(2)^2 \times SO(3)^2$ | ✓ |

group $F'$ such that $\mathcal{F}_{\mathfrak{T}} = F'/\mathbb{Z}_2$. The Postnikov class $[\mathcal{P}_3]$ of the 2-group can be described in each case as in (2) where $[\nu_2]$ describes a characteristic class in $H^2(B\mathcal{F}_{\mathfrak{T}}, \mathbb{Z}_2)$ which describes the obstruction of lifting $\mathcal{F}_{\mathfrak{T}}$ bundles to $F'$ bundles. The short exact sequence defining the Bockstein homomorphism in (2) takes the following form in every case

$$0 \to \mathbb{Z}_{2p} \to \mathbb{Z}_{4p} \to \mathbb{Z}_2 \to 0\,, \tag{6}$$

where $p$ is determined by identifying the first group $\mathbb{Z}_{2p}$ with the 1-form symmetry group $\mathcal{O}_{\mathfrak{T}}$ of $\mathfrak{T}$. Finally, the fifth column lists whether the presence of a non-trivial 2-group has been confirmed, i.e. whether it is known to the authors that the Postnikov class of the 2-group is non-trivial.

## 2 2-Groups and Global Form of Flavor in Gauge Theories

In this section we will develop the general theory, to describe the 2-group symmetries in 5d gauge theories, as well as the global form of the flavor symmetry group. Key to this analysis is the concept of the structure group, which we introduce first. In the next section, we will exemplify this general theory with the case of the rank 1 theory $SU(2)_0$, which exemplifies all of these aspects.

## 2.1 The Structure Group and 2-Group Symmetry

Consider a gauge theory with

$G$ : Gauge group, with center $Z_G$ and Lie algebra $\mathfrak{g}$

$F$ : Simply connected group, with center $Z_F$, associated to the flavor symmetry algebra $\mathfrak{f}$

Define $\mathcal{E}$ such that

$$Z_G \times Z_F \supset \mathcal{E} = \text{maximal subgroup that acts trivially on matter fields.} \tag{7}$$

The structure group of the theory is then

$$S = \frac{G \times F}{\mathcal{E}}. \tag{8}$$

The 1-form symmetry of the theory is the subgroup of $\mathcal{E}$ formed by elements of the form $(*,0)$

$$\mathcal{O} = \{(*,0) \in \mathcal{E} \subset Z_G \times Z_F\}. \tag{9}$$

We define the projections

$$\begin{aligned}
\pi_G : \quad & Z_G \times Z_F \to Z_G \\
& (\alpha,\beta) \mapsto \alpha \\
\pi_F : \quad & Z_G \times Z_F \to Z_F \\
& (\alpha,\beta) \mapsto \beta.
\end{aligned} \tag{10}$$

We then have

$$\begin{array}{ccccc}
\mathcal{E}' & \xleftarrow{\;\pi_G\;} & \mathcal{E} & \xrightarrow{\;\pi_F\;} & \mathcal{Z} \\
\uparrow & & \uparrow & & \\
\mathcal{O}' & \xleftarrow{\;\pi_G\;} & \mathcal{O} & \xrightarrow{\;\pi_F\;} & 0
\end{array} \tag{11}$$

Notice that $\mathcal{O}' \simeq \mathcal{O}$. The 0-form flavor symmetry *group* of the theory is then

$$\mathcal{F} = \frac{F}{\pi_F(\mathcal{E})} = \frac{F}{\mathcal{Z}}. \tag{12}$$

Notice that the above groups form a short exact sequence[9]

$$0 \to \mathcal{O} \to \mathcal{E} \to \mathcal{Z} \to 0. \tag{13}$$

Let us assume that the projection $\pi_G$ of $\mathcal{E}$ to $Z_G$ is injective, which is equivalent to the condition $\mathcal{E}' \simeq \mathcal{E}$. Let us also define

$$\mathcal{Z}' := \frac{\mathcal{E}'}{\mathcal{O}'}. \tag{14}$$

Notice that $\mathcal{Z}' \simeq \mathcal{Z}$ if the above assumption holds. Then, the primed groups satisfy a short exact sequence

$$0 \to \mathcal{O}' \to \mathcal{E}' \to \mathcal{Z}' \to 0, \tag{15}$$

which is isomorphic to (13).

---

[9]In this paper we represent all abelian groups as additive groups. Thus, the trivial group is denoted by 0 instead of 1.

When the 0-form symmetry background is trivial, the 1-form symmetry background can be described by an element $B \in H^2(M, \mathcal{O})$, where $M$ is the spacetime manifold. The gauge theory then sum over $G/\mathcal{O}'$ gauge bundles with

$$w_2 = B, \tag{16}$$

where $w_2$ captures the obstruction to lifting a $G/\mathcal{O}'$ bundle to a $G$ bundle.

Let us turn on a 0-form symmetry background described by a $\mathcal{F} = F/\mathcal{Z}$ bundle with $v_2 \in H^2(M, \mathcal{Z})$ capturing the obstruction of lifting the $F/\mathcal{Z}$ bundle to an $F$ bundle. Now the gauge theory sums over a class of $G/\mathcal{E}'$ bundles. Let $v_2' \in H^2(M, \mathcal{Z}')$ capture the obstruction of lifting a $G/\mathcal{E}'$ bundle to a $G/\mathcal{O}'$ bundle. The 1-form symmetry background $B$ fixes the further obstruction $w_2$ of lifting $G/\mathcal{O}'$ bundle to a $G$ bundle. Since we do not have a genuine $G/\mathcal{O}'$ bundle, $w_2$ is not necessarily closed anymore. We have

$$\delta w_2 = \text{Bock}(v_2'), \tag{17}$$

where Bock is the Bockstein map in the long exact sequence in cohomology

$$
\begin{aligned}
\cdots \to H^2(M, \mathcal{O}') &\longrightarrow H^2(M, \mathcal{E}') \longrightarrow H^2(M, \mathcal{Z}') \\
&\overset{\text{Bock}}{\longrightarrow} H^3(M, \mathcal{O}') \longrightarrow H^3(M, \mathcal{E}') \longrightarrow H^3(M, \mathcal{Z}') \to \cdots
\end{aligned}
\tag{18}
$$

associated to the short exact sequence (15). Moreover, the structure group (8) implies that

$$v_2' = v_2. \tag{19}$$

The gauge theory sums over all $G/\mathcal{E}'$ bundles with $v_2'$ specified by $v_2$ as in (19), $w_2$ specified by $B$ as in (16), and $v_2', w_2$ related by the Bockstein as in (17).

Substituting (16) and (19) into (17), we obtain the relationship

$$\delta B = \text{Bock}(v_2) \tag{20}$$

between the 1-form and 0-form backgrounds, which expresses the fact that the 1-form symmetry $\mathcal{O}$ and 0-form symmetry $\mathcal{F}$ sit in a 2-group whose Postnikov class is

$$\text{Bock}(\widehat{v}_2) \in H^3(B\mathcal{F}, \mathcal{O}), \tag{21}$$

where $\widehat{v}_2 \in H^2(B\mathcal{F}, \mathcal{Z})$ is the characteristic class capturing obstruction of lifting $\mathcal{F} = F/\mathcal{Z}$ bundles to $F$ bundles.

We will also need to consider a small extension of the above situation. Assume that the projection $\mathcal{E} \to \mathcal{E}'$ is not injective, but we can write $\mathcal{E} = \mathcal{E}_1 \times \mathcal{E}_2$ such that $\mathcal{O} \subseteq \mathcal{E}_1$, the projection of $\mathcal{E}_1$ onto $Z_G$ is injective, and the elements in $\mathcal{E}_2$ are of the form $(0, *) \in Z_G \times Z_F$. Let us define $\mathcal{E}_1', \mathcal{E}_2'$ as projections of $\mathcal{E}_1, \mathcal{E}_2$ onto $Z_G$, and $\mathcal{Z}_1, \mathcal{Z}_2$ as projections of $\mathcal{E}_1, \mathcal{E}_2$ onto $Z_G$. Notice that $\mathcal{E}_2' = 0$, $\mathcal{E}' = \mathcal{E}_1' \simeq \mathcal{E}_1$, $\mathcal{Z}_2 \simeq \mathcal{E}_2$ and $\mathcal{Z} = \mathcal{Z}_1 \times \mathcal{Z}_2$. These groups sit in a short exact sequence

$$0 \to \mathcal{O} \to \mathcal{E}_1 \to \mathcal{Z}_1 \to 0. \tag{22}$$

We can decompose $v_2 \in H^2(M, \mathcal{Z}_1 \times \mathcal{Z}_2)$ as $v_2 = v_{2,1} + v_{2,2}$ where $v_{2,1} \in H^2(M, \mathcal{Z}_1)$ describes the obstruction of lifting an $F/\mathcal{Z}$ bundle to an $F/\mathcal{Z}_2$ bundle and $v_{2,2} \in H^2(M, \mathcal{Z}_2)$ describes the obstruction of lifting an $F/\mathcal{Z}$ bundle to an $F/\mathcal{Z}_1$ bundle. We sum over $G/\mathcal{E}'$ bundles such

that $v_2' = v_{2,1}$ where $v_2' \in H^2(M, \mathcal{Z}_1)$ captures the obstruction of lifting $G/\mathcal{E}'$ bundles to $G/\mathcal{O}'$ bundles. This leads us to a 2-group with

$$\delta B = \text{Bock}(v_{2,1}), \tag{23}$$

where Bock is the Bockstein associated to (22). The Postnikov class is

$$\text{Bock}(\widehat{v}_{2,1}) \in H^3(B\mathcal{F}, \mathcal{O}), \tag{24}$$

where $\widehat{v}_{2,1} \in H^2(B\mathcal{F}, \mathcal{Z}_1)$ is the characteristic class capturing obstruction of lifting $\mathcal{F} = F/\mathcal{Z}$ bundles to $F/\mathcal{Z}_2$ bundles. We recover the previous case, where $\mathcal{E} \to \mathcal{E}'$ is an injective map, if $\mathcal{E}_2 = 0$.

**Summary criteria for 2-group symmetries.** In summary, we can provide two clear criteria when a non-trivial 2-group symmetry is present. In the first criterion, we assume that the following two conditions are satisfied:

- The projection map $\pi_G : \mathcal{E} \to Z_G$ is an injective homomorphism.

- The Postnikov class (21) is a non-trivial element of $H^3(B\mathcal{F}, \mathcal{O})$. In particular, the short exact sequence $0 \to \mathcal{O} \to \mathcal{E} \to \mathcal{Z} \to 0$ cannot split, i.e. one cannot write $\mathcal{E} \simeq \mathcal{O} \times \mathcal{Z}$, since in that case the Bockstein homomorphisms associated to the short exact sequence become trivial and the Postnikov class (21) must be trivial.

If these two conditions are satisfied, then we have a non-trivial 2-group structure between 1-form symmetry group $\mathcal{O}$ and 0-form symmetry group $\mathcal{F}$ with Postnikov class (21).

In the second criterion, we assume that the following two conditions are satisfied:

- We can write $\mathcal{E} = \mathcal{E}_1 \times \mathcal{E}_2$ such that $\mathcal{O} \subseteq \mathcal{E}_1$, the projection map $\pi_G : \mathcal{E}_1 \to Z_G$ is an injective homomorphism, and the projection $\pi_G(\mathcal{E}_2) = 0$.

- The Postnikov class (24) is a non-trivial element of $H^3(B\mathcal{F}, \mathcal{O})$. In particular, the short exact sequence $0 \to \mathcal{O} \to \mathcal{E}_1 \to \mathcal{Z}_1 \to 0$ cannot split.

If these two conditions are satisfied, then we have a non-trivial 2-group structure between the 1-form symmetry group $\mathcal{O}$ and the 0-form symmetry group $\mathcal{F}$ with Postnikov class (24).

We emphasize that the first condition in both criteria is a technical assumption. It should be possible to drop this assumption and study the most general case, but all the examples appearing in this paper can be studied using the above two criteria.

## 2.2 Application to 5d SCFTs

We now analyze global forms of flavor symmetry groups and 2-groups for 5d SCFTs and 5d SQFTs obtained by mass-deforming the 5d SCFTs.

### 2.2.1 At Conformal Point

In this paper, we apply the general setup discussed in the previous subsection to 5d SCFTs that can be constructed as M-theory compactified on singular Calabi-Yau three-folds that admit crepant (i.e. Calabi-Yau) resolutions. Actually, M-theory compactified on such a three-fold only produces a relative 5d theory which has a spectrum of extended operators that are non-local with each other. An absolute 5d theory is defined by choosing a maximal subset of extended operators that are mutually local. In this class of theories, there always exists a canonical choice such that the corresponding absolute 5d SCFT may have 1-form symmetries

but cannot have 2-form symmetries. In the M-theory realization this is realized by the choice of asymptotic $G_4$-fluxes [20]. It is believed that there are no further obstructions that disallow this choice[10]. In this paper, a 5d SCFT always refers to a theory in the above-discussed class of M-theory compactifications and furthermore, unless otherwise stated, it is always assumed that it is the absolute theory obtained by making the above-mentioned canonical choice for the spectrum of extended operators.

Consider such a 5d SCFT $\mathfrak{T}$. Let us denote the non-abelian part of the 0-form flavor symmetry algebra of $\mathfrak{T}$ as $\mathfrak{f}$ [11]. Let the simply connected group associated to $\mathfrak{f}$ be denoted as $F$, and the center of $F$ be denoted as $Z_F$. Also, let $\mathcal{O}_{\mathfrak{T}}$ denote the 1-form symmetry group of $\mathfrak{T}$. The fully resolved threefold $X_{\mathfrak{T}}$ associated to $\mathfrak{T}$ carries irreducible compact Kähler surfaces $\boldsymbol{S}_i$ labeled by $i = 1, 2, \cdots, r$ and irreducible $\mathbb{P}^1$ fibered non-compact Kähler surfaces $\boldsymbol{N}_i$ labeled by $i = 1, 2, \cdots, r_f$, where $r_f$ is the rank of $\mathfrak{f}$ and $r$ is often called the "rank of 5d SCFT $\mathfrak{T}$". The full dictionary was determined in [46], with the complete characterization of the Coulomb branch to geometry determined in [47]. In addition $X_{\mathfrak{T}}$ contains various compact curves $C_i$, with the $\mathbb{P}^1$ fibers $f_i$ of $\boldsymbol{N}_i$ forming a subset of all compact curves $C_i$. At the conformal point of $\mathfrak{T}$, the volumes of all $\boldsymbol{S}_i, C_i$ are zero.

Let us now move onto a generic Coulomb branch (CB) vacuum of $\mathfrak{T}$ [12]. The compact surfaces $\boldsymbol{S}_i$ acquire non-zero volumes. On the other hand, the $\mathbb{P}^1$ fibers $f_i$ of $\boldsymbol{N}_i$ remain at zero volume, since their volumes are proportional to mass parameters of $\mathfrak{T}$. The M-theory 3-form gauge field $C_3$ reduced along $\boldsymbol{S}_i$ gives rise to a dynamical ordinary (1-form) gauge field which is associated to a $U(1)_i^{(g)}$ gauge *group*. The above discussed canonical choice for the absolute 5d theory is realized as follows: for every $i$, there exists a Wilson line operator $W_i$ which has charge $+1$ under $U(1)_i^{(g)}$ and charge 0 under $U(1)_j^{(g)}$ for $j \neq i$. In other words, for every $i$, a single M2-brane is allowed to wrap the non-compact 1-cycle $C_i$ which is Poincare dual to the surface $\boldsymbol{S}_i$. Thus, at a generic point in CB, we obtain a 5d $\mathcal{N} = 1$ abelian gauge theory with gauge group $G = \prod_i U(1)_i^{(g)} \simeq U(1)^r$ and non-abelian part of flavor symmetry algebra being $\mathfrak{f}$. M2-branes wrapping compact curves in $X_{\mathfrak{T}}$ give rise to matter content for the abelian gauge theory[13]. The $U(1)_i^{(g)}$ charge $q_i^{(g)}(C)$ of matter content associated to a curve $C$ can be obtained as

$$q_i^{(g)}(C) = -\boldsymbol{S}_i \cdot C, \tag{25}$$

where $\boldsymbol{S}_i \cdot C$ is the intersection number between $\boldsymbol{S}_i$ and $C$ inside $X_{\mathfrak{T}}$. This is also the charge under the center $Z_G$ of $G$ since $Z_G = G$. Similarly, one can also determine $Z_a^{(f)}$ charge $q_a^{(f)}(C)$ of matter content associated to $C$ where $Z_F = \prod_a Z_a^{(f)}$ such that each $Z_a^{(f)} \simeq \mathbb{Z}_{n_a}$ for some $n_a > 1$. We have [21]

$$q_a^{(f)}(C) = -\boldsymbol{N}_a \cdot C \pmod{n_a}, \tag{26}$$

such that

$$\boldsymbol{N}_a = \sum_i \alpha_{a,i} \boldsymbol{N}_i, \tag{27}$$

where the coefficients $\alpha_{a,i}$ are collected in appendix A.2.

We can now apply the analysis of section 2.1 to the above gauge theory with gauge group $G = \prod_i U(1)_i^{(g)} \simeq U(1)^r$ and matter content provided by all compact curves $C_i$ of $X_{\mathfrak{T}}$. This

---

[10]But other choices related to this canonical choice by gauging of 1-form symmetries may be obstructed, for example by 't Hooft anomaly of 1-form symmetry etc.

[11]We believe that including the abelian parts, which exist in some SCFTs, will lead to futher extension of this result. However, we leave the incorporation of abelian symmetries for future work since we currently do not completely understand how abelian symmetries are encoded in M-theory.

[12]Note that we do not turn on any mass parameters.

[13]The matter content associated to a compact curve $C$ is massless or massive respectively depending on whether $C$ has zero volume or non-zero volume.

method was used by [20] to determine the 1-form symmetry group $\mathcal{O}_{\mathfrak{T}}$ of the 5d SCFT $\mathfrak{T}$ by proposing that

$$\mathcal{O}_{\mathfrak{T}} \simeq \mathcal{O}, \tag{28}$$

where $\mathcal{O}$ is the group appearing in the analysis of section 2.1. In a similar vein, we propose that the 0-form symmetry group $\mathcal{F}_{\mathfrak{T}}$ of the 5d SCFT $\mathfrak{T}$ is[14]

$$\mathcal{F}_{\mathfrak{T}} \simeq \mathcal{F}, \tag{29}$$

where $\mathcal{F}$ is obtained by applying the analysis of section 2.1 to the abelian gauge theory arising on the CB of $\mathfrak{T}$. Moreover, we propose that the 0-form symmetry $\mathcal{F}_{\mathfrak{T}}$ and 1-form symmetry $\mathcal{O}_{\mathfrak{T}}$ of the 5d SCFT form a 2-group structure with Postnikov class (24). The 2-group structure is non-trivial if the Postnikov class is non-trivial.

Our proposals for $\mathcal{F}_{\mathfrak{T}}$ and 2-group structures are justified by the fact that we have included the contributions from all the curves in the geometry to compute these. This means that we have included the contributions of arbitrarily massive states[15] in the theory (e.g. massive BPS particles and their non-bound combinations). Thus, $\mathcal{F}_{\mathfrak{T}}$ and 2-group structure continue to hold at arbitrarily high energies leading us to conclude that $\mathcal{F}_{\mathfrak{T}}$ and 2-group structure should be regarded as properties of the UV 5d SCFT [16].

### 2.2.2 After Mass Deformations

We can also consider 5d supersymmetric quantum field theories (SQFTs) obtained by mass deforming 5d SCFTs. Let us consider a mass deformation such that the resulting SQFT $\mathfrak{T}'$ has flavor symmetry $\mathfrak{f}' \oplus \mathfrak{u}(1)^{r_f - r'_f} \subset \mathfrak{f}$ where $\mathfrak{f}'$ is a non-abelian flavor algebra whose Dynkin diagram embeds into the Dynkin diagram of $\mathfrak{f}$. The subalgebra $\mathfrak{f}'$ is associated to irreducible non-compact surfaces $N_{i'}$ for $i' = 1, \cdots, r'_f$ where $r'_f$ is rank of $\mathfrak{f}'$. The $\mathbb{P}^1$ fibers $f_{i'}$ of $N_{i'}$ remain at zero volume under the deformation. On the other hand, the $\mathbb{P}^1$ fibers $f_q$ of remaining $r_f - r'_f$ irreducible non-compact surfaces $N_q$ for $q = 1, \cdots, r_f - r'_f$ acquire a non-zero volume under the deformation.

Let the simply connected group associated to $\mathfrak{f}' \oplus \mathfrak{u}(1)^{r_f - r'_f}$ be $F' \times U(1)^{r_f - r'_f}$ with the center of $F'$ being $Z'_F$. Naively one might think that $U(1)^{r_f - r'_f}$ is associated to the surfaces $N_q$. However, the $\mathbb{P}^1$ fibers $f_{i'}$ of $\mathcal{N}_{i'}$ are in general charged under $N_q$. As a consequence the centers $Z'_F$ and $U(1)^{r_f - r'_f}$ mix with each other. To rectify this, one defines[17] the surfaces $N'_q$

$$N'_q := n_q \left( N_q + \sum_{i'} \alpha_{q,i'} N_{i'} \right), \tag{30}$$

where the coefficients $\alpha_{q,i'} \in \mathbb{Q}$ are determined by requiring that

$$N'_q \cdot f_{i'} = 0 \qquad \forall \ i' \tag{31}$$

---

[14]It should be noted that this is only the "non-abelian part" of the full 0-form flavor symmetry group of $\mathfrak{T}$. Say the full flavor algebra is $\mathfrak{f} \oplus \mathfrak{u}(1)^k$ where $\mathfrak{f}$ is non-abelian. The full flavor group can then be expressed as $\frac{\mathcal{F}_{\mathfrak{T}} \times U(1)^k}{\Gamma}$ where $\Gamma$ is some subgroup of the center of $\mathcal{F}_{\mathfrak{T}} \times U(1)^k$ such that the kernel of the projection map from $\Gamma$ to the center of $U(1)^k$ is trivial. If $k = 0$, then the full flavor group is $\mathcal{F}_{\mathfrak{T}}$.

[15]We emphasize that it is not important for these states to be supersymmetric. Our argument only requires us to identify the lattice spanned by flavor and gauge charges of arbitrary states in the theory which is known if the intersections of all the curves and surfaces in the threefold are known.

[16]To mitigate potential confusions, let us emphasize again that we have not turned on any mass parameters, so we have *not* deformed the 5d SCFT. We have simply chosen a non-conformal CB vacuum for the conformal theory (to perform the computation for $\mathcal{F}_{\mathfrak{T}}$ and 2-group), so the UV microscopic defining theory is the SCFT itself (albeit placed in a non-conformal vacuum).

[17]This is related to the Shioda map in F-theory.

and $n_q$ is the smallest positive integer such that

$$N'_q \cdot C_i \in \mathbb{Z} \tag{32}$$

for all compact curves $C_i$. The $U(1)^{r_f - r'_f}$ factor is then associated to the new non-compact surfaces $N'_q$.

Let us now move onto a generic point on the CB of $\mathfrak{T}'$. The compact surfaces $S_i$ acquire non-zero volumes while the $\mathbb{P}^1$ fibers $f_{i'}$ remain at zero volume. As before, we obtain a 5d $\mathcal{N} = 1$ abelian gauge theory with gauge group $G = \prod_i U(1)_i^{(g)} \simeq U(1)^r$. The matter content for the gauge theory is again specified by compact curves $C_i$. We can now again apply the formalism of section 2.1 to this abelian gauge theory with $F := F' \times U(1)^{r_f - r'_f}$ with a crucial ingredient being that the $U(1)^{r_f - r'_f}$ charges of matter content associated to a compact curve $C_i$ are obtained by computing $-N'_q \cdot C_i$. We then propose that the 0-form flavor symmetry group[18] $\mathcal{F}_{\mathfrak{T}'}$ of the SQFT $\mathfrak{T}'$ is

$$\mathcal{F}_{\mathfrak{T}'} \simeq \mathcal{F}, \tag{33}$$

where $\mathcal{F}$ is obtained by applying the analysis of section 2.1 to the above abelian gauge theory arising on the CB of $\mathfrak{T}'$. The 1-form symmetry group $\mathcal{O}_{\mathfrak{T}'}$ of $\mathfrak{T}'$ is again identified with $\mathcal{O}$ appearing in the analysis of section 2.1 and we have $\mathcal{O}_{\mathfrak{T}'} \simeq \mathcal{O}_{\mathfrak{T}}$. Moreover, we propose that the 0-form symmetry $\mathcal{F}_{\mathfrak{T}'}$ and 1-form symmetry $\mathcal{O}_{\mathfrak{T}'}$ of the 5d SQFT $\mathfrak{T}'$ form a 2-group structure with Postnikov class (24) obtained after applying the analysis of section 2.1 to the above abelian gauge theory. The 2-group structure is non-trivial if the Postnikov class is non-trivial.

### 2.2.3 From 5d $\mathcal{N} = 1$ Non-Abelian Gauge Theory

Consider a 5d SCFT $\mathfrak{T}$, which admits a mass deformation such that the theory after the deformation reduces in the IR to a 5d $\mathcal{N} = 1$ non-abelian gauge theory $\mathfrak{G}$ with a simple[19] gauge algebra $\mathfrak{g}$ and simply connected gauge group $G$. Furthermore, assume that the flavor symmetry algebra $\mathfrak{f}_{\mathfrak{T}}$ of the SCFT can be understood as

$$\mathfrak{f}_{\mathfrak{T}} = \mathfrak{f}_{\text{hyp}} \oplus \mathfrak{u}(1)_I, \tag{34}$$

where $\mathfrak{f}_{\text{hyp}}$ is the flavor symmetry algebra associated to hypermultiplet content of $\mathfrak{G}$, and $\mathfrak{u}(1)_I$ is the flavor symmetry algebra associated to the instanton symmetry of $\mathfrak{G}$ generated by the instanton current $\text{Tr}(F \wedge F)$. For a general 5d SCFT, we only have $\mathfrak{f}_{\text{hyp}} \oplus \mathfrak{u}(1)_I \subseteq \mathfrak{f}_{\mathfrak{T}}$ with the rank of $\mathfrak{f}_{\mathfrak{T}}$ equal to the rank of $\mathfrak{f}_{\text{hyp}} \oplus \mathfrak{u}(1)_I$, but in what follows in this subsection we assume that (34) holds.

In such a situation, we can obtain the 1-form symmetry group $\mathcal{O}_{\mathfrak{T}}$, the 0-form symmetry group $\mathcal{F}_{\mathfrak{T}}$ associated to the non-abelian part $\mathfrak{f}_{\text{hyp}}$ of the flavor symmetry algebra $\mathfrak{f}_{\mathfrak{T}}$ of the 5d SCFT $\mathfrak{T}$, and the 2-group structure between $\mathcal{O}_{\mathfrak{T}}$ and $\mathcal{F}_{\mathfrak{T}}$, by applying the methods of section 2.1 to the gauge theory $\mathfrak{G}$ with matter content provided by hypermultiplet content of $\mathfrak{G}$ and an extra massive BPS particle which is an instanton of $G$. The charge $q_g^{\text{inst}}$ of the instanton under the center $Z_G$ of $G$ can be taken to be as follows [21]:

- $q_g^{\text{inst}} = k - \frac{A(R)}{2} \pmod n$ if $G = SU(n)$, $n \neq 6$ with CS level $k$. Here $R$ is the representation formed by all the full hypers in $\mathfrak{G}$ and $A(R)$ is the anomaly coefficient associated to $R$ (see [48] for more details). Note that we do not need to worry about the existence

---

[18]Even though we have included contribution of the abelian factors in the decomposition $\mathfrak{f} \to \mathfrak{f}' \oplus \mathfrak{u}(1)^{r_f - r'_f}$, we have still not included contributions from the abelian factors in the flavor symmetry of the theory at the conformal point. See the last footnote in previous subsection for more discussion.

[19]One can also relax the constraints on $\mathfrak{g}$ to include non-simple $\mathfrak{g}$, but we do not expand on this general case here.

of genuine half-hypers that cannot be paired into full hypers since we assume that $\mathfrak{G}$ descends from a 5d SCFT $\mathfrak{T}$, as according to the classification of [42] such a situation does not for $n \neq 6$.

- $q_g^{\text{inst}} = k - \frac{A(R)}{2} - \frac{3}{2}s$ (mod 6) if $G = SU(6)$ with CS level $k$. Here $R$ is the representation formed by all the full hypers in $\mathfrak{G}$, $A(R)$ is the anomaly coefficient associated to $R$, and $s = 1$ if there exists a half-hyper in 3-index antisymmetric irrep of $\mathfrak{su}(6)$ that cannot be paired into a full hyper (otherwise $s = 0$).

- $q_g^{\text{inst}} = m$ (mod 2) if $G = Sp(n)$ with discrete theta angle $m\pi$. Here we are assuming that the hypermultiplet content is such that a discrete theta angle is allowed.

- Take $G = \text{Spin}(12)$ such that there exists a half-hyper in a spinor irrep $S$ of $\mathfrak{so}(12)$ that cannot be paired into a full hyper. We have $Z_G = \mathbb{Z}_2^S \times \mathbb{Z}_2^C$ such that under $\mathbb{Z}_2^S$ the spinor irrep $S$ has charge 1 (mod 2) and the cospinor irrep $C$ has charge 0 (mod 2), and under $\mathbb{Z}_2^C$ the spinor irrep $S$ has charge 0 (mod 2) and $C$ has charge 1 (mod 2). In such a situation, we have $q_g^{\text{inst}} = 0$ (mod 2) under $\mathbb{Z}_2^S$ and $q_g^{\text{inst}} = 1$ (mod 2) under $\mathbb{Z}_2^C$.

- $q_g^{\text{inst}} = 0$ otherwise.

The charges $q_f^{\text{inst}}$ of the instanton under the centers of the simply connected groups associated to simple, non-abelian factors in $\mathfrak{f}_{\text{hyp}}$ can be taken to be as follows:

- Consider $G = \text{Spin}(2n)$ with matter content containing $m$ hypers transforming in vector irrep of $G$. Then $q_f^{\text{inst}} = m$ (mod 2) under the $\mathbb{Z}_2$ center of the simply connected group $Sp(m)$ associated to the flavor symmetry algebra $\mathfrak{sp}(m)$ rotating these $m$ hypers.

- Consider $G = SU(4)$ with matter content containing $m$ hypers transforming in 2-index antisymmetric irrep of $G$. Then $q_f^{\text{inst}} = m$ (mod 2) under the $\mathbb{Z}_2$ center of the simply connected group $Sp(m)$ associated to the flavor symmetry algebra $\mathfrak{sp}(m)$ rotating these $m$ hypers.

- Consider $G = Sp(n)$ with matter content containing $2m$ hypers transforming in fundamental irrep of $G$. Then $q_f^{\text{inst}} = \left(1 \ (\text{mod } 2), 0 \ (\text{mod } 2)\right)$ under the $\mathbb{Z}_2^S \times \mathbb{Z}_2^C$ center of the simply connected group $\text{Spin}(4m)$ associated to the flavor symmetry algebra $\mathfrak{so}(4m)$ rotating these $2m$ hypers. Note that the fundamental hypers have charges $\left(1 \ (\text{mod } 2), 1 \ (\text{mod } 2)\right)$ under the $\mathbb{Z}_2^S \times \mathbb{Z}_2^C$.

- Consider $G = Sp(n)$ with matter content containing $2m+1$ hypers transforming in fundamental irrep of $G$. Then $q_f^{\text{inst}} = 1$ (mod 4) under the $\mathbb{Z}_4$ center of the simply connected group $\text{Spin}(4m+2)$ associated to the flavor symmetry algebra $\mathfrak{so}(4m+2)$ rotating these $2m+1$ hypers.

- Consider $G = Sp(3)$ with matter content containing $2m+1$ *half*-hypers transforming in fundamental irrep of $G$ and 1 half-hyper transforming in 3-index antisymmetric irrep of $G$. Then $q_f^{\text{inst}} = 1$ (mod 2) under the $\mathbb{Z}_2$ center of the simply connected group $\text{Spin}(2m+1)$ associated to the flavor symmetry algebra $\mathfrak{so}(2m+1)$ rotating the $2m+1$ half-hypers.

- Consider $G = \text{Spin}(7)$ with matter content containing $m$ hypers transforming in spinor irrep of $G$. Then $q_f^{\text{inst}} = m$ (mod 2) under the $\mathbb{Z}_2$ center of the simply connected group $Sp(m)$ associated to the flavor symmetry algebra $\mathfrak{sp}(m)$ rotating these $m$ hypers.

- Consider $G = \mathrm{Spin}(8)$ with matter content containing $m$ hypers transforming in the spinor/cospinor irrep of $G$. Then $q_f^{\mathrm{inst}} = m$ (mod 2) under the $\mathbb{Z}_2$ center of the simply connected group $Sp(m)$ associated to the flavor symmetry algebra $\mathfrak{sp}(m)$ rotating these $m$ hypers.

- For $G = E_7$ with matter content containing $m$ *half*-hypers transforming in irrep of dimension **56** of $G$, we do not have a definite answer for $q_f^{\mathrm{inst}}$ under the center of the simply connected group $\mathrm{Spin}(m)$ associated to the flavor symmetry algebra $\mathfrak{so}(m)$ rotating the $m$ half-hypers, since at the time of writing this paper we do not know the intersection properties of $\mathbb{P}^1$ fibered non-compact surfaces realizing the $\mathfrak{so}(2m+1)$ flavor symmetry in this case.

- $q_f^{\mathrm{inst}} = 0$ for all other cases.

The above rules for $q_f^{\mathrm{inst}}$ are derived in appendix A.3. Now we can apply the analysis of section 2.1 to the 5d non-abelian gauge theory $\mathfrak{G}$ with matter content specified by hypers and the instanton particle whose charges have been collected above. In this way, one would obtain the same results for the 5d SCFT $\mathfrak{T}$ as obtained in section 2.2.1 where the analysis of section 2.1 was instead applied to the 5d *abelian* gauge theory arising on the CB of the 5d SCFT $\mathfrak{T}$.

## 2.3 0-form/2-form Symmetry Mixed Anomaly

If the 2-group symmetry in a 5d SCFT $\mathfrak{T}$ does not have a 't Hooft anomaly, then it can be thought of alternatively as a mixed 't Hooft anomaly between 0-form and 2-form symmetries of the 5d SCFT $\widetilde{\mathfrak{T}}$ obtained after gauging the 1-form symmetry $\mathcal{O}_{\mathfrak{T}}$ of $\mathfrak{T}$ [25]. The anomaly can be captured by

$$\omega_{\mathfrak{T}} = \int_{M_6} \mathrm{Bock}(\widehat{v}_{2,1}) \cup B_3 \,, \tag{35}$$

where $M_6$ is a manifold whose boundary is the spacetime $M_5$, $B_3$ describes a background for the $\mathcal{O}_{\mathfrak{T}}$ *2-form symmetry* of $\widetilde{\mathfrak{T}}$, and $\widehat{v}_{2,1}$, $\mathrm{Bock}$ are discussed towards the end of section 2.1.

This can be established quite concretely in the gauge theory context of section 2.1. After gauging 1-form symmetry $\mathcal{O}'$, one obtains a gauge theory with gauge group $\widetilde{G} = G/\mathcal{O}'$ and a dual $(d-3)$-form symmetry $\mathcal{O}'$ where $d$ is the dimension of spacetime $M$. A non-trivial background $B_{d-2} \in H^{d-2}(M, \mathcal{O}')$ is turned on by adding a term to the Lagrangian of the form

$$w_2 \cup B_{d-2} \,, \tag{36}$$

where $w_2 \in H^2(M, \mathcal{O}')$ captures the obstruction of lifting $\widetilde{G} = G/\mathcal{O}'$ bundles to $G$ bundles.

The center $Z_{\widetilde{G}}$ of $\widetilde{G}$ is

$$Z_{\widetilde{G}} = \frac{Z_G}{\mathcal{O}'} \,. \tag{37}$$

Furthermore, the structure group of the theory is again defined as

$$\widetilde{S} = \frac{\widetilde{G} \times F}{\widetilde{\mathcal{E}}} \,, \tag{38}$$

where

$$\widetilde{\mathcal{E}} = \frac{\mathcal{E}}{\mathcal{O}} \,. \tag{39}$$

The projection of $\widetilde{E}$ onto $Z_F$ is still $\mathcal{Z}$, thus the 0-form symmetry group $\mathcal{F}$ is left unchanged

$$\mathcal{F} = \frac{F}{\mathcal{Z}} \tag{40}$$

as expected. The projection $\widetilde{\mathcal{E}}'$ of $\widetilde{\mathcal{E}}$ onto $Z_{\widetilde{G}}$ is

$$\widetilde{\mathcal{E}}' \simeq \frac{\mathcal{E}'}{\mathcal{O}'} \simeq \mathcal{Z}_1 \,, \tag{41}$$

where the definition of $\mathcal{Z}_1$ can be found in section 2.1.

Let us turn on a 0-form symmetry background described by a $F/\mathcal{Z}$ bundle with $v_2 \in H^2(M, \mathcal{Z})$ capturing the obstruction of lifting the $F/\mathcal{Z}$ bundle to an $F$ bundle. From section 2.1, $\mathcal{Z}$ decomposes as $\mathcal{Z}_1 \times \mathcal{Z}_2$, due to which $v_2$ decomposes as $v_{2,1} + v_{2,2}$ where $v_{2,1} \in H^2(M, \mathcal{Z}_1)$ describes the obstruction of lifting an $F/\mathcal{Z}$ bundle to an $F/\mathcal{Z}_2$ bundle and $v_{2,2} \in H^2(M, \mathcal{Z}_2)$ describes the obstruction of lifting an $F/\mathcal{Z}$ bundle to an $F/\mathcal{Z}_1$ bundle. Now, from $\widetilde{S}$ we see that the gauge theory sums over $\widetilde{G}/\widetilde{\mathcal{E}}'$ bundles with

$$v_2' = v_{2,1} \,, \tag{42}$$

where $v_2' \in H^2(M, \mathcal{Z}_1)$ captures the obstruction of lifting $\widetilde{G}/\widetilde{\mathcal{E}}'$ bundles to $\widetilde{G}$ bundles. As before, we have

$$\delta w_2 = \text{Bock}(v_{2,1}) \,, \tag{43}$$

where Bock is the Bockstein associated to

$$0 \to \mathcal{O}' \to \mathcal{E}_1' \to \mathcal{Z}_1 \to 0 \,. \tag{44}$$

Combining (43) with (36), we obtain the anomaly

$$\omega = \int_{M_{d+1}} \text{Bock}(v_{2,1}) \cup B_{d-2} \,, \tag{45}$$

where $M_{d+1}$ is a manifold whose boundary is the $d$-dimensional spacetime $M$.

We can apply this general analysis to the 5d $\mathcal{N} = 1$ abelian gauge theory arising at a generic point on CB of the 5d SCFT $\mathfrak{T}$ discussed in section 2.2. It then seems reasonable to propose that the anomaly (45) visible at the level 5d $\mathcal{N} = 1$ abelian gauge theory lifts to the anomaly (35) at the level of the 5d SCFT $\widetilde{\mathfrak{T}}$.

# 3 The $SU(2)_0$ SCFT

The first example we consider is the 5d SCFT $\mathfrak{T}$ that admits a mass deformation upon which it reduces in the IR to a pure $SU(2)$ gauge theory with discrete theta angle 0. It is the simplest 5d SCFT with a non-trivial 1-form symmetry and whose 0-form flavor symmetry algebra contains a non-trivial non-abelian component. We show that the global form of 0-form symmetry group of this theory is $\mathcal{F} = SO(3)$. Moreover, the $\mathbb{Z}_2$ 1-form and $SO(3)$ 0-form symmetries of this 5d SCFT form a 2-group. We provide evidence for this 0-form symmetry and the existence of 2-group by noticing that this 2-group symmetry can also be easily seen from a special singular point in the Coulomb branch of the 4d $\mathcal{N} = 2$ KK theory obtained by compactifying this 5d SCFT on a circle of non-zero size.

We also discuss a mixed anomaly between the $U(1)_I$ 0-form and 1-form symmetries that arises in the 5d $\mathcal{N} = 1$ $SU(2)_0$ non-abelian gauge theory arising in the IR after mass deforming the SCFT $\mathfrak{T}$ [18]. This anomaly might lift to an anomaly of the 2-group at the conformal point, or the theory might contain an additional topological sector that in the IR also contributes and eventually cancels the anomaly. In the latter case the total anomaly vanishes and hence the 2-group symmetry at the conformal point can be non-anomalous without spoiling consistency. We discuss such a potential anomaly cancellation mechanism. If this mechanism is realized

leading to a non-anomalous 2-group in the 5d SCFT $\mathfrak{T}$, then the 5d SCFT $\widetilde{\mathfrak{T}}$ (which, after a mass deformation, reduces in the IR to a pure $SO(3)_0$ gauge theory) produced by gauging the center 1-form symmetry of $\mathfrak{T}$ also has $SO(3)$ 0-form symmetry group and a $\mathbb{Z}_2$ 2-form symmetry group with a mixed 0-2-form symmetry anomaly.

We also study some related 5d SCFTs, namely the rank 1 Seiberg theories, that reduce, after a mass deformation, to $SU(2)$ gauge theories with $n$ full hypers in fundamental representation, where $1 \leq n \leq 7$. These theories do not admit a 1-form symmetry, and hence do not admit any 2-group symmetries. We determine the global form of the 0-form flavor symmetry group associated to the non-abelian part of flavor symmetry algebra of these theories.

## 3.1 Various Phases and Their Associated Geometry

Consider the 5d SCFT $\mathfrak{T}$, which is the UV completion of 5d $\mathcal{N} = 1$ pure $SU(2)$ gauge theory with vanishing discrete theta angle. The corresponding geometry is

$$N \xrightarrow{\quad f^N \qquad\quad e \quad} \mathbf{1}_2 = S_1 \ , \tag{46}$$

where $N$ is a non-compact surface (which has a $\mathbb{P}^1 = f^N$-fibration over $\mathbb{C}$), and a compact surface $S_1$ indicated by $\mathbf{1}_2$, which is a Hirzebruch surface $\mathbb{F}_2$. Alternatively the toric diagram is

$$SU(2)_0 : \qquad \tag{47}$$

In general in $\mathbb{F}_n$ the cone of effective curves is spanned by two rational curves $e$ (shown in blue) and $f$ (green) with intersection numbers in $\mathbb{F}_n$

$$e \cdot_{\mathbb{F}_n} e = -n \,, \qquad f \cdot_{\mathbb{F}_n} f = 0 \,, \qquad e \cdot_{\mathbb{F}_n} f = 1 \,. \tag{48}$$

The surfaces are glued along $f^N$ and $e$, respectively. From this it follows from the intersections of the surfaces with the curves, what their charges under gauge and flavor groups discussed later are:

| Curve in $S_1$ | $U(1)$ | $U(1)_N$ | $Z_F \simeq \mathbb{Z}_2$ | $U(1)_I$ |
|:---:|:---:|:---:|:---:|:---:|
| $e$ | 0 | 2 | 0 (mod 2) | 1 |
| $f$ | 2 | $-1$ | 1 (mod 2) | 0 |

$$\tag{49}$$

Note that the intersection numbers between surfaces and curves are computed as follows. If a compact curve $C$ lies in a compact surface $D$, then

$$D \cdot C = K_D \cdot_D C = 2g(C) - 2 - C \cdot_D C \,, \tag{50}$$

where $\cdot_D$ denotes intersection number of curves inside the surface $D$, $K_D$ is the dual of canonical class of $D$ and $g(C)$ is the genus of $C$. If $D$ is a Hirzebruch surface $\mathbb{F}_n$, then we can also use the expression

$$K_{\mathbb{F}_n} = -2e - (n+2)f \tag{51}$$

to evaluate the above intersection number. On the other hand, if a compact curve $C$ lies in a compact or non-compact surface $D_1$, and we want to evaluate its intersection number with some other compact or non-compact surface $D_2$, then

$$D_2 \cdot C = C_{12} \cdot_{D_1} C \,, \tag{52}$$

where $C_{12}$ the curve in $D_1$ that glues $D_1$ to $D_2$, or in other words, that describes the intersection locus of $D_1$ and $D_2$ from the perspective of $D_1$.

At the conformal point, both the curves $e$ and $f$ have zero volume, so M2-branes wrapping them provide some massless "matter" contribution[20] to the conformal theory[21]. Since $e$ is identified with $f^N$, the $\mathbb{P}^1$ fiber $f^N$ of the non-compact surface $N$ also has zero volume. This signals that the conformal theory has an $\mathfrak{f} = \mathfrak{su}(2)$ 0-form flavor symmetry algebra. The charge $q_F(C)$ of massless "matter" corresponding to curve $C$ (where $C$ is either $e$ or $f$) under the center $Z_F = \mathbb{Z}_2$ of the simply connected group $F = SU(2)$ associated to $\mathfrak{f}$ is computed as

$$q_F(C) = -N \cdot C \;(\text{mod } 2). \tag{53}$$

We can compute

$$q_F(e) = -N \cdot e \;(\text{mod } 2) = -e \cdot_{S_1} e \;(\text{mod } 2) = 2 \;(\text{mod } 2), \tag{54}$$

and

$$q_F(f) = -N \cdot f \;(\text{mod } 2) = -e \cdot_{S_1} f \;(\text{mod } 2) = -1 \;(\text{mod } 2). \tag{55}$$

Going onto the Coulomb branch (CB) of the conformal theory corresponds to providing non-zero volume to the $f$ curve, but $e$ curve remains at zero volume. Consequently, the compact surface $S_1$ obtains a non-zero volume, and the low-energy theory is a $U(1)$ gauge theory whose gauge field is obtained by compactifying the M-theory 3-form gauge field $C_3$ on $S_1$. M2-brane wrapping $f$ now produces a massive BPS particle since the volume of $f$ is non-zero. Similarly, wrapping M2-brane on $e + f$ produces another massive BPS particle whose mass is the same as the mass of BPS particle associated to $f$. The charge $q(C)$ of BPS particle corresponding to curve $C$ (where $C$ is either $f$ or $e + f$) under $U(1)$ gauge group is computed as

$$q(C) = -S_1 \cdot C, \tag{56}$$

We can compute

$$q(f) = -S_1 \cdot f = -2g(f) + 2 + f \cdot_{S_1} f = 0 + 2 + 0 = 2, \tag{57}$$

and to compute $q(e + f)$, we compute $q(e)$

$$q(e) = -S_1 \cdot e = -2g(e) + 2 + e \cdot_{S_1} e = 0 + 2 - 2 = 0, \tag{58}$$

which leads to

$$q(e + f) = q(e) + q(f) = 2. \tag{59}$$

Since $f^N$ remains at zero volume, the CB theory has an $\mathfrak{f} = \mathfrak{su}(2)$ 0-form flavor symmetry algebra. The charges under the center $Z_F$ are computed as in (53). The BPS particles associated to $f$ and $e + f$ form a doublet[22] under the $\mathfrak{su}(2)$ flavor symmetry. But the $\mathbb{Z}_2$ center of $F = SU(2)$ is actually a part of the $U(1)$ gauge group since these BPS particles carry a non-trivial charge under the gauge $U(1)$. This means that the true 0-form flavor symmetry group of the theory

---

[20]It should be noted that the local operators corresponding to this "matter" are not genuine local operators since they carry gauge charges when the theory is deformed to Coulomb branch, i.e. they do not exist on their own independent of some higher-dimensional defects. Instead, they are proposed to arise at the end of a line operator $2\mathcal{L}$ which is the square of a "fundamental" line operator $\mathcal{L}$ in the 5d SCFT, and thus these local operators screen $2\mathcal{L}$. This leads to the conclusion that the 5d SCFT has a $\mathbb{Z}_2$ 1-form symmetry whose charged line operator is $\mathcal{L}$. Geometrically $\mathcal{L}$ arises by wrapping a single M2-brane on the non-compact curve dual to the compact surface $S$.

[21]Actually $e$ decouples from the conformal theory but $e + f$ does not. Since we are only concerned with the charges of $e + f$ and $f$, we can equally well work with the charges of $e$ and $f$, as we do in what follows.

[22]Indeed $e + f$ and $f$ both carry charge 1 under $N$ modulo 2.

on the CB is $\mathcal{F} = F/\mathbb{Z}_2 = SO(3)$ which should be identified as the 0-form symmetry group of the UV SCFT (see end of Section 2.2.1 for a justification).

Starting from the conformal point, one can also turn on a mass parameter to deform the 5d SCFT $\mathfrak{T}$ to a 5d SQFT $\mathfrak{T}'$. This mass parameter provides a non-zero volume to $e$, but $f$ remains at zero volume. Since $f$ is the $\mathbb{P}^1$ fiber of $S = \mathbb{F}_2$, the low-energy theory has $\mathfrak{su}(2)$ gauge algebra[23] with M2-brane wrapping $f$ providing the massless gauge bosons which combine with the gauge boson descending from $C_3$ to form gauge bosons for $\mathfrak{su}(2)$ gauge algebra. M2-brane wrapping $e + f$ leads to massive BPS particle which is identified as the BPS instanton of $SU(2)$ gauge group. Since $f^N$ has non-zero volume, the $\mathfrak{su}(2)$ flavor symmetry breaks to its $\mathfrak{u}(1)$ cartan which we denote by $\mathfrak{u}(1)_N$. The charge $q_N(C)$ of matter content associated to curve $C$ under a group $U(1)_N$ whose Lie algebra is $\mathfrak{u}(1)_N$ can be obtained by computing

$$q_N(C) = -N \cdot C. \tag{60}$$

From this, we see that the zero volume $\mathbb{P}^1$ fiber $f$ of $S$ has non-trivial charge under $U(1)_N$. This means that the true $\mathfrak{u}(1)$ symmetry is obtained by following the procedure described in section 2.2.2. We need to define a surface

$$N' := n(N + \alpha S), \tag{61}$$

such that $N' \cdot f = 0$ and $n$ is the smallest positive integer such that $N' \cdot e \in \mathbb{Z}$. Solving these conditions leads to

$$N' = \frac{1}{2}N + \frac{1}{4}S, \tag{62}$$

which describes the correct $\mathfrak{u}(1)$ symmetry which we denote by $\mathfrak{u}(1)_I$. The charge $q_I(C)$ of matter associated to curve $C$ under a group $U(1)_I$ whose Lie algebra is $\mathfrak{u}(1)_I$ can be obtained by computing

$$q_I(C) = -N' \cdot C. \tag{63}$$

The reason for subscript $I$ in $U(1)_I$ is that this symmetry can be identified as the instanton $U(1)$ symmetry of the non-abelian $SU(2)$ gauge theory, since the charge of $f$ under $U(1)_I$ is 0 and the charge of the instanton $e + f$ under $U(1)_I$ is 1.

Giving non-zero volumes to both $e$ and $f$ corresponds to moving onto the Coulomb branch of the 5d SQFT $\mathfrak{T}'$, which can also be understood as the Coulomb branch of the low-energy $SU(2)$ gauge theory. M2-brane wrapping $f$ curve gives rise to W-bosons corresponding to the $SU(2) \to U(1)$ Higgsing occuring from the viewpoint of the low energy gauge theory. The charge under gauge $U(1)$ of a BPS particle associated to curve $C$ is computed as in (56). The $\mathfrak{u}(1)_N$ flavor symmetry is now a good flavor symmetry to use as no $\mathbb{P}^1$ fiber having zero volume is charged under it anymore. The charge under $U(1)_N$ of a BPS particle associated to curve $C$ is computed as in (60).

## 3.2 2-Group Symmetry

Recall from the discussion in the previous subsection, that the low-energy theory on the Coulomb branch (without turning on any mass paramters) of the 5d SCFT $\mathfrak{T}$ is a $U(1)$ gauge theory with two massive BPS particles $f$ and $e + f$ of charge 2 and equal mass. Moreover, the two particles form a doublet under an $\mathfrak{f} = \mathfrak{su}(2)$ 0-form flavor symmetry, and hence have charge 1 (mod 2) under the center $Z_F = \mathbb{Z}/2\mathbb{Z}$ of the simply connected group $F = SU(2)$

---

[23]Since we have chosen the 5d SCFT to have a "fundamental" line operator $\mathcal{L}$ charged under a $\mathbb{Z}_2$ 1-form symmetry, the gauge group is $SU(2)$, which allows for a Wilson line in fundamental representation charged non-trivially under the center of $SU(2)$.

associated to $\mathfrak{f}$. Thus, the full structure group (combining the gauge and flavor parts) acting faithfully on the theory can be written as

$$S = \frac{U(1) \times F}{\mathcal{E}}, \tag{64}$$

where $\mathcal{E} \simeq \mathbb{Z}_4$ is a subgroup of $U(1) \times Z_F$ generated by $\left(\frac{1}{4}, 1\right)$ where we have represented $U(1)$ as $\mathbb{R}/\mathbb{Z}$ and hence $\frac{1}{4}$ denotes an order four element of $U(1)$, and we have represented $Z_F$ as $\mathbb{Z}/2\mathbb{Z}$ and hence 1 denotes the order two element of $Z_F$.

From this, we can see that the $\mathbb{Z}_2$ subgroup of $\mathcal{E} \simeq \mathbb{Z}_4$ generated by $\left(\frac{1}{2}, 0\right) \in (\mathbb{R}/\mathbb{Z}) \times (\mathbb{Z}/2\mathbb{Z})$ involves only the elements in the center of the gauge group. Thus the 1-form symmetry $\mathcal{O}$ of the abelian gauge theory can be identified with this $\mathbb{Z}_2$ subgroup. According to the proposal of [20], we should identify $\mathcal{O}$ with the 1-form symmetry $\mathcal{O}_{\mathfrak{T}}$ of the 5d SCFT $\mathfrak{T}$, i.e

$$\mathcal{O}_{\mathfrak{T}} = \mathbb{Z}_2. \tag{65}$$

Now projecting $\mathcal{E}$ onto $Z_F$ we obtain the subgroup $\mathcal{Z}$ of $Z_F$ generated by $1 \in \mathbb{Z}/2\mathbb{Z}$, and hence $\mathcal{Z} = Z_F$. Thus, the 0-form flavor symmetry group of the abelian gauge theory is $\mathcal{F} = F/Z_F = SO(3)$. According to the proposal of section 2.2.1, we should identify $\mathcal{F}$ with the 0-form symmetry group $\mathcal{F}_{\mathfrak{T}}$ of the 5d SCFT $\mathfrak{T}$, i.e

$$\mathcal{F}_{\mathfrak{T}} = SO(3). \tag{66}$$

The groups $\mathcal{O}$, $\mathcal{E}$ and $\mathcal{Z}$ sit in a short exact sequence (3), which becomes in our case

$$0 \to \mathbb{Z}_2 \to \mathbb{Z}_4 \to \mathbb{Z}_2 \to 0. \tag{67}$$

The projection $\mathcal{E}'$ of $\mathcal{E}$ onto $U(1)$ is the $\mathbb{Z}_4$ subgroup of $U(1)$ generated by $\frac{1}{4}$. Thus, we satisfy the conditions for first criterion of section 2.1. Following the analysis of section 2.1, we obtain the result that the 1-form and 0-form symmetries $\mathcal{O}$ and $\mathcal{F}$ respectively of the abelian gauge theory sit in a 2-group symmetry whose associated Postnikov class is

$$[\mathcal{P}_3] = \text{Bock}(w_2) = w_3, \tag{68}$$

where $w_2 \in H^2(BSO(3), \mathbb{Z}_2)$ is the characteristic class capturing the obstruction of lifting $SO(3)$ bundles to $SU(2)$ bundles, also known as the second Stiefel-Whitney class. It is well-known that the Bockstein homomorphism associated to (67) when applied to $w_2$ gives rise to $w_3$, which is the third Stiefel-Whitney class. According to the proposal in section 2.2.1, the 1-form and 0-form symmetries $\mathcal{O}_{\mathfrak{T}}$ and $\mathcal{F}_{\mathfrak{T}}$ respectively of the 5d SCFT should also sit in a 2-group symmetry whose associated Postnikov class is again (68).

Let us now study the abelian gauge theory arising on the Coulomb branch of the SQFT $\mathfrak{T}'$ obtained after mass deformation of $\mathfrak{T}$. As we recall from previous subsection, here we have a $\mathfrak{u}(1)_N$ flavor symmetry and two massive BPS particles whose charges under gauge $U(1)$ and $U(1)_N$ are collected in (49). The structure group of this abelian gauge theory is

$$S = \frac{U(1) \times U(1)_N}{\mathcal{E}},$$
$$\mathcal{E} \cong \mathbb{Z}_4 = \left\langle \left(\frac{1}{4}, \frac{1}{2}\right) \right\rangle \subset U(1) \times U(1)_N \cong \mathbb{R}/\mathbb{Z} \times \mathbb{R}/\mathbb{Z}. \tag{69}$$

Here $\mathcal{E}$ can be computed from the Smith normal form of the charge matrix (49)

$$\mathcal{N} = \begin{pmatrix} 0 & -2 \\ -2 & 1 \end{pmatrix}, \tag{70}$$

which is

$$\text{SNF} = \mathcal{M} = \begin{pmatrix} 1 & 0 \\ 0 & 4 \end{pmatrix} = A \cdot \mathcal{N} \cdot B, \qquad A = \begin{pmatrix} -1 & -1 \\ 1 & 2 \end{pmatrix}, \qquad B = \begin{pmatrix} 0 & -1 \\ 1 & 2 \end{pmatrix} \tag{71}$$

Then

$$\mathbb{Z}^2/\mathcal{M}\mathbb{Z}^2 = \left\langle \begin{pmatrix} 0 \\ 1 \end{pmatrix} a, \ a \in \mathbb{Z}_4 \quad \left(\text{mod } \mathcal{M}\mathbb{Z}^2\right) \right\rangle, \tag{72}$$

and using the change of basis with integral 1-1 maps $A$ and $B$

$$\mathbb{Z}^2/\mathcal{N}\mathbb{Z}^2 = \left\langle A^{-1} \begin{pmatrix} 0 \\ 1 \end{pmatrix} a, \ a \in \mathbb{Z}_4 \quad \left(\text{mod } \mathcal{N}\mathbb{Z}^2\right) \right\rangle. \tag{73}$$

A representative of this is $(1,2)$, which confirms the realization of $\mathcal{E}$ in (69).

The subgroup $\mathcal{O}$ generated by elements of the form $(*,0) \in \mathcal{E}$ equals the 1-form symmetry of the theory and is

$$\mathcal{O}_{\mathfrak{T}} = \mathcal{O} = \left\langle \left( \frac{1}{2}, 0 \right) \right\rangle \cong \mathbb{Z}_2 \subset U(1) \times U(1)_N. \tag{74}$$

The image under the projection map $\mathcal{E} \to Z_F$ is

$$\mathcal{Z} \cong \mathbb{Z}_2 = \left\langle \frac{1}{2} \right\rangle \subset U(1)_N \cong \mathbb{R}/\mathbb{Z}. \tag{75}$$

Thus, the abelian gauge theory has 0-form symmetry group

$$\mathcal{F} = \frac{U(1)_N}{\mathbb{Z}_2}. \tag{76}$$

The short exact sequence (13) $0 \to \mathcal{O} \to \mathcal{E} \to \mathcal{Z} \to 0$ becomes

$$0 \to \mathbb{Z}_2 \to \mathbb{Z}_4 \to \mathbb{Z}_2 \to 0. \tag{77}$$

The projection of $\mathcal{E}$ onto $U(1)$ is $\mathcal{E}' \simeq \mathbb{Z}_4$ generated by $\frac{1}{4} \in U(1) \simeq \mathbb{R}/\mathbb{Z}$, implying that the projection map is injective. Thus, following the arguments of section 2.1, we find a potential 2-group symmetry in the abelian gauge theory, formed by 0-form symmetry group $\mathcal{F} = U(1)_N/\mathbb{Z}_2$ and 1-form symmetry group $\mathcal{O} = \mathbb{Z}_2$, with the Postnikov class being

$$[\mathcal{P}_3] = \text{Bock}(\widehat{v}_2) \in H^3(B\mathcal{F}, \mathbb{Z}_2), \tag{78}$$

where $\widehat{v}_2 \in H^2(B\mathcal{F}, \mathbb{Z}_2)$ is the characteristic class capturing obstruction of lifting $\mathcal{F} = U(1)_N/\mathbb{Z}_2$ bundles to $F = U(1)_N$ bundles, and Bock is the Bockstein corresponding to the above short exact sequence (77). However, the above Postnikov class $[\mathcal{P}_3]$ is trivial due to the following argument. We can identify $\widehat{v}_2$ as $c_1 \pmod 2$ where $c_1$ is the first Chern class for $\mathcal{F} = U(1)_N/\mathbb{Z}_2 \simeq U(1)$ bundles. Moreover, we can recognize $\widehat{v}_2$ as the image of the element $c_1 \pmod 4 \in H^2(B\mathcal{F}, \mathbb{Z}_4)$ under the map $H^2(B\mathcal{F}, \mathbb{Z}_4) \to H^2(B\mathcal{F}, \mathbb{Z}_2)$ induced by the projection map $\mathbb{Z}_4 \to \mathbb{Z}_2$ in (77). Since this map is the first map and Bockstein is the second map in the following part of the long exact sequence in cohomology

$$\cdots \to H^2(B\mathcal{F}, \mathbb{Z}_4) \to H^2(B\mathcal{F}, \mathbb{Z}_2) \to H^3(B\mathcal{F}, \mathbb{Z}_2) \to \cdots \tag{79}$$

we find that $[\mathcal{P}_3] = 0$. Thus, there is no 2-group symmetry in this abelian gauge theory. Using the proposal of section 2.2.2, we are lead to the conclusion that the SQFT $\mathfrak{T}'$ does not admit a 2-group symmetry.

This conclusion can be much more easily reached from the point of the view of the special point on the CB of $\mathfrak{T}'$ where the low-energy theory is the $SU(2)_0$ non-abelian gauge theory. In

this case the flavor symmetry is $U(1)_I$ associated to the surface $N'$ and the structure group is simply

$$S = SU(2)_0 \times U(1)_I \,, \tag{80}$$

with no possibility of a 2-group structure. The 0-form symmetry group is

$$\mathcal{F} = U(1)_I \,, \tag{81}$$

which by comparing with (76) leads to the conclusion that $U(1)_I = U(1)_N/\mathbb{Z}_2$. This is related to the factor of $\frac{1}{2}$ relating $N$ and $N'$ in (62).

In summary: we provided evidence, that the $SU(2)_0$ SCFT $\mathfrak{T}$ has a 2-group symmetry (when the inverse gauge coupling is 0) which can be easily observed when a non-conformal vacuum lying on the CB of vacua is chosen. However, on the non-abelian gauge theory locus, i.e. in the 5d SQFT $\mathfrak{T}'$ (obtained after turning on inverse gauge coupling) there is *no* 2-group symmetry, which can be observed from both kinds of CB vacua of $\mathfrak{T}'$ – having respectively non-abelian gauge theory and abelian gauge theory at low-energies.

### 3.3 An Equivalent Perspective in Terms of Chern-Simons Couplings

A similar perspective is given by the study of 5d Chern-Simons couplings, and the condition that it must be integral on $M_5$ [46, 49]. This leads to a rather strong necessary condition on the possible allowed background. Let us focus on the $SU(2)_0$ example, where the particle and charges are given in (49). We can now compute the prepotential [46, 50], which reads

$$\mathcal{P} = \frac{1}{6} \left( 8\phi_g^3 - 6\phi_g \phi_N^2 \right) \,, \tag{82}$$

where $\phi^g$ and $\phi_N$ are the Coulomb branch scalars of $U(1)_g$ and $U(1)_N$. We are interested in the Chern-Simons terms in the effective Lagrangian, where we have that $CS_5 = \int_{Y_6} \mathcal{L}_6$, where $\partial Y_6 = M_5$,

$$\begin{aligned}
\mathcal{L}_6 &= \frac{\partial^3 \mathcal{P}}{\partial \phi_g^3} F_g \wedge F_g \wedge F_g + \frac{\partial^3 \mathcal{P}}{\partial \phi_g \partial \phi_N^2} F_g \wedge F_N \wedge F_N + \frac{\partial^3 \mathcal{P}}{\partial \phi_g^2 \partial \phi_N} F_g \wedge F_g \wedge F_N \\
&= 8 F_g \wedge F_g \wedge F_g - 2 F_g \wedge F_N \wedge F_N \,.
\end{aligned} \tag{83}$$

For consistency of the theory this quantity must be integral on a $Y_6$ [46, 49]. As we have seen in the previous sections and in appendix B the structure group in the coulomb branch may allow fractional periods of $F_g$ and $F_N$. The most general choice of background which is consistent with integrality of (83) is given by,

$$F_g = \frac{v_2^g}{4} \,(\text{mod } \mathbb{Z})\,, \qquad F_N = \frac{\tilde{v}_2}{4} \,(\text{mod } \mathbb{Z})\,, \qquad \tilde{v}_2 = 2v_2^g \,, \tag{84}$$

where $\tilde{v}_2$ and $v_2^g$ are the obstruction to the lifting $U(1)_N/\mathbb{Z}_2$ and $U(1)_g/\mathbb{Z}_4$ to $U(1)_N$ and $U(1)_g$ respectively. This choice of background is consistent with the structure group highlighted in the previous section

$$S = \frac{U(1)_g \times U(1)_N}{\mathbb{Z}_4} \,. \tag{85}$$

## 3.4 The 2-Group from Five-brane Webs

It is possible to describe the 2-group from the IIB $(p,q)$ 5-brane web realization of the $SU(2)_0$ theory

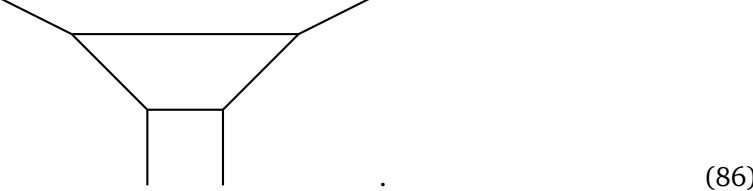

$$\tag{86}$$

First of all, the 1-form symmetry is encoded in the $(p,q)$ 5-branes charges at infinity (of the infinitely extended 5-branes). The Smith Normal Form (SNF) of the matrix of $(p,q)$ charges [21]

$$M = \begin{pmatrix} q_1 & p_1 \\ q_2 & p_2 \\ \vdots & \vdots \\ q_N & p_N \end{pmatrix} \tag{87}$$

satisfies $M = A_1^T(\text{SNF}(M))A_2$, where $A_1$ is a $N \times N$ matrix and $A_2$ is a $2 \times 2$ matrix. $M$ and $A_2$ act on the IIB 2-form field $SL(2,\mathbb{Z})$ doublet $(B_2, C_2)$. For the $SU(2)_0$ web with manifest $\mathfrak{g} = \mathfrak{su}(2)$ flavor symmetry algebra we have

$$M = \begin{pmatrix} 0 & -1 \\ 0 & -1 \\ 2 & 1 \\ -2 & 1 \end{pmatrix}, \quad \text{SNF}(M) = \begin{pmatrix} 1 & 0 \\ 0 & 2 \\ 0 & 0 \\ 0 & 0 \end{pmatrix}, \quad A_2 = \begin{pmatrix} 1 & -1 \\ 0 & 1 \end{pmatrix}. \tag{88}$$

Applying $A_2$ on the doublet $(B_2, C_2)$ we deduce that the IIB field responsible for the 1-form symmetry background is $C_2$, and in particular that $2dC_2 = 2F_3 = 0$. This forces the condition $[F_3] \in H^3(M_5, \mathbb{Z}_2)$.

Having activated the 1-form symmetry background identified with $[F_3] \in H^3(M_5, \mathbb{Z}_2)$, we can ask now what happens on the world-volume of the two parallel NS5-branes responsible for the $\mathfrak{g} = \mathfrak{su}(2)$ flavor symmetry algebra. For convenience, we work with the S-dual version of this web (which consist of a rotation of 90 degrees on the web plane (86)). The flavor symmetry algebra is now given by parallel D5s and the 1-form symmetry is $[H] = [dB_2] \in H^3(M_5, \mathbb{Z}_2)$. It occurs that the 1-form symmetry background has a non-trivial effect on the world-volume of the D5-branes, [51]. Ignoring the normal bundle contribution, it was shown in [51] that in order to avoid global worldsheet anomalies on the strings ending on the D-branes, the following condition must hold,

$$[H] = \text{Bock}(w_2), \tag{89}$$

where $w_2$ is the obstruction of lifting $SU(2)/\mathbb{Z}_2$ bundles to a $U(2)$ bundle and $[H] \in H^3(M_5, \mathbb{Z})$. Moreover Bock is the Bockstein homomorphism

$$H^2(M_5, \mathbb{Z}_2) \to H^3(M_5, \mathbb{Z}), \tag{90}$$

associated to the short exact sequence

$$0 \to \mathbb{Z} \to \mathbb{Z} \to \mathbb{Z}_2 \to 0. \tag{91}$$

The IIB brane-web requires $[H] \in H^3(M_5, \mathbb{Z}_2)$, which implies that there exist a non-trivial reduction of (91) to (67). This leads to the identification of $w_2$ with the obstruction of lifting $SO(3)$ bundles to $SU(2)$ and $\text{Bock}(w_2)$ in (89) with the one in (68).

A similar argument can be applied to the IIB web engineering of $SU(3)_3$ gauge theory [52] with $\mathcal{O} = \mathbb{Z}_3$, and its superconformal UV completion with $\mathfrak{g} = \mathfrak{su}(2)$ flavor symmetry algebra. However, the reduction of the sequence (91), leads to $0 \to \mathbb{Z}_3 \to \mathbb{Z}_6 \to \mathbb{Z}_2 \to 0$, which has trivial Bock($w_2$), since $\mathbb{Z}_6 \simeq \mathbb{Z}_2 \times \mathbb{Z}_3$. In conclusion, the 1-form symmetry background in the IIB webs corresponds to non-trivial torsional configuration for NSNS or RR 3-form flux on the 5d space-time where the theory lives. This implies that the fluxes are automatically pulled back on the brane stack realising the flavor symmetry in the brane configuration. A 2-group is therefore induced due to worldsheet anomaly cancellation for the strings ending on the branes.

## 3.5 Circle Compactification to 4d $\mathcal{N} = 2$

To provide further evidence for the 2-group structure, we consider the 4d $\mathcal{N} = 2$ theory obtained by compactification of the $SU(2)_0$ SCFT on a finite radius $S^1$. As shown in [44], this gives rise to a rank 1 theory with the following Seiberg-Witten geometry: the elliptic fibration has two $I_1$ fibers and one $I_2$ fiber. At the $I_2$ point the IR theory is a $U(1)_g$ gauge theory with two massless monopoles of charge $(q_m, q_e) = (1, 0)$. In addition there is a massive dyon of charge $(1, -2)$. Combining the two, we obtain two massive excitations carrying purely electric charge $(0, -2)$. We identify the $\mathfrak{su}(2)$ flavor symmetry rotating the electric excitations (and hence also the massless monopoles) as an avatar of the $\mathfrak{su}(2)$ 0-form symmetry of the 5d SCFT $\mathfrak{T}$. Moreover, since the electric charges are even integers, the theory admits a $\mathbb{Z}_2$ electric 1-form symmetry descending from the center of the $U(1)_g$ gauge group, which we identify as the avatar of $\mathbb{Z}_2$ 1-form symmetry of the 5d parent. The full structure group is

$$S = \frac{U(1)_g \times SU(2)_F}{\mathbb{Z}_4} \, . \tag{92}$$

Again we can apply the same arguments as in its 5d parent: the 0-form symmetry is $SO(3)_F = SU(2)_F / \mathbb{Z}_2$, which forms a non-trivial 2-group with the 1-form symmetry $\mathbb{Z}_2$, with the Postnikov class being given by (68) [24]. This 2-group structure is the same as the proposed 2-group structure of the 5d parent. We take this as evidence in favor of the presence of 2-group symmetry in the 5d parent theory, namely the $SU(2)_0$ SCFT.

## 3.6 Comments on 0-/1-form Symmetry Anomaly

The $\mathcal{N} = 1$ $SU(2)_0$ gauge theory in 5d has a mixed 't Hooft anomaly between the $U(1)_I$ instantonic 0-form symmetry and the 1-form symmetry $\mathcal{O} = \mathbb{Z}_2$ [18]. The anomaly takes the form (see appendix B)

$$\mathcal{A}_{IR} = \exp\left( \frac{2\pi i}{4} \int_{M_6} c_1(I) \cup \mathfrak{P}(B) \right), \tag{93}$$

where $c_1(I)$ is the first Chern class for the background $U(1)_I$ bundle, $B$ is the background field for the 1-form symmetry[25] and $\mathfrak{P}(B)$ denotes the Pontryagin square[26] of $B$.

The question we would like to address is how this anomaly is realized in the conformal theory obtained after turning off the mass deformation. We can consider the following three options:

(1) The anomaly (93) lifts to a 2-group anomaly, i.e. an anomaly for the full 2-group symmetry.

---

[24]The magnetic 1-form symmetry is completely broken by the massless monopoles and hence we do not worry about its contribution to the 2-group structure.

[25]We work with integer valued cochains. More precisely, the background for 1-form symmetry is $B$ (mod 2).

[26]This is defined as $\mathfrak{P}(B) = B \cup B - \delta B \cup_1 B$ by using cup and higher-cup products.

(2) The effective field theory after mass deformation is not only the $SU(2)_0$ gauge theory, but also contains an additional topological sector that carries an anomaly that is inverse of (93). The total anomaly is trivial, and then the conformal theory would also have no anomaly.

(3) A combination of the above two scenarios: a topological sector modifies the anomaly (93), and the modified anomaly lifts to a 2-group anomaly.

Recent work [45] seems to indicate that option 1 is the most likely option. We leave the determination of the precise fate of the anomaly (93) in the conformal theory to future work.

For now let us assume that there are no additional topological sectors and that the anomaly (93) lifts to a 2-group anomaly $\mathcal{A}_{UV}$. Then the anomaly needs to be matched along the Higgs branch of vacua of the conformal theory. We do not know the precise form of $\mathcal{A}_{UV}$, but we should at least be able to match (93). Let us discuss the matching of (93) on the Higgs branch in the rest of this subsection.

First of all, the 1-form symmetry is spontaneously broken on the Higgs branch. We take this to indicate, in a fashion similar to in $4d$ [1], that the low energy effective theory on the Higgs branch contains a 5d $\mathbb{Z}_2$ gauge theory whose kinetic term takes the form

$$\exp\left(\frac{2\pi i}{2}\int_{M_5} b_1 \delta c_3\right), \tag{94}$$

where $b_1$ and $c_3$ are dynamical $\mathbb{Z}_2$ 1-form and 3-form gauge fields respectively. The HB is $\mathbb{C}^2/\mathbb{Z}_2$ whose $SO(3)$ isometry is identified with the $SO(3)_F$ 0-form symmetry group of the conformal theory. So the 0-form symmetry is also completely spontaneously broken, and the low-energy theory contains a sigma model on $\mathbb{C}^2/\mathbb{Z}_2$. The $U(1)_I$ symmetry is non-linearly realized. We now propose that the anomaly is realized by having a coupling between the above two sectors: namely the $\mathbb{Z}_2$ gauge theory and the sigma model. The coupling takes the form

$$\exp\left(\frac{2\pi i}{4}\int_{M_5} c_1(I) \cup b_1 \cup \delta b_1\right). \tag{95}$$

The 1-form symmetry background makes $b_1$ non-closed

$$\delta b_1 = B. \tag{96}$$

The non-closedness of $b_1$ makes (95) depend on the integer lift $b_1$ of the $\mathbb{Z}_2$ gauge field. As shown in [37], this can be cured by adding the following term to the action

$$\exp\left(\frac{-2\pi i}{4}\int_{M_5} c_1(I) \cup \left(b_1 \cup B + B \cup b_1\right)\right). \tag{97}$$

Now, for the total action to be well-defined, the coboundary of the total action should not depend on the dynamical field $b_1$. We can compute the coboundary to be

$$\exp\left(\frac{2\pi i}{4}\int_{M_6} c_1(I) \cup \left(\delta\left(\delta B \cup_1 b_1\right) + \mathfrak{P}(B)\right)\right), \tag{98}$$

whose dependence on $b_1$ can be canceled by adding an additional term to the action of the form

$$\exp\left(\frac{-2\pi i}{4}\int_{M_5} c_1(I) \cup \delta B_e \cup_1 b_1\right). \tag{99}$$

After adding this additional term, we obtain a well-defined action, and the remaining terms in (98) give rise to an anomaly of the form

$$\exp\left(\frac{2\pi i}{4}\int_{M_6} c_1(I) \cup \mathfrak{P}(B)\right), \tag{100}$$

which matches (93).

## 3.7 Potential Mixed Anomaly of the $SO(3)_0$ SCFT

As we have discussed in section 2.3, if a 5d theory $\mathfrak{T}$ has a 2-group symmetry which is non-anomalous, then the theory $\widetilde{\mathfrak{T}}$ obtained by gauging the 1-form symmetry of $\mathfrak{T}$ has a dual 2-form symmetry which has a mixed anomaly with the 0-form symmetry of $\widetilde{\mathfrak{T}}$ (which is the same as the 0-form symmetry of $\mathfrak{T}$ participating in the 2-group).

In the previous subsection, we discussed an anomaly (93) which can potentially lift to an anomaly of the 2-group symmetry of the $SU(2)_0$ 5d SCFT $\mathfrak{T}$. However, there is also a scenario in which this anomaly is canceled by a topological sector, in which case the conformal theory has no anomaly. If this scenario is realized, then the 2-group symmetry of the $SU(2)_0$ 5d SCFT would be non-anomalous, and one can apply the analysis of section 2.3 to conclude that the $SO(3)_0$ 5d SCFT $\widetilde{\mathfrak{T}}$ obtained by gauging the $\mathcal{O}_{\mathfrak{T}} = \mathbb{Z}_2$ 1-form symmetry would have a 0-form flavor symmetry group

$$\mathcal{F}_{\widetilde{\mathfrak{T}}} = SO(3) \tag{101}$$

and a 2-form symmetry group

$$\mathcal{T}_{\widetilde{\mathfrak{T}}} = \mathbb{Z}_2 \simeq \mathcal{O}_{\mathfrak{T}}, \tag{102}$$

with a mixed anomaly

$$\mathcal{A}_{0-2} = \exp\left(\frac{2\pi i}{2}\int_{M_6} B_3 \cup w_3\right), \tag{103}$$

where $B_3$ is the background field for $\mathbb{Z}_2$ 2-form symmetry $w_3$ is the third Stiefel-Whitney class of background $SO(3)$ bundle. We emphasize again that these results about the $SO(3)_0$ SCFT can only be trusted if the 2-group symmetry of the $SU(2)_0$ SCFT is non-anomalous (which would be the case if the scenario proposed in the previous subsection is actually realized in the mass deformed $SU(2)_0$ SCFT).

## 3.8 Superconformal Index for $SU(2)_0$ and $SO(3)_0$

In this ongoing section we have studied the global symmetry of both the $SU(2)_0$ and $SO(3)_0$ theories. For $SU(2)_0$ theory we can compute with strong confidence that the 0-form symmetry group is $SO(3)$, while for $SO(3)_0$ theory we need to rely on some assumptions regarding the anomalies to argue that the 0-form symmetry group should be $SO(3)$. One way to test these predictions is to compute the super-conformal index of these theories.

The superconformal index of $G = SU(2)_0$ theory is [12]

$$\mathcal{I} = 1 + \chi_3(q)t^2 + (1 + \chi_3(q))\chi_2(u)t^3 + O(t^4), \tag{104}$$

where $q$, $t$, $u$ are the fugacities for $\mathfrak{u}(1)_I$, the Cartan of $\mathfrak{su}(2)_D \subset \mathfrak{su}(2)_+ \oplus \mathfrak{su}(2)_R$, and the Cartan of $\mathfrak{su}(2)_-$. Here, $\mathfrak{so}(4) = \mathfrak{su}(2)_+ \oplus \mathfrak{su}(2)_-$ is the rotation symmetry of the spatial part of Euclidean spacetime and $\mathfrak{su}(2)_R$ is the R-symmetry of the 5d $\mathcal{N} = 1$ SCFT. $\chi_\mathbf{n}$ denotes the character for $n$-dimensional irreducible representation of $\mathfrak{su}(2)_F$ which is the enhancement of $\mathfrak{u}(1)_I$. Notice that only those representations of $\mathfrak{su}(2)_F$ appear that have trivial charge under the center of the simply connected group $SU(2)_F$ associated to $\mathfrak{su}(2)_F$. This remains

to be true at even higher orders of $t$ as can be seen from the result displayed [12]. Thus, the superconformal index is consistent with our prediction that the $SU(2)_0$ SCFT has $SO(3)_F$ flavor symmetry group. This argument was also used by [18] to argue for $SO(3)_F$ flavor symmetry of the $SU(2)_0$ theory.

We indirectly argue that the superconformal index of $G = SO(3)_0$ theory should be the same as that of $G = SU(2)_0$ theory due to the equivalence of the instanton partition functions of two different theories [53]. The 5d superconformal index [12] is a partition function of a SCFT on $S^4 \times S^1$ and it is computed by localization technique, which is applied to the IR gauge theory of the SCFT. In short, the ingredient of the index computation is the instanton partition function and perturbative 1-loop determinants. The perturbative 1-loop determinants of the two gauge theories match easily, since they are determined by Lie algebra, which is shared by the two gauge theories. Instanton partition function involves an extra subtlety[27] and is not fixed by a single data. The author of [53] argues that the instanton partition functions for the two gauge groups should match. Therefore, we conclude that the resulting superconformal index for $G = SU(2)_0$ and $G = SO(3)_0$ theories should match. With this conclusion and (104), we conclude that the flavor symmetry group of $G = SO(3)_0$ theory should again be $SO(3)_F$.

### 3.9 Global Flavor Symmetry of Rank 1 Seiberg Theories

In this subsection we use our method to determine the global form of 0-form flavor symmetry groups of Seiberg theories, which are 5d SCFTs that admit a mass deformation after which the theories can be described in the IR in terms of 5d $\mathcal{N} = 1$ non-abelian gauge theories with gauge group $SU(2)$ and $n$ full hypers in fundamental representation for $0 \leq n \leq 7$. The case $n = 0$ was discussed in previous subsections. We discuss other cases below.

In every case $n \geq 2$, we find that the flavor symmetry group is centerless, which matches with the expectation from the superconformal indices for these theories [12]. We also discuss how our results match with the expectations from the ray indices for these theories [19].

#### 3.9.1 Geometric Computation of Flavor Symmetry Group

$SU(2) + \boldsymbol{F}$ : For this case the non-abelian part of flavor symmetry of the 5d SCFT $\mathfrak{T}$ is $\mathfrak{f}_{\mathfrak{T}} = \mathfrak{su}(2)$. Thus, $Z_F \simeq \mathbb{Z}/2\mathbb{Z}$. The corresponding geometry is

$$\boldsymbol{N} \xrightarrow{\quad f \qquad e \quad} \boldsymbol{1}_2^1 \quad . \tag{105}$$

Here $\boldsymbol{1}_n^k$ indicates a Hirzebruch $\mathbb{F}_n$ blown up $k$ times. $\boldsymbol{N}$ describes the $\mathfrak{su}(2)$ flavor symmetry of $\mathfrak{T}$. We find that $\mathcal{E} = 0$ which implies that the non-abelian part of the 0-form flavor symmetry group $\mathcal{F}_{\mathfrak{T}}$ of $\mathfrak{T}$ is

$$\mathcal{F}_{\mathfrak{T}} = SU(2). \tag{106}$$

---

[27]One outstanding difference is the following. In $G = SU(2)_0 = Sp(1)_0$ theory, the dual gauge group on one instanton is $\hat{G} = O(1) = \mathbb{Z}_2$, which is rank 0. When we compute the instanton partition function, we integrate over Haar measure of $\hat{G}$, so there is no integral to perform in this case. However, for $G = SO(3)_0$ theory, the dual gauge group on one instanton is $\hat{G} = Sp(1)$, so we need to perform an integral to get the instanton partition function. Moreover, even though two instanton moduli spaces are the same, the bundles on the moduli spaces are not the same.

$\underline{SU(2)+2F}$ **:** The corresponding geometry is

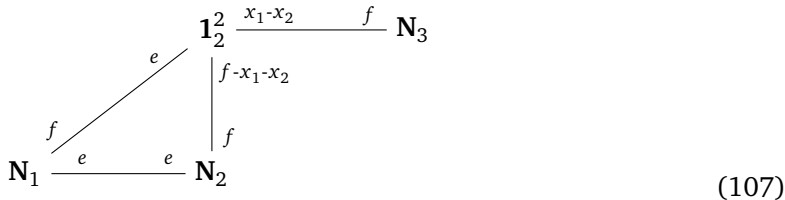

$$\text{(107)}$$

$\mathbf{N_1}$ and $\mathbf{N_2}$ describe an $\mathfrak{su}(3)$ flavor symmetry of $\mathfrak{T}$, and $\mathbf{N_3}$ describes an $\mathfrak{su}(2)$ flavor symmetry of $\mathfrak{T}$. Thus, $Z_F \simeq \mathbb{Z}/3\mathbb{Z} \times \mathbb{Z}/2\mathbb{Z}$. We find that $\mathcal{E} \simeq \mathbb{Z}_2 \times \mathbb{Z}_3$ and it is generated by $\left(\frac{1}{2}, 0, 1\right) \in Z_G \times Z_F \simeq \mathbb{R}/\mathbb{Z} \times \mathbb{Z}/3\mathbb{Z} \times \mathbb{Z}/2\mathbb{Z}$ and $\left(\frac{1}{3}, 2, 0\right) \in Z_G \times Z_F$. From this, we find that $\mathcal{Z} \simeq \mathbb{Z}_2 \times \mathbb{Z}_3$ generated by $(0, 1) \in Z_F \simeq \mathbb{Z}/3\mathbb{Z} \times \mathbb{Z}/2\mathbb{Z}$ and $(2, 0) \in Z_F$. Thus, the 0-form flavor symmetry group $\mathcal{F}_{\mathfrak{T}}$ of $\mathfrak{T}$ is

$$\mathcal{F}_{\mathfrak{T}} = PSU(3) \times SO(3). \tag{108}$$

$\underline{SU(2)+3F}$ **:** The corresponding geometry is

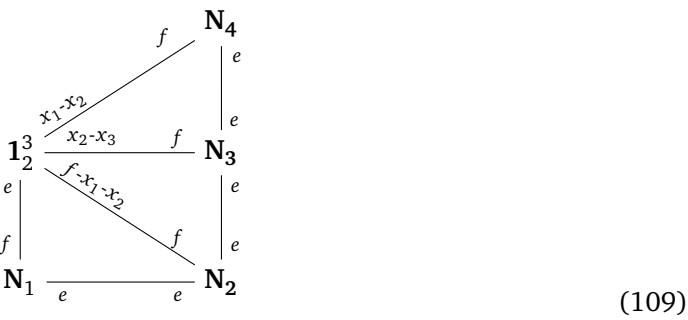

$$\text{(109)}$$

$\mathbf{N}_i$ describe an $\mathfrak{su}(5)$ flavor symmetry of $\mathfrak{T}$. Thus, $Z_F = \mathbb{Z}/5\mathbb{Z}$. We find that $\mathcal{E} \simeq \mathbb{Z}_5$ generated by $\left(\frac{3}{5}, 1\right) \in Z_G \times Z_F \simeq \mathbb{R}/\mathbb{Z} \times \mathbb{Z}/5\mathbb{Z}$. From this, we find that $\mathcal{Z} \simeq \mathbb{Z}_5$ generated by $1 \in Z_F$. Thus, the 0-form flavor symmetry group $\mathcal{F}_{\mathfrak{T}}$ of $\mathfrak{T}$ is

$$\mathcal{F}_{\mathfrak{T}} = SU(5)/\mathbb{Z}_5 = PSU(5). \tag{110}$$

$\underline{SU(2)+4F}$ **:** The corresponding geometry is

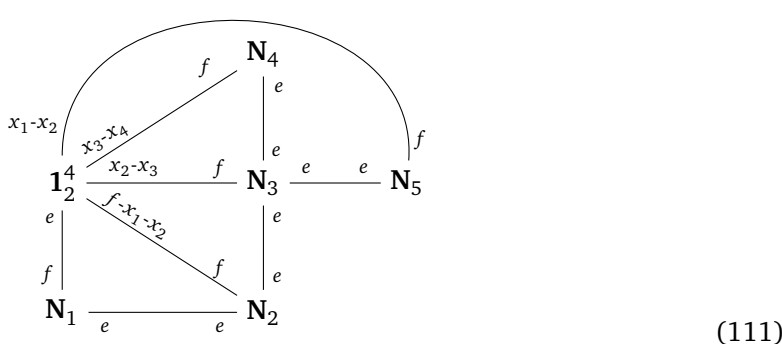

$$\text{(111)}$$

$\mathbf{N}_i$ describe an $\mathfrak{so}(10)$ flavor symmetry of $\mathfrak{T}$. Thus, $Z_F = \mathbb{Z}/4\mathbb{Z}$. We find that $\mathcal{E} \simeq \mathbb{Z}_4$ generated by $\left(\frac{3}{4}, 1\right) \in Z_G \times Z_F \simeq \mathbb{R}/\mathbb{Z} \times \mathbb{Z}/4\mathbb{Z}$. From this, we find that $\mathcal{Z} \simeq \mathbb{Z}_4$ generated by $1 \in Z_F$. Thus, the 0-form flavor symmetry group $\mathcal{F}_{\mathfrak{T}}$ of $\mathfrak{T}$ is

$$\mathcal{F}_{\mathfrak{T}} = \text{Spin}(10)/\mathbb{Z}_4 = SO(10)/\mathbb{Z}_2. \tag{112}$$

$\underline{SU(2) + 5\mathbf{F}}$ :   The corresponding geometry is

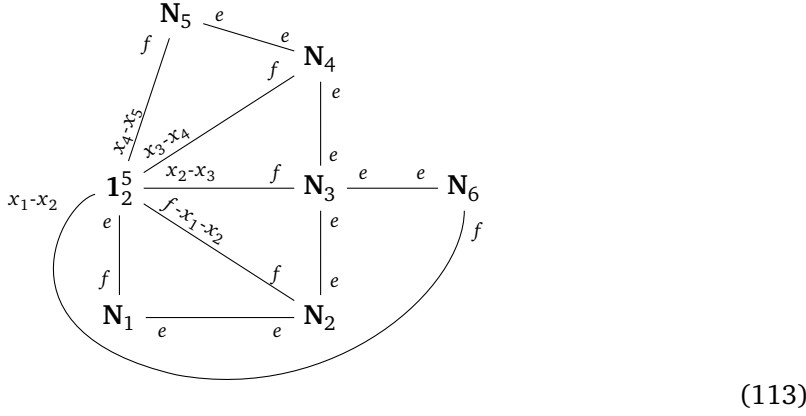

$$(113)$$

$\mathbf{N}_i$ describe an $\mathfrak{e}_6$ flavor symmetry of $\mathfrak{T}$. Thus, $Z_F = \mathbb{Z}/3\mathbb{Z}$. We find that $\mathcal{E} \simeq \mathbb{Z}_3$ generated by $\left(\frac{2}{3}, 1\right) \in Z_G \times Z_F \simeq \mathbb{R}/\mathbb{Z} \times \mathbb{Z}/3\mathbb{Z}$. From this, we find that $\mathcal{Z} \simeq \mathbb{Z}_3$ generated by $1 \in Z_F$. Thus, the 0-form flavor symmetry group $\mathcal{F}_{\mathfrak{T}}$ of $\mathfrak{T}$ is

$$\mathcal{F}_{\mathfrak{T}} = E_6/\mathbb{Z}_3 \,. \tag{114}$$

$\underline{SU(2) + 6\mathbf{F}}$ :   The corresponding geometry is

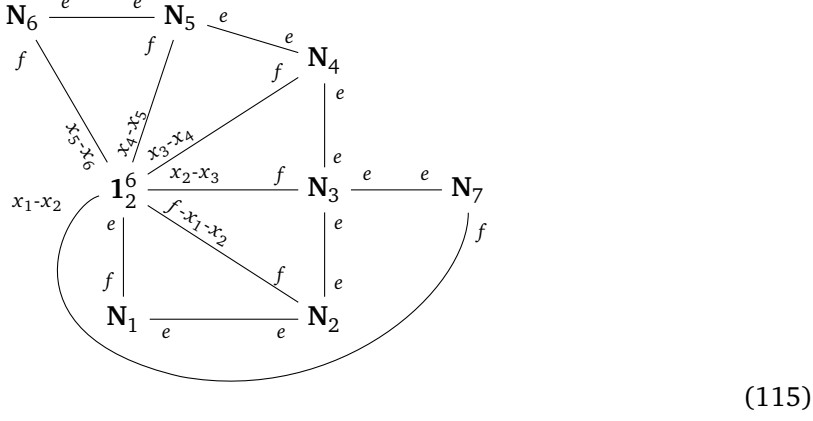

$$(115)$$

$\mathbf{N}_i$ describe an $\mathfrak{e}_7$ flavor symmetry of $\mathfrak{T}$. Thus, $Z_F = \mathbb{Z}/2\mathbb{Z}$. We find that $\mathcal{E} \simeq \mathbb{Z}_2$ generated by $\left(\frac{1}{2}, 1\right) \in Z_G \times Z_F \simeq \mathbb{R}/\mathbb{Z} \times \mathbb{Z}/2\mathbb{Z}$. From this, we find that $\mathcal{Z} \simeq \mathbb{Z}_2$ generated by $1 \in Z_F$. Thus, the 0-form flavor symmetry group $\mathcal{F}_{\mathfrak{T}}$ of $\mathfrak{T}$ is

$$\mathcal{F}_{\mathfrak{T}} = E_7/\mathbb{Z}_2 \,. \tag{116}$$

$\underline{SU(2) + 7\mathbf{F}}$ :   The flavor symmetry for this theory is $\mathfrak{f}_{\mathfrak{T}} = \mathfrak{e}_8$ which has a trivial center. Thus, the 0-form flavor symmetry group $\mathcal{F}_{\mathfrak{T}}$ of $\mathfrak{T}$ is

$$\mathcal{F}_{\mathfrak{T}} = E_8 \,. \tag{117}$$

### 3.9.2   Relationship to the Ray Index

The work of [19] provides non-trivial confirmations for our above proposals. Let us discuss it in the context of the $E_7$ Seiberg theory. The other cases can also be discussed in a similar fashion.

For the $E_7$ Seiberg theory, [19] argue that there is a line operator $\mathcal{L}$ at the conformal point such that after the mass deformation (that turns on inverse gauge coupling for $SU(2)$) $\mathcal{L}$ reduces to the Wilson line defect transforming in the fundamental representation of the $SU(2)$ gauge group. They compute a ray index associated to $\mathcal{L}$ which counts the non-genuine local operators[28] living at the end of $\mathcal{L}$, and find that these non-genuine local operators carry a non-trivial charge under the $\mathbb{Z}_2$ center of $E_7$. After the mass deformation, the non-genuine local operators living at the end of $\mathcal{L}$ become non-genuine local operators living at the end of fundamental Wilson line, and whose charges are

$$(1,1,1,0)+(0,\alpha,0,\alpha)\in\widehat{Z}_G\times\widehat{Z}_{F,IR}\simeq\mathbb{Z}/2\mathbb{Z}\times\mathbb{Z}/2\mathbb{Z}\times\mathbb{Z}/2\mathbb{Z}\times\mathbb{Z}\,,\qquad(118)$$

where $\alpha\in\mathbb{Z}$. The first element denotes the charge under $Z_G\simeq Z_2$ center of $SU(2)$ gauge group, the second and third elements denote charge under $\mathbb{Z}_2\times\mathbb{Z}_2$ center of the simply connected group Spin(12) associated to the $\mathfrak{so}(12)$ flavor symmetry algebra rotating the 6 fundamental hypers, and the fourth element denotes the charge under $U(1)_I$ 0-form symmetry associated to the instanton current[29]. The non-genuine-ness of the local operators (118) reflects in the fact that they carry non-trivial charges under $Z_G$ and hence must arise at the end of a Wilson line charged non-trivially under $Z_G$, so that the whole configuration is gauge invariant.

Now, let us study the matter content predicted by the geometry that carries non-trivial charge under $Z_G$. The geometry displaying only the $\mathfrak{so}(12)$ part of the flavor symmetry is

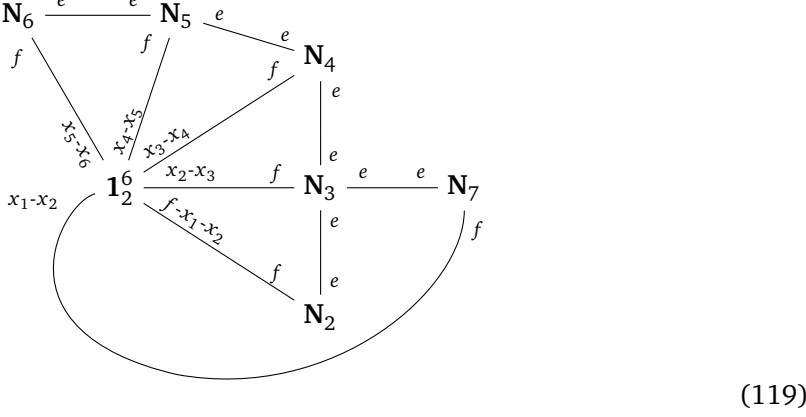

$$(119)$$

From this we see that a blowup $x_i$ has charge 1 (mod 2) under the $\mathbb{Z}_2$ center of $SU(2)$, charge (1 (mod 2), 1 (mod 2)) under the $\mathbb{Z}_2\times\mathbb{Z}_2$ center of Spin(12) and charge 0 under $U(1)_I$ since it is not an instanton. In other words, it carries charge $(1,1,1,0)\in\widehat{Z}_G\times\widehat{Z}_{F,IR}\simeq\mathbb{Z}/2\mathbb{Z}\times\mathbb{Z}/2\mathbb{Z}\times\mathbb{Z}/2\mathbb{Z}\times\mathbb{Z}$. On the other hand, the curve $f$ of the compact surface $\mathbf{S}_1$ carries charge $(0,0,0,0)\in\widehat{Z}_G\times\widehat{Z}_{F,IR}\simeq\mathbb{Z}/2\mathbb{Z}\times\mathbb{Z}/2\mathbb{Z}\times\mathbb{Z}/2\mathbb{Z}\times\mathbb{Z}$, and the curve $e$ of the compact surface $\mathbf{S}_1$ carries charge $(0,1,0,1)\in\widehat{Z}_G\times\widehat{Z}_{F,IR}\simeq\mathbb{Z}/2\mathbb{Z}\times\mathbb{Z}/2\mathbb{Z}\times\mathbb{Z}/2\mathbb{Z}\times\mathbb{Z}$. Thus, an arbitrary curve having a non-trivial charge under $Z_G$ takes the form

$$C=\alpha e+\beta f+\sum_i\gamma_i x_i\,,\qquad(120)$$

with $\alpha,\beta,\gamma_i\in\mathbb{Z}$ such that $\sum_i\gamma_i=2n+1$. The full charge of such a $C$ is

$$(1,1,1,0)+(0,\alpha,0,\alpha)\in\widehat{Z}_G\times\widehat{Z}_{F,IR}\simeq\mathbb{Z}/2\mathbb{Z}\times\mathbb{Z}/2\mathbb{Z}\times\mathbb{Z}/2\mathbb{Z}\times\mathbb{Z}\,,\qquad(121)$$

which precisely matches the charges (118) seen by the ray index.

---

[28]A local operator is non-genuine if it cannot exist independently of a higher-dimensional defect.

[29]Notice that after the mass deformation the flavor symmetry is only $\mathfrak{so}(12)\oplus\mathfrak{u}(1)_I$.

Now, we can compute the charge of (120) under the center of $E_7$ arising in the conformal theory (when the inverse gauge coupling has been turned off) by using the full geometry (115) which sees the full $\mathfrak{e}_7$ symmetry. Under the $\mathbb{Z}_2$ center of $E_7$, the charge of $e$ is 0 (mod 2), the charge of $f$ is 0 (mod 2) and the charge of each $x_i$ is 1 (mod 2). Thus the charge of $C$ in (120) is 1 (mod 2), which is precisely what [19] find as discussed above. Thus, the whole information about the charges deduced from the geometry can equivalently be deduced from the ray index!

One might worry that the presence of these non-local operators would imply that $E_7/\mathbb{Z}_2$ bundles that do not lift to $E_7$ bundles cannot be turned on, i.e. the flavor symmetry of the conformal theory is $E_7$ instead of $E_7/\mathbb{Z}_2$. We propose that the conformal theory knows how to make sense of such non-genuine local operators in the presence of any $E_7/\mathbb{Z}_2$ background irrespective of whether it lifts to $E_7$ background or not. This can be easily seen by moving away from the conformal vacuum to a non-conformal vacuum on the CB of the conformal theory (but keeping inverse gauge coupling zero). According to the equality of (118) and (121), these non-genuine operators are associated to the curve (120), from which we can see that after moving onto the CB, these operators acquire a non-trivial charge $2\alpha + 2\beta + 2n + 1$ under the $U(1)$ gauge group arising at low-energies while their charge remains 1 (mod 2) under the center of $E_7$. These operators then arise at the ends of Wilson lines of charges $2\alpha + 2\beta + 2n + 1$ under the $U(1)$ gauge group. The abelian theory knows how to make sense of these non-genuine operators in the presence of an arbitrary $E_7/\mathbb{Z}_2$ bundle. This follows from the form of structure group

$$\mathcal{S} = \frac{U(1)_G \times E_7}{\mathbb{Z}_2} \tag{122}$$

of the low-energy abelian theory arising on CB, where $\mathbb{Z}_2$ is the diagonal combination of the $\mathbb{Z}_2$ subgroup of $U(1)_G$ and the $\mathbb{Z}_2$ center of $E_7$. The consequence of this structure group is that a gauge Wilson line of charge $2m + 1$ under $U(1)_G$ comes attached to a flavor Wilson line of charge 1 (mod 2) under the $\mathbb{Z}_2$ center of $E_7$. Thus, a non-genuine local operator living at the end of a gauge Wilson line of charge $2m + 1$ is left invariant under gauge transformations of $\mathcal{S}$ bundles.

## 4  2-Groups and Global Flavor: Higher Rank

In this section, we study the 2-group symmetry and global form of 0-form flavor symmetry groups of 5d SCFTs that UV complete 5d $\mathcal{N} = 1$ non-abelian gauge theories carrying simple, simply connected gauge groups. The M-theory construction of these SCFTs was discussed in [43] and the encoding of flavor symmetries in the geometry was discussed in [10] [30]. We only consider cases that have a non-trivial 1-form symmetry group (which can be determined by applying the analysis of [20,21]) and a non-trivial non-abelian part of the 0-form symmetry algebra (since the analysis of section 2.2.1 only applies to non-abelian flavor factors). Finally, all the theories exhibiting potentially non-trivial 2-group symmetries are collected in table 1 in section 1.3.

These examples show interesting interplay between perturbative and instantonic 2-group symmetries of the low-energy 5d gauge theories and the corresponding 5d SCFTs. We have already seen an example in the previous section, namely the $SU(2)_0$ SCFT, whose 2-group symmetry comes from the instantonic flavor symmetry, and hence is invisible from the point of view of the low-energy non-abelian gauge theory description.

---

[30]The geometric encoding of the flavor symmetries in a subset of these theories was also discussed in [5–7] which also studied many other classes of theories.

In this section, we will observe cases of 5d SCFTs carrying 2-group symmetries that are formed by perturbative flavor symmetries and are visible in the low energy non-abelian gauge theory description. One such example is the $SU(4)$ gauge theory with Chern-Simons (CS) level 2 and a hyper in 2-index antisymmetric irreducible representation (irrep) of the $SU(4)$ gauge group. There is a perturbative $SO(3)$ flavor symmetry group rotating the antisymmetric hyper which forms a 2-group with the $\mathbb{Z}_2$ 1-form symmetry. See the discussion surrounding (194) for a detailed study of this example.

On the other hand, we also observe cases of 5d SCFTs which do not carry a 2-group symmetry even though the associated low energy non-abelian gauge theory carries a 2-group symmetry. As we explain in section 2.2.3, this is due to the effect of instanton BPS particles which break the 2-group symmetry. Similar considerations hold for the global form of flavor group, which might acquire more center charges in passing from IR non-abelian gauge theory to UV SCFT, due to the contributions of instantons. One such example is the $SU(4)$ gauge theory with a hyper in 2-index antisymmetric irrep but now with CS level 0. The low-energy non-abelian gauge theory predicts that the global form of the $\mathfrak{su}(2)$ flavor symmetry algebra rotating the antisymmetric hyper is $SO(3)$ which forms a 2-group with the $\mathbb{Z}_2$ 1-form symmetry. However, including the contribution of instantons following section 2.2.3, we find that in the corresponding 5d SCFT the global form associated to $\mathfrak{su}(2)$ flavor algebra is $SU(2)$ and there is no 2-group symmetry.

## 4.1  General Rank

$\underline{SU(n)_n}$:  Consider the 5d SCFT $\mathfrak{T}$ which is the UV completion of 5d pure $SU(n)$ gauge theory with CS level $n$. The corresponding geometry is

$$N \xrightarrow{\; f \quad e \;} \mathbf{1}_2 \xrightarrow{\; h \quad e \;} \mathbf{2}_4 \xrightarrow{\; h \;} \cdots \xrightarrow{\; e \;} (\mathbf{n-1})_{2n-2} \; . \tag{123}$$

On the CB of $\mathfrak{T}$, we have $G = \prod_{i=1}^{n-1} U(1)_i$ associated to the $n-1$ compact surfaces $S_i$ and $F = SU(2)$ associated to the non-compact surface $N$. We compute that $\mathcal{E} \simeq \mathbb{Z}_{2n}$ generated by $\left(\frac{n-1}{2n}, \frac{n-2}{2n}, \cdots, \frac{1}{2n}, 1\right) \in Z_G \times Z_F \simeq (\mathbb{R}/\mathbb{Z})^{n-1} \times (\mathbb{Z}/2\mathbb{Z})$. From this, we compute $\mathcal{O} \simeq \mathbb{Z}_n$ generated by $\left(\frac{n-1}{n}, \frac{n-2}{n}, \cdots, \frac{1}{n}, 0\right) \in Z_G \times Z_F$ which is identified with the 1-form symmetry group $\mathcal{O}_{\mathfrak{T}}$ of $\mathfrak{T}$. We also have $\mathcal{Z} \simeq \mathbb{Z}_2$ generated by $1 \in Z_F$. Thus, according to our proposal 0-form symmetry group $\mathcal{F}_{\mathfrak{T}}$ of $\mathfrak{T}$ is

$$\mathcal{F}_{\mathfrak{T}} = \frac{SU(2)}{\mathbb{Z}_2} = SO(3). \tag{124}$$

The short exact sequence (13) becomes

$$0 \to \mathbb{Z}_n \to \mathbb{Z}_{2n} \to \mathbb{Z}_2 \to 0. \tag{125}$$

The projection of $\mathcal{E}$ onto $Z_G$ is $\mathcal{E}' \simeq \mathbb{Z}_{2n}$ generated by $\left(\frac{n-1}{2n}, \frac{n-2}{2n}, \cdots, \frac{1}{2n}\right) \in Z_G$, implying that the projection map is injective. Thus, following the arguments of section 2.1 and using our proposal of section 2.2.1, we find that $\mathfrak{T}$ has a potential 2-group structure formed by 0-form symmetry group $\mathcal{F}_{\mathfrak{T}} = SO(3)$ and 1-form symmetry group $\mathcal{O}_{\mathfrak{T}} \simeq \mathbb{Z}_n$ with the Postnikov class being

$$\text{Bock}(\widehat{v}_2) \in H^3(BSO(3), \mathbb{Z}_2), \tag{126}$$

where $\widehat{v}_2 \in H^2(BSO(3), \mathbb{Z}_2)$ is the characteristic class capturing obstruction of lifting $\mathcal{F}_{\mathfrak{T}} = SO(3)$ bundles to $SU(2)$ bundles, and Bock is the Bockstein corresponding to the above short exact sequence (125).

The short exact sequence (125) splits for odd $n$ implying that Bock is a trivial homomorphism for odd $n$. Thus, the 2-group structure is trivial for odd $n$, i.e. the total symmetry group



is a direct product of 1-form symmetry group $\mathbb{Z}_2$ and 0-form symmetry group $SO(3)$. For even $n$, (125) does not split, and hence the theory for even $n$ admits a non-trivial 2-group structure if $\text{Bock}(\widehat{v}_2)$ is a non-trivial element of $H^3(BSO(3), \mathbb{Z}_2)$.

$\underline{SU(2m)_{m+2} + \mathbf{\Lambda}^2;\ m > 2:}$   Consider the 5d SCFT $\mathfrak{T}$ which is the UV completion of 5d $SU(2m)$ gauge theory having a hyper in 2-index antisymmetric irrep and CS level $m + 2$, with $m > 2$. The corresponding geometry is

$$N \xrightarrow{\quad f \quad\quad e \quad} \mathbf{1}_2 \xrightarrow{\quad h \quad\quad e \quad} \mathbf{2}_4 \xrightarrow{\ h\ } \cdots \xrightarrow{\ e\ } (\mathbf{2m-2})^1_{4m-4} \xrightarrow{\quad h \quad\quad e \quad} (\mathbf{2m-1})_{4m-2} .$$
$$\tag{127}$$

Notice that we have one blowup on $S_{2m-2}$. $N$ describes the $\mathfrak{su}(2)$ flavor symmetry of $\mathfrak{T}$. Thus, $Z_F \simeq \mathbb{Z}/2\mathbb{Z}$. We find that $\mathcal{E} \simeq \mathbb{Z}_2$ generated by $\left(\frac{1}{2}, 0, \frac{1}{2}, 0, \cdots, \frac{1}{2}, 0\right) \in Z_G \times Z_F \simeq (\mathbb{R}/\mathbb{Z})^{2m-1} \times \mathbb{Z}/2\mathbb{Z}$. From this we see that $\mathcal{O} = \mathcal{E} \simeq \mathbb{Z}_2$ which is identified with the 1-form symmetry group $\mathcal{O}_{\mathfrak{T}} \simeq \mathbb{Z}_2$ of $\mathfrak{T}$. Thus, the non-abelian part of the 0-form flavor symmetry group is

$$\mathcal{F}_{\mathfrak{T}} = SU(2) \tag{128}$$

and there is no non-trivial 2-group structure between $\mathcal{O}_{\mathfrak{T}}$ and $\mathcal{F}_{\mathfrak{T}}$.

$\underline{Sp(2m-1)_0 + \mathbf{\Lambda}^2 = SU(2m)_{m+4} + \mathbf{\Lambda}^2;\ Sp(2m)_0 + \mathbf{\Lambda}^2:}$   Consider 5d SCFT $\mathfrak{T}$ which is UV completion of 5d $Sp(n)$ gauge theory having a hyper in 2-index antisymmetric irrep and discrete theta angle 0. For odd $n = 2m - 1$, $\mathfrak{T}$ is also the UV completion of 5d $SU(2m)$ gauge theory having a hyper in 2-index antisymmetric irreducible representation and CS level $m + 4$. The corresponding geometry is

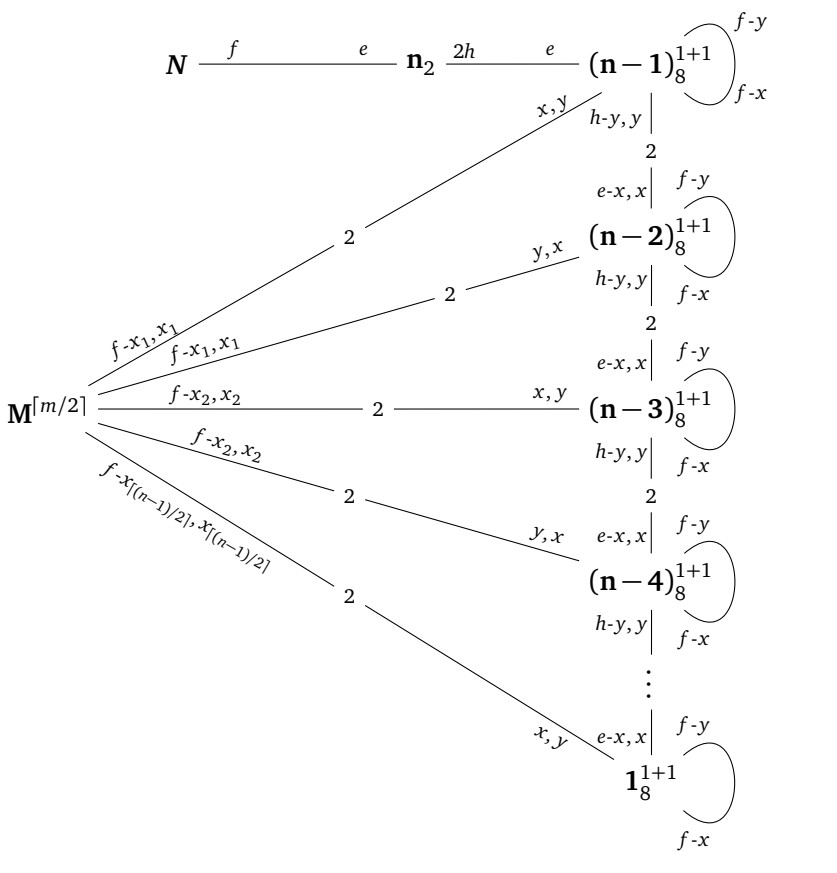

$$\tag{129}$$

$N$ and $\mathbf{M}$ describe $\mathfrak{su}(2) \oplus \mathfrak{su}(2)$ flavor symmetry of $\mathfrak{T}$. Thus, $Z_F \simeq (\mathbb{Z}/2\mathbb{Z})_N \times (\mathbb{Z}/2\mathbb{Z})_M$. We find that $\mathcal{E} \simeq \mathbb{Z}_4$ generated by $\left(\frac{1}{4}, \frac{2}{4}, \cdots, \frac{n}{4}, 1, 0\right) \in Z_G \times Z_F \simeq (\mathbb{R}/\mathbb{Z})^n \times (\mathbb{Z}/2\mathbb{Z})_N \times (\mathbb{Z}/2\mathbb{Z})_M$. From this we see that $\mathcal{O} \simeq \mathbb{Z}_2$ generated by $\left(\frac{2}{4}, \frac{4}{4}, \cdots, \frac{2n}{4}, 0, 0\right) \in Z_G \times Z_F$, which is identified with the 1-form symmetry group $\mathcal{O}_{\mathfrak{T}} \simeq \mathbb{Z}_2$ of $\mathfrak{T}$. We have $\mathcal{Z} \simeq \mathbb{Z}_2$ generated by $(1, 0) \in Z_F$. Thus, the non-abelian part of the 0-form flavor symmetry group is

$$\mathcal{F}_{\mathfrak{T}} = SO(3)_N \times SU(2)_M . \tag{130}$$

The projection of $\mathcal{E}$ onto $Z_G$ is $\mathcal{E}' \simeq \mathbb{Z}_4$ generated by $\left(\frac{1}{4}, \frac{2}{4}, \cdots, \frac{n}{4}\right) \in Z_G$, implying that the projection map is injective. The short exact sequence (13) becomes

$$0 \to \mathbb{Z}_2 \to \mathbb{Z}_4 \to \mathbb{Z}_2 \to 0 . \tag{131}$$

Thus, the 1-form symmetry group $\mathcal{O}_{\mathfrak{T}} \simeq \mathbb{Z}_2$ and $SO(3)_N$ part of $\mathcal{F}_{\mathfrak{T}}$ form a non-trivial 2-group with the Postnikov class being

$$\mathrm{Bock}(\widetilde{v}_2) \in H^3\left(BSO(3)_N, \mathbb{Z}_2\right) , \tag{132}$$

where $\widetilde{v}_2 \in H^2\left(BSO(3)_N, \mathbb{Z}_2\right)$ is the characteristic class capturing obstruction of lifting $SO(3)_N$ bundles to $SU(2)_N$ bundles, and Bock is the Bockstein associated to the above short exact sequence. As discussed in the case of $SU(2)_0$, $\widetilde{v}_2$ can be identified as the second Stiefel-Whitney class $w_2$ and the Postnikov class $\mathrm{Bock}(\widetilde{v}_2)$ can be identified as the third Stiefel-Whitney class $w_3$. Since the Postnikov class is non-trivial, we have a non-trivial 2-group structure.

$\underline{SU(2m+2)_4 + 2\Lambda^2; \; m > 1:}$   Consider 5d SCFTs which are UV completions of 5d $SU(2m+2)$ gauge theories having 2 hypers in 2-index antisymmetric irrep and CS level 4, with $m > 1$. Let us first consider the $m = 2n$ case. The corresponding geometry is

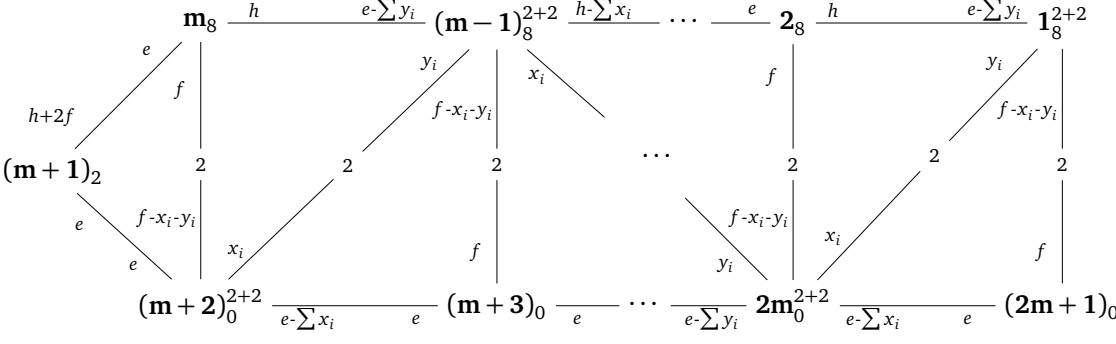

$$\tag{133}$$

with extra gluing rules:

- $e - y_1 - y_2$ in $S_{m+2}$ is glued to $f$ in $N$.

- $x_1, x_2$ in $S_1$ are glued to $x_1, x_2$ in $\mathbf{P}$.

- $e$ in $S_{2m+1}$ is glued to $f - x_1 - x_2$ in $\mathbf{P}$.

- $x_1 - x_2, y_2 - y_1$ in $S_{m+2i}$ are glued to $f, f$ in $\mathbf{M}$ for $i = 1, \cdots, \frac{m}{2}$.

- $x_2 - x_1, y_1 - y_2$ in $S_{m+1-2i}$ are glued to $f, f$ in $\mathbf{M}$ for $i = 1, \cdots, \frac{m}{2}$.

- $x_2 - x_1$ in $\mathbf{P}$ is glued to $f$ in $\mathbf{M}$.

$M, N, P$ describe three $\mathfrak{su}(2)$ flavor symmetries of $\mathfrak{T}$. The $\mathfrak{su}(2)_M$ flavor symmetry is perturbative while the $\mathfrak{su}(2)_{N,P}$ flavor symmetries are instantonic. Thus, $Z_F \simeq (\mathbb{Z}/2\mathbb{Z})_M \times (\mathbb{Z}/2\mathbb{Z})_N \times (\mathbb{Z}/2\mathbb{Z})_P$. We find that $\mathcal{E} \simeq \mathbb{Z}_4 \times \mathbb{Z}_2$. The $\mathbb{Z}_4$ subfactor of $\mathcal{E}$ is generated by $(\alpha_1, \cdots, \alpha_{2m+1}, 1, 0, 1) \in Z_G \times Z_F \simeq (\mathbb{R}/\mathbb{Z})^{2m+1} \times (\mathbb{Z}/2\mathbb{Z})_N \times (\mathbb{Z}/2\mathbb{Z})_M \times (\mathbb{Z}/2\mathbb{Z})_P$ where $\alpha_i = \frac{4-i}{4}$ for $1 \le i \le m+2$ and $\alpha_i = \frac{i-2m}{4}$ for $m+3 \le i \le 2m+1$. The $\mathbb{Z}_2$ subfactor of $\mathcal{E}$ is generated by $(\beta_1, \cdots, \beta_{2m+1}, 1, 1, 0) \in Z_G \times Z_F$ where $\beta_i = \frac{1}{2}$ if $i = m+1+2j$ for $j \ge 1$ and $\beta_i = 0$ otherwise. From this we see that $\mathcal{O} \simeq \mathbb{Z}_2$ generated by $(2\alpha_1, \cdots, 2\alpha_{2m+1}, 0, 0, 0) \in Z_G \times Z_F$ which is identified with the 1-form symmetry group $\mathcal{O}_{\mathfrak{T}} \simeq \mathbb{Z}_2$ of $\mathfrak{T}$. We have $\mathcal{Z} \simeq \mathbb{Z}_2^2$ generated by $(1, 0, 1) \in Z_F$ and $(1, 1, 0) \in Z_F$. Thus, the 0-form flavor symmetry group $\mathcal{F}_{\mathfrak{T}}$ of $\mathfrak{T}$ is

$$\mathcal{F}_{\mathfrak{T}} = SU(2)_N \times SO(3)_{N,M} \times SO(3)_{N,P}, \tag{134}$$

where $SO(3)_{N,M} = SU(2)_{N,M}/\mathbb{Z}_2^{N,M}$ where $SU(2)_{N,M}$ is the diagonal combination of $SU(2)_N$ and $SU(2)_M$, and $\mathbb{Z}_2^{N,M}$ is the center of $SU(2)_{N,M}$. Similarly $SO(3)_{N,P} = SU(2)_{N,P}/\mathbb{Z}_2^{N,P}$ where $SU(2)_{N,P}$ is the diagonal combination of $SU(2)_N$ and $SU(2)_P$, and $\mathbb{Z}_2^{N,P}$ is the center of $SU(2)_{N,P}$.

The projection of $\mathcal{E}$ onto $Z_G$ is $\mathcal{E}' \simeq \mathbb{Z}_4 \times \mathbb{Z}_2$ generated by $(\alpha_1, \cdots, \alpha_{2m+1}) \in Z_G$ and $(\beta_1, \cdots, \beta_{2m+1}) \in Z_G$ implying that the projection map is injective. The short exact sequence (13) becomes

$$0 \to \mathbb{Z}_2 \to \mathbb{Z}_4 \times \mathbb{Z}_2 \to \mathbb{Z}_2^{N,P} \times \mathbb{Z}_2^{N,M} \to 0, \tag{135}$$

whose non-trivial part is only

$$0 \to \mathbb{Z}_2 \to \mathbb{Z}_4 \to \mathbb{Z}_2^{N,P} \to 0. \tag{136}$$

Thus, we learn that $\mathcal{O}_{\mathfrak{T}}$ and $SO(3)_{N,P} \times SO(3)_{N,M}$ part of $\mathcal{F}_{\mathfrak{T}}$ form a non-trivial 2-group with the Postnikov class being

$$\text{Bock}(\widetilde{\nu}_2) \in H^3\left(B[SO(3)_{N,P} \times SO(3)_{N,M}], \mathbb{Z}_2\right), \tag{137}$$

where $\widetilde{\nu}_2 \in H^2\left(B[SO(3)_{N,P} \times SO(3)_{N,M}], \mathbb{Z}_2^{N,P}\right)$ is the characteristic class capturing obstruction of lifting $SO(3)_{N,P} \times SO(3)_{N,M}$ bundles to $SU(2)_{N,P} \times SO(3)_{N,M}$ bundles, and Bock is the Bockstein associated to the above short exact sequence (136).

Let us now consider the $m = 2n+1$ case. The corresponding geometry is

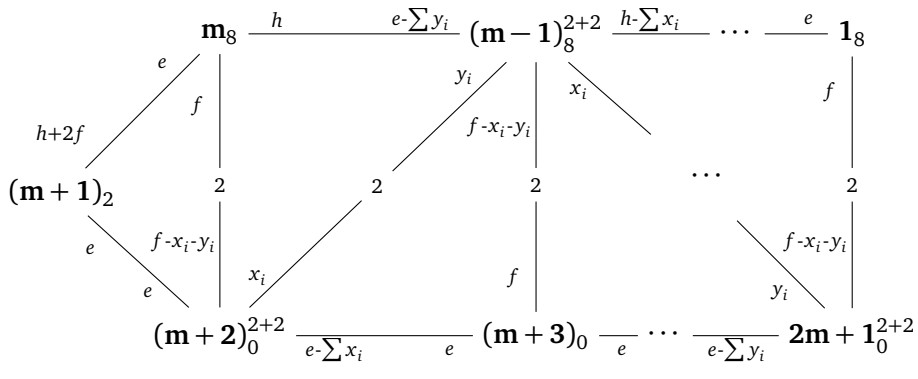

$$\tag{138}$$

with extra gluing rules:

- $e - y_1 - y_2$ in $S_{m+2}$ is glued to $f$ in $N$.

- $e - x_1 - x_2$ in $\mathbf{S}_{2m+1}$ is glued to $f$ in $\mathbf{P}$.

- $x_1 - x_2, y_2 - y_1$ in $\mathbf{S}_{m+2i}$ are glued to $f, f$ in $\mathbf{M}$ for $i = 1, \cdots, \frac{m+1}{2}$.

- $x_2 - x_1, y_1 - y_2$ in $\mathbf{S}_{m+1-2i}$ are glued to $f, f$ in $\mathbf{M}$ for $i = 1, \cdots, \frac{m-1}{2}$.

$\mathbf{M}, N, \mathbf{P}$ describe three $\mathfrak{su}(2)$ flavor symmetries of $\mathfrak{T}$. The $\mathfrak{su}(2)_M$ flavor symmetry is perturbative while the $\mathfrak{su}(2)_{N,P}$ flavor symmetries are instantonic. Thus, $Z_F \simeq (\mathbb{Z}/2\mathbb{Z})_M \times (\mathbb{Z}/2\mathbb{Z})_N \times (\mathbb{Z}/2\mathbb{Z})_P$. We find that $\mathcal{E} \simeq \mathbb{Z}_4 \times \mathbb{Z}_2$. The $\mathbb{Z}_4$ subfactor of $\mathcal{E}$ is generated by $(\alpha_1, \cdots, \alpha_{2m+1}, 1, 0, 1) \in Z_G \times Z_F \simeq (\mathbb{R}/\mathbb{Z})^{2m+1} \times (\mathbb{Z}/2\mathbb{Z})_N \times (\mathbb{Z}/2\mathbb{Z})_M \times (\mathbb{Z}/2\mathbb{Z})_P$ where $\alpha_i = \frac{4-i}{4}$ for $1 \le i \le m+2$ and $\alpha_i = \frac{i-2m}{4}$ for $m+3 \le i \le 2m+1$. The $\mathbb{Z}_2$ subfactor of $\mathcal{E}$ is generated by $(\beta_1, \cdots, \beta_{2m+1}, 1, 1, 1) \in Z_G \times Z_F$ where $\beta_i = \frac{1}{2}$ if $i = m+1+2j$ for $j \ge 1$ and $\beta_i = 0$ otherwise. From this we see that $\mathcal{O} \simeq \mathbb{Z}_2$ generated by $(2\alpha_1, \cdots, 2\alpha_{2m+1}, 0, 0, 0) \in Z_G \times Z_F$ which is identified with the 1-form symmetry group $\mathcal{O}_{\mathfrak{T}} \simeq \mathbb{Z}_2$ of $\mathfrak{T}$. We have $\mathcal{Z} \simeq \mathbb{Z}_2^2$ generated by $(1, 0, 1) \in Z_F$ and $(0, 1, 0) \in Z_F$. Thus, the 0-form flavor symmetry group $\mathcal{F}_{\mathfrak{T}}$ of $\mathfrak{T}$ is

$$\mathcal{F}_{\mathfrak{T}} = SO(3)_M \times SU(2)_N \times SO(3)_{N,M,P}, \tag{139}$$

where $SO(3)_{N,M,P} = SU(2)_{N,M,P}/\mathbb{Z}_2^{N,M,P}$ where $SU(2)_{N,M,P}$ is the diagonal combination of $SU(2)_N$, $SU(2)_M$ and $SU(2)_P$, and $\mathbb{Z}_2^{N,M,P}$ is the center of $SU(2)_{N,M,P}$.

The projection of $\mathcal{E}$ onto $Z_G$ is $\mathcal{E}' \simeq \mathbb{Z}_4 \times \mathbb{Z}_2$ generated by $(\alpha_1, \cdots, \alpha_{2m+1}) \in Z_G$ and $(\beta_1, \cdots, \beta_{2m+1}) \in Z_G$ implying that the projection map is injective. The short exact sequence (13) becomes

$$0 \to \mathbb{Z}_2 \to \mathbb{Z}_4 \times \mathbb{Z}_2 \to \mathbb{Z}_2^M \times \mathbb{Z}_2^{N,M,P} \to 0, \tag{140}$$

where $\mathbb{Z}_2^M$ is the center of $SU(2)_M$ and $\mathbb{Z}_2^{N,M,P}$ is the diagonal combination of the centers of $SU(2)_M$, $SU(2)_N$ and $SU(2)_P$. The non-trivial part of the above short exact sequence is

$$0 \to \mathbb{Z}_2 \to \mathbb{Z}_4 \to \mathbb{Z}_2^M \to 0. \tag{141}$$

Thus, we learn that $\mathcal{O}_{\mathfrak{T}}$ and $SO(3)_{N,M,P} \times SO(3)_M$ part of $\mathcal{F}_{\mathfrak{T}}$ form a non-trivial 2-group with the Postnikov class being

$$\text{Bock}(\widetilde{\nu}_2) \in H^3\left(B\left[SO(3)_{N,M,P} \times SO(3)_M\right], \mathbb{Z}_2\right), \tag{142}$$

where $\widetilde{\nu}_2 \in H^2\left(B\left[SO(3)_{N,M,P} \times SO(3)_M\right], \mathbb{Z}_2^M\right)$ is the characteristic class capturing obstruction of lifting $SO(3)_{N,M,P} \times SO(3)_M$ bundles to $SO(3)_{N,M,P} \times SU(2)_M$ bundles, and Bock is the Bockstein associated to the above short exact sequence (141).

$\underline{SU(2m+2)_0 + 2\Lambda^2; \ m > 1}$ : Consider 5d SCFTs which are UV completions of 5d $SU(2m+2)$ gauge theories having 2 hypers in 2-index antisymmetric irrep and CS level 0, with $m > 1$. Let

us first consider the $m = 2n$ case. The corresponding geometry is

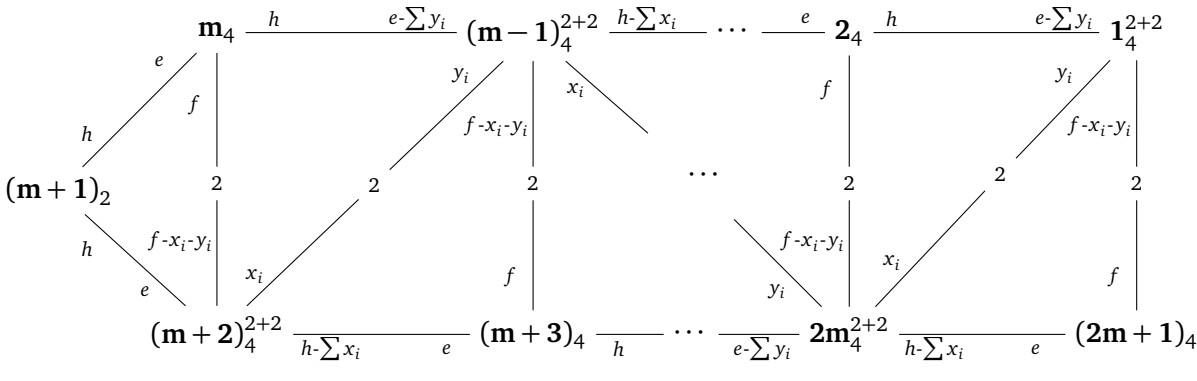

$$(143)$$

with extra gluing rules:

- $e$ in $S_{m+1}$ is glued to $f$ in $N$.

- $x_1 - x_2, y_2 - y_1$ in $S_{m+2i}$ are glued to $f, f$ in $M$ for $i = 1, \cdots, \frac{m}{2}$.

- $x_2 - x_1, y_1 - y_2$ in $S_{m+1-2i}$ are glued to $f, f$ in $M$ for $i = 1, \cdots, \frac{m}{2}$.

$M, N$ describe two $\mathfrak{su}(2)$ flavor symmetries of $\mathfrak{T}$. The $\mathfrak{su}(2)_M$ flavor symmetry is perturbative while the $\mathfrak{su}(2)_N$ flavor symmetry is instantonic. Thus, $Z_F \simeq (\mathbb{Z}/2\mathbb{Z})_M \times (\mathbb{Z}/2\mathbb{Z})_N$. We find that $\mathcal{E} \simeq \mathbb{Z}_4$ generated by $(\alpha_1, \cdots, \alpha_{2m+1}, 1, 0) \in Z_G \times Z_F \simeq (\mathbb{R}/\mathbb{Z})^{2m+1} \times (\mathbb{Z}/2\mathbb{Z})_N \times (\mathbb{Z}/2\mathbb{Z})_M$ where $\alpha_{2m+2-i} = \alpha_i = \frac{4-i}{4}$ for $1 \le i \le m+1$. From this we see that $\mathcal{O} \simeq \mathbb{Z}_2$ generated by $(2\alpha_1, \cdots, 2\alpha_{2m+1}, 0, 0) \in Z_G \times Z_F$ which is identified with the 1-form symmetry group $\mathcal{O}_{\mathfrak{T}} \simeq \mathbb{Z}_2$ of $\mathfrak{T}$. We have $\mathcal{Z} \simeq \mathbb{Z}_2$ generated by $(1, 0) \in Z_F$. Thus, the 0-form flavor symmetry group $\mathcal{F}_{\mathfrak{T}}$ of $\mathfrak{T}$ is

$$\mathcal{F}_{\mathfrak{T}} = SO(3)_N \times SU(2)_M. \tag{144}$$

The projection of $\mathcal{E}$ onto $Z_G$ is $\mathcal{E}' \simeq \mathbb{Z}_4$ generated by $(\alpha_1, \cdots, \alpha_{2m+1}) \in Z_G$ implying that the projection map is injective. The short exact sequence (13) becomes

$$0 \to \mathbb{Z}_2 \to \mathbb{Z}_4 \to \mathbb{Z}_2 \to 0. \tag{145}$$

Thus, we learn that $\mathcal{O}_{\mathfrak{T}} \simeq \mathbb{Z}_2$ and $SO(3)_N$ part of $\mathcal{F}_{\mathfrak{T}}$ form a non-trivial 2-group with the Postnikov class being

$$\text{Bock}(\widetilde{\nu}_2) \in H^3(BSO(3)_N, \mathbb{Z}_2), \tag{146}$$

where $\widetilde{\nu}_2 \in H^2(BSO(3)_N, \mathbb{Z}_2)$ is the characteristic class capturing obstruction of lifting $SO(3)_N$ bundles to $SU(2)_N$ bundles, and Bock is the Bockstein associated to the above short exact sequence.

Let us now consider the $m = 2n + 1$ case. The corresponding geometry is

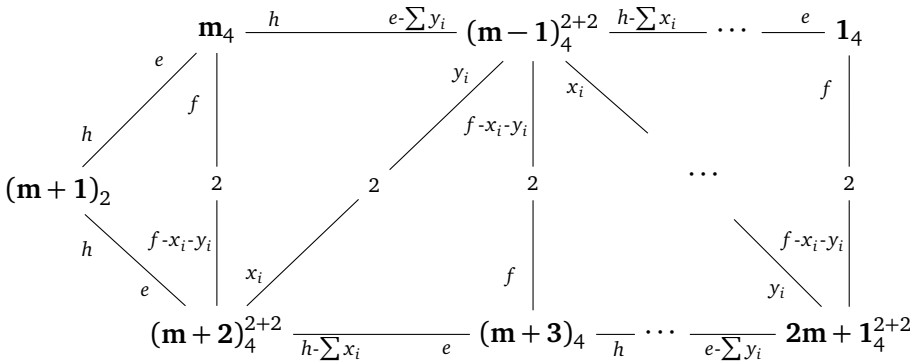

$$N \qquad\qquad M \tag{147}$$

with extra gluing rules:

- $e$ in $S_{m+1}$ is glued to $f$ in $N$.

- $x_1 - x_2, y_2 - y_1$ in $S_{m+2i}$ are glued to $f, f$ in $M$ for $i = 1, \cdots, \frac{m+1}{2}$.

- $x_2 - x_1, y_1 - y_2$ in $S_{m+1-2i}$ are glued to $f, f$ in $M$ for $i = 1, \cdots, \frac{m-1}{2}$.

$M, N$ describe two $\mathfrak{su}(2)$ flavor symmetries of $\mathfrak{T}$. The $\mathfrak{su}(2)_M$ flavor symmetry is perturbative while the $\mathfrak{su}(2)_N$ flavor symmetry is instantonic. Thus, $Z_F \simeq (\mathbb{Z}/2\mathbb{Z})_M \times (\mathbb{Z}/2\mathbb{Z})_N$. We find that $\mathcal{E} \simeq \mathbb{Z}_4 \times \mathbb{Z}_2$. The $\mathbb{Z}_4$ subfactor of $\mathcal{E}$ is generated by $(\alpha_1, \cdots, \alpha_{2m+1}, 1, 0) \in Z_G \times Z_F \simeq (\mathbb{R}/\mathbb{Z})^{2m+1} \times (\mathbb{Z}/2\mathbb{Z})_N \times (\mathbb{Z}/2\mathbb{Z})_M$ where $\alpha_{2m+2-i} = \alpha_i = \frac{4-i}{4}$ for $1 \le i \le m+1$. The $\mathbb{Z}_2$ subfactor of $\mathcal{E}$ is generated by $(\beta_1, \cdots, \beta_{2m+1}, 1, 1) \in Z_G \times Z_F$ where $\beta_i = \frac{1}{2}$ if $i = m + 2j$ for $j \ge 1$ and $\beta_i = 0$ otherwise. From this we see that $\mathcal{O} \simeq \mathbb{Z}_2$ generated by $(2\alpha_1, \cdots, 2\alpha_{2m+1}, 0, 0) \in Z_G \times Z_F$ which is identified with the 1-form symmetry group $\mathcal{O}_{\mathfrak{T}} \simeq \mathbb{Z}_2$ of $\mathfrak{T}$. We have $\mathcal{Z} \simeq \mathbb{Z}_2^2$ generated by $(1, 0) \in Z_F$ and $(0, 1) \in Z_F$. Thus, the 0-form flavor symmetry group $\mathcal{F}_{\mathfrak{T}}$ of $\mathfrak{T}$ is

$$\mathcal{F}_{\mathfrak{T}} = SO(3)_N \times SO(3)_{N,M}. \tag{148}$$

The projection of $\mathcal{E}$ onto $Z_G$ is $\mathcal{E}' \simeq \mathbb{Z}_4 \times \mathbb{Z}_2$ generated by $(\alpha_1, \cdots, \alpha_{2m+1}) \in Z_G$ and $(\beta_1, \cdots, \beta_{2m+1}) \in Z_G$ implying that the projection map is injective. The short exact sequence (13) becomes

$$0 \to \mathbb{Z}_2 \to \mathbb{Z}_4 \to \mathbb{Z}_2^N \times \mathbb{Z}_2^{N,M} \to 0. \tag{149}$$

The non-trivial part of the above short exact sequence is

$$0 \to \mathbb{Z}_2 \to \mathbb{Z}_4 \to \mathbb{Z}_2^N \to 0. \tag{150}$$

Thus, we learn that $\mathcal{O}_{\mathfrak{T}}$ and $\mathcal{F}_{\mathfrak{T}}$ form a non-trivial 2-group with the Postnikov class being

$$\mathrm{Bock}(\widetilde{v}_2) \in H^3\big(B\big[SO(3)_N \times SO(3)_{N,M}\big], \mathbb{Z}_2\big), \tag{151}$$

where $\widetilde{v}_2 \in H^2\big(B\big[SO(3)_N \times SO(3)_{N,M}\big], \mathbb{Z}_2^N\big)$ is the characteristic class capturing obstruction of lifting $SO(3)_N \times SO(3)_{N,M}$ bundles to $SU(2)_N \times SO(3)_{N,M}$ bundles, and Bock is the Bockstein associated to the above short exact sequence (150).

$\underline{SU(2m+2)_2 + 2\Lambda^2;\ m > 1}$ : Consider 5d SCFTs which are UV completions of 5d $SU(2m+2)$ gauge theories having 2 hypers in 2-index antisymmetric irrep and CS level 2, with $m > 1$. The 1-form symmetry for such a theory is $\mathcal{O}_{\mathfrak{T}} \simeq \mathbb{Z}_2$. The non-abelian part of the flavor symmetry algebra is $\mathfrak{f}_{\mathfrak{T}} = \mathfrak{su}(2)$.

Since there is no enhancement of flavor symmetry at the conformal point, we can apply the analysis in terms of section 2.2.3. We have $Z_G = \mathbb{Z}_{2m+2}$, $Z_F = \mathbb{Z}_2$. The hypers are charged as $(2, 1)$ under $Z_G \times Z_F$ and the instanton particle can be taken to have to charge $(4 - 2m, 0)$ under $Z_G \times Z_F$. From this we compute that $\mathcal{E} = \mathcal{O} = \mathbb{Z}_2$ and thus the 0-form flavor symmetry group is

$$\mathcal{F}_{\mathfrak{T}} = SU(2), \tag{152}$$

and there is no 2-group structure.

$\underline{SU(2n+2)_{n-1} + S^2}$ : Consider the 5d SCFT $\mathfrak{T}$ which is the UV completion of 5d $SU(2n+2)$ gauge theory having a hyper in 2-index symmetric irrep and CS level $n-1$. The corresponding geometry is

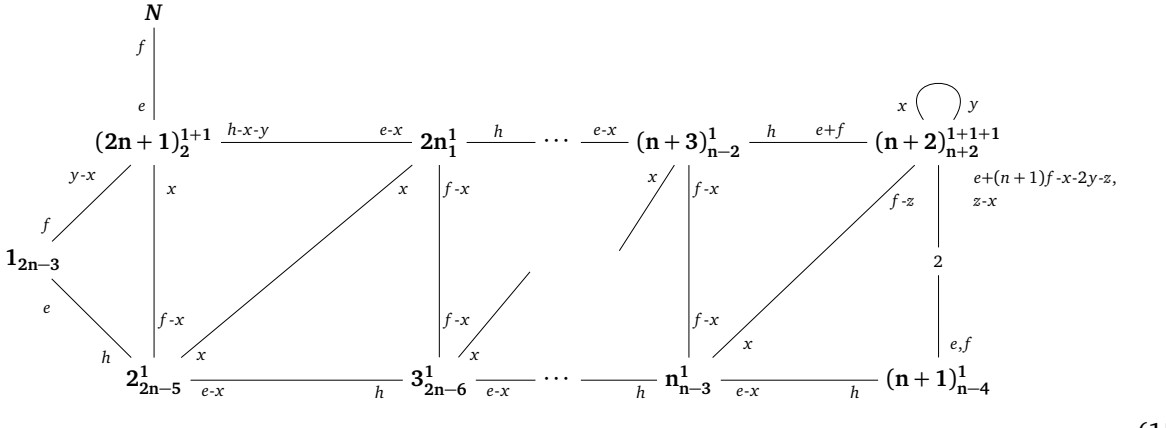

$$\tag{153}$$

$N$ describes an $\mathfrak{su}(2)$ flavor symmetry of $\mathfrak{T}$. Thus, $Z_F \simeq \mathbb{Z}/2\mathbb{Z}$. We find that $\mathcal{E} \simeq \mathbb{Z}_2$ generated by $\left(\frac{1}{2}, 0, \frac{1}{2}, 0, \cdots, \frac{1}{2}, 0, \frac{1}{2}, 0\right) \in Z_G \times Z_F \simeq (\mathbb{R}/\mathbb{Z})^{2n+1} \times \mathbb{Z}/2\mathbb{Z}$. From this we see that $\mathcal{O} = \mathcal{E} \simeq \mathbb{Z}_2$ which is identified with the 1-form symmetry group $\mathcal{O}_{\mathfrak{T}} \simeq \mathbb{Z}_2$ of $\mathfrak{T}$. Thus, the non-abelian part of the 0-form flavor symmetry group is

$$\mathcal{F}_{\mathfrak{T}} = SU(2), \tag{154}$$

and there is no non-trivial 2-group structure between $\mathcal{O}_{\mathfrak{T}}$ and $\mathcal{F}_{\mathfrak{T}}$.

**Spin**$(m+3)+m\boldsymbol{F}$ : Consider the 5d SCFT $\mathfrak{T}$ which is the UV completion of 5d Spin$(m+3)$ gauge theory having $m$ hypers in vector irrep. For $m=2n$, the corresponding geometry is

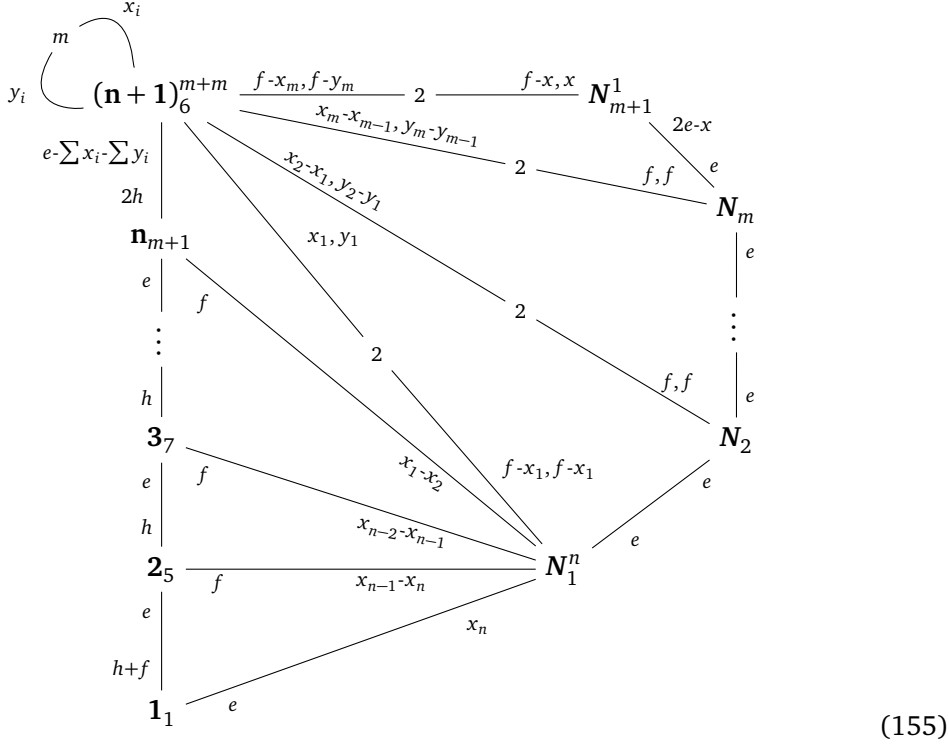

$$\tag{155}$$

$N_i$ describe the $\mathfrak{sp}(m+1)$ flavor symmetry of $\mathfrak{T}$. Thus, $Z_F \simeq \mathbb{Z}/2\mathbb{Z}$. We find that $\mathcal{E} \simeq \mathbb{Z}_2$ generated by $\left(\frac{1}{2}, 0, 0, \cdots, 0, 0\right) \in Z_G \times Z_F \simeq (\mathbb{R}/\mathbb{Z})^{n+1} \times \mathbb{Z}/2\mathbb{Z}$. From this we see that $\mathcal{O} = \mathcal{E} \simeq \mathbb{Z}_2$ which is identified with the 1-form symmetry group $\mathcal{O}_{\mathfrak{T}} \simeq \mathbb{Z}_2$ of $\mathfrak{T}$. Thus, the 0-form flavor symmetry group is

$$\mathcal{F}_{\mathfrak{T}} = Sp(m+1) \tag{156}$$

and there is no non-trivial 2-group structure between $\mathcal{O}_{\mathfrak{T}}$ and $\mathcal{F}_{\mathfrak{T}}$.

For $m = 2n - 1$, the corresponding geometry is

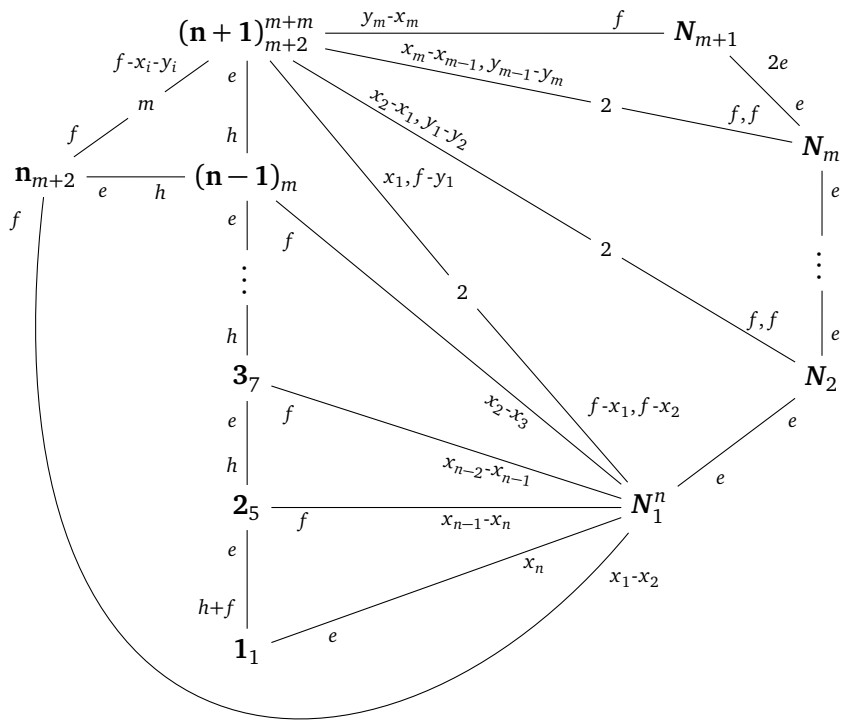

$$(157)$$

$N_i$ describe the $\mathfrak{sp}(m+1)$ flavor symmetry of $\mathfrak{T}$. Thus, $Z_F \simeq \mathbb{Z}/2\mathbb{Z}$.

For $n = 2k$, we find that $\mathcal{E} \simeq \mathbb{Z}_2 \times \mathbb{Z}_2$ generated by $\left(0, \frac{1}{2}, 0, \frac{1}{2}, \cdots, 0, \frac{1}{2}, 0, 1\right) \in Z_G \times Z_F \simeq (\mathbb{R}/\mathbb{Z})^{n+1} \times \mathbb{Z}/2\mathbb{Z}$ and $\left(0, 0, \cdots, 0, \frac{1}{2}, \frac{1}{2}, 0\right) \in Z_G \times Z_F$. From this we see that $\mathcal{O} \simeq \mathbb{Z}_2$ is the second $\mathbb{Z}_2$ subfactor of $\mathcal{E}$ which is identified with the 1-form symmetry group $\mathcal{O}_{\mathfrak{T}} \simeq \mathbb{Z}_2$ of $\mathfrak{T}$. We have $\mathcal{Z} = Z_F$. Thus, the 0-form flavor symmetry group is

$$\mathcal{F}_{\mathfrak{T}} = \frac{Sp(m+1)}{\mathbb{Z}_2}. \tag{158}$$

The projection of $\mathcal{E}$ onto $Z_G$ is $\mathcal{E}' \simeq \mathbb{Z}_2 \times \mathbb{Z}_2$ generated by $\left(0, \frac{1}{2}, 0, \frac{1}{2}, \cdots, 0, \frac{1}{2}, 0\right) \in Z_G$ and $\left(0, 0, \cdots, 0, \frac{1}{2}, \frac{1}{2}\right) \in Z_G$, implying that the projection map is injective. The short exact sequence (13) becomes

$$0 \to \mathbb{Z}_2 \to \mathbb{Z}_2 \times \mathbb{Z}_2 \to \mathbb{Z}_2 \to 0. \tag{159}$$

Since the short exact sequence splits, there is no non-trivial 2-group structure between $\mathcal{O}_{\mathfrak{T}}$ and $\mathcal{F}_{\mathfrak{T}}$.

On the other hand, for $n = 2k + 1$, we find that $\mathcal{E} \simeq \mathbb{Z}_4$ generated by $\left(0, \frac{1}{2}, 0, \frac{1}{2}, \cdots, 0, \frac{1}{2}, \frac{1}{4}, \frac{3}{4}, 1\right) \in Z_G \times Z_F \simeq (\mathbb{R}/\mathbb{Z})^{n+1} \times \mathbb{Z}/2\mathbb{Z}$. From this we see that $\mathcal{O} \simeq \mathbb{Z}_2$ generated by $\left(0, 0, \cdots, 0, \frac{1}{2}, \frac{1}{2}, 0\right) \in Z_G \times Z_F$ which is identified with the 1-form symmetry group $\mathcal{O}_{\mathfrak{T}} \simeq \mathbb{Z}_2$ of $\mathfrak{T}$. We have $\mathcal{Z} = Z_F$. Thus, the 0-form flavor symmetry group is

$$\mathcal{F}_{\mathfrak{T}} = \frac{Sp(m+1)}{\mathbb{Z}_2} = PSp(m+1). \tag{160}$$

The projection of $\mathcal{E}$ onto $Z_G$ is $\mathcal{E}' \simeq \mathbb{Z}_4$ generated by $\left(0, \frac{1}{2}, 0, \frac{1}{2}, \cdots, 0, \frac{1}{2}, \frac{1}{4}, \frac{3}{4}\right) \in Z_G$, implying that the projection map is injective. The short exact sequence (13) becomes

$$0 \to \mathbb{Z}_2 \to \mathbb{Z}_4 \to \mathbb{Z}_2 \to 0. \tag{161}$$

Thus, we learn that $\mathcal{O}_{\mathfrak{T}}$ and $\mathcal{F}_{\mathfrak{T}}$ form a non-trivial 2-group with the Postnikov class being

$$\text{Bock}(\widetilde{v}_2) \in H^3\left(BPSp(m+1), \mathbb{Z}_2\right), \tag{162}$$

where $\widetilde{v}_2 \in H^2(BPSp(m+1), \mathbb{Z}_2)$ is the characteristic class capturing obstruction of lifting $PSp(m+1)$ bundles to $Sp(m+1)$ bundles, and Bock is the Bockstein associated to the above short exact sequence. Note that 2-group structure is non-trivial because the Postnikov class $\text{Bock}(\widetilde{v}_2)$ is a non-trivial element of $H^3(BPSp(m+1), \mathbb{Z}_2)$ for $m+1 = 4k+2$ [54].

$\underline{\textbf{Spin}(m+4) + m\boldsymbol{F}}$ **:** Consider the 5d SCFT $\mathfrak{T}$ which is the UV completion of 5d $\text{Spin}(m+4)$ gauge theory having $m$ hypers in vector irrep. For $m = 2n-1$, the corresponding geometry is

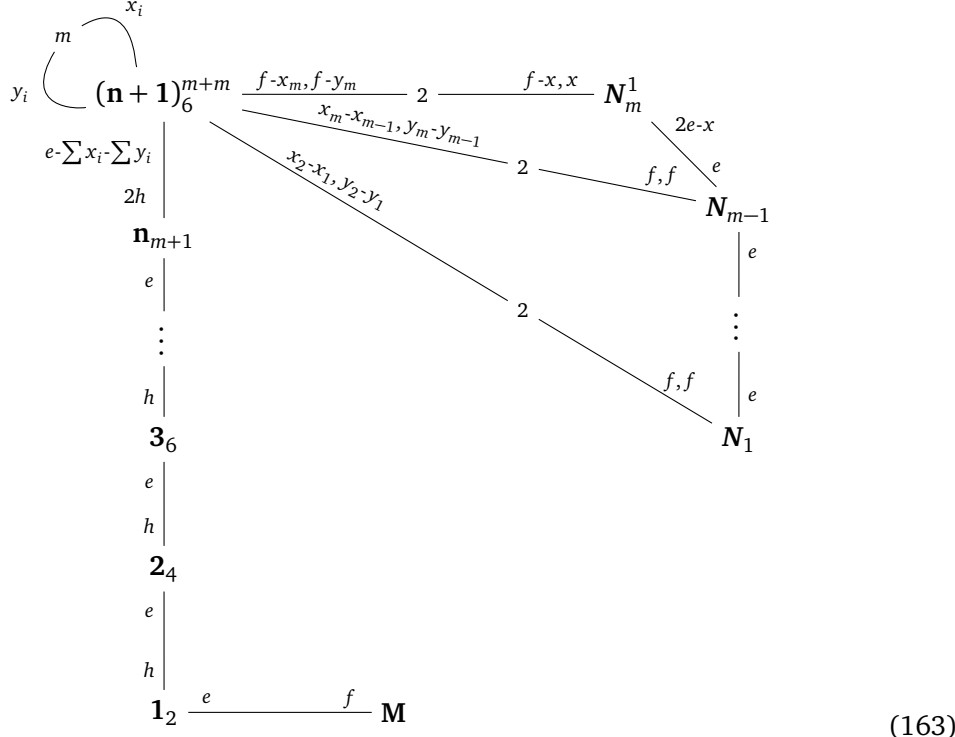

$$\tag{163}$$

$N_i, \mathbf{M}$ describe the $\mathfrak{sp}(m)_N \oplus \mathfrak{su}(2)_M$ flavor symmetry of $\mathfrak{T}$. Thus, $Z_F \simeq (\mathbb{Z}/2\mathbb{Z})_M \times (\mathbb{Z}/2\mathbb{Z})_N$. We find that $\mathcal{E} \simeq \mathbb{Z}_4$ generated by $\left(\frac{1}{2}, \frac{1}{2}, \cdots, \frac{1}{2}, \frac{1}{4}, 1, 0\right) \in Z_G \times Z_F \simeq (\mathbb{R}/\mathbb{Z})^{n+1} \times (\mathbb{Z}/2\mathbb{Z})_M \times (\mathbb{Z}/2\mathbb{Z})_N$. From this we see that $\mathcal{O} = \mathcal{E} \simeq \mathbb{Z}_2$ generated by $\left(0, 0, \cdots, 0, \frac{1}{2}, 0, 0\right) \in Z_G \times Z_F$ which is identified with the 1-form symmetry group $\mathcal{O}_{\mathfrak{T}} \simeq \mathbb{Z}_2$ of $\mathfrak{T}$. We have $\mathcal{Z} \simeq \mathbb{Z}_2$ generated by $(1, 0) \in Z_F$. Thus, the 0-form flavor symmetry group is

$$\mathcal{F}_{\mathfrak{T}} = Sp(m)_N \times SO(3)_M. \tag{164}$$

The projection of $\mathcal{E}$ onto $Z_G$ is $\mathcal{E}' \simeq \mathbb{Z}_4$ generated by $\left(\frac{1}{2}, \frac{1}{2}, \cdots, \frac{1}{2}, \frac{1}{4}\right) \in Z_G$, implying that the projection map is injective. The short exact sequence (13) becomes

$$0 \to \mathbb{Z}_2 \to \mathbb{Z}_4 \to \mathbb{Z}_2^M \to 0, \tag{165}$$

where $\mathbb{Z}_2^M$ is the center of $SU(2)_M$. Thus, we learn that $\mathcal{O}_{\mathfrak{T}}$ and $SO(3)$ subfactor of $\mathcal{F}_{\mathfrak{T}}$ form a non-trivial 2-group with the Postnikov class being

$$\text{Bock}(\widetilde{v}_2) \in H^3\left(BSO(3)_M, \mathbb{Z}_2\right), \tag{166}$$

where $\widetilde{v}_2 \in H^2(BSO(3), \mathbb{Z}_2)$ is the characteristic class capturing obstruction of lifting $SO(3)_M$ bundles to $SU(2)_M$ bundles, and Bock is the Bockstein associated to the above short exact sequence.

For $m = 2n - 2$, the corresponding geometry is

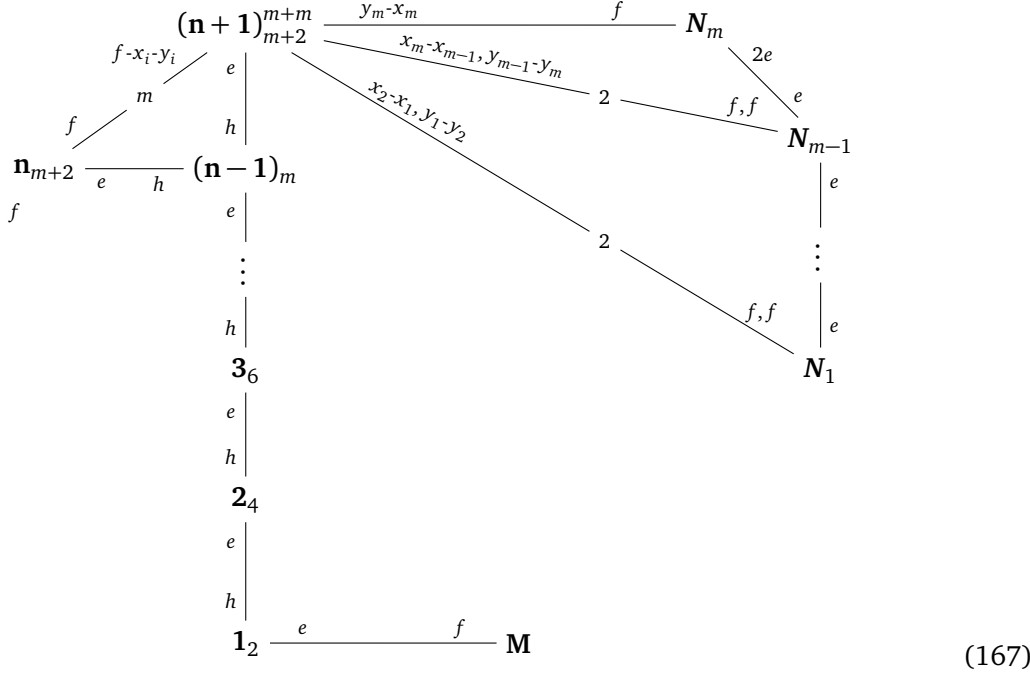

$$(167)$$

$N_i, \mathbf{M}$ describe the $\mathfrak{sp}(m) \oplus \mathfrak{su}(2)$ flavor symmetry of $\mathfrak{T}$. Thus, $Z_F \simeq (\mathbb{Z}/2\mathbb{Z})_M \times (\mathbb{Z}/2\mathbb{Z})_N$.

For $n = 2k$, we find that $\mathcal{E} \simeq \mathbb{Z}_4 \times \mathbb{Z}_2$ generated by $\left(\frac{1}{2}, 0, \frac{1}{2}, 0, \cdots, \frac{1}{2}, 0, \frac{1}{2}, \frac{1}{4}, \frac{3}{4}, 0, 1\right) \in Z_G \times Z_F \simeq (\mathbb{R}/\mathbb{Z})^{n+1} \times (\mathbb{Z}/2\mathbb{Z})_M \times (\mathbb{Z}/2\mathbb{Z})_N$ and $\left(0, \frac{1}{2}, 0, \frac{1}{2}, \cdots, 0, \frac{1}{2}, 0, 1, 1\right) \in Z_G \times Z_F$. From this we see that $\mathcal{O} \simeq \mathbb{Z}_2$ generated by $\left(0, 0, \cdots, 0, \frac{1}{2}, \frac{1}{2}, 0, 0\right) \in Z_G \times Z_F$ which is identified with the 1-form symmetry group $\mathcal{O}_{\mathfrak{T}} \simeq \mathbb{Z}_2$ of $\mathfrak{T}$. We have $\mathcal{Z} = Z_F$. Thus, the 0-form flavor symmetry group is

$$\mathcal{F}_{\mathfrak{T}} = \frac{Sp(m)_N}{\mathbb{Z}_2^N} \times SO(3)_M , \tag{168}$$

where $\mathbb{Z}_2^N$ is the center of $Sp(m)_N$. The projection of $\mathcal{E}$ onto $Z_G$ is $\mathcal{E}' \simeq \mathbb{Z}_4 \times \mathbb{Z}_2$ generated by $\left(\frac{1}{2}, 0, \frac{1}{2}, 0, \cdots, \frac{1}{2}, 0, \frac{1}{2}, \frac{1}{4}, \frac{3}{4}\right) \in Z_G$ and $\left(0, \frac{1}{2}, 0, \frac{1}{2}, \cdots, 0, \frac{1}{2}, 0\right) \in Z_G$, implying that the projection map is injective. The short exact sequence (13) becomes

$$0 \to \mathbb{Z}_2 \to \mathbb{Z}_4 \times \mathbb{Z}_2 \to \mathbb{Z}_2^N \times \mathbb{Z}_2^{N,M} \to 0 , \tag{169}$$

where $\mathbb{Z}_2^{N,M}$ is the diagonal combination of the centers of $Sp(m)_N$ and $SU(2)_M$. The non-trivial part of the above short exact sequence is

$$0 \to \mathbb{Z}_2 \to \mathbb{Z}_4 \to \mathbb{Z}_2^N \to 0 . \tag{170}$$

Thus, we learn that $\mathcal{O}_{\mathfrak{T}}$ and $\mathcal{F}_{\mathfrak{T}}$ form a non-trivial 2-group with the Postnikov class being

$$\mathrm{Bock}(\widetilde{v}_2) \in H^3\left( B\left[ \frac{Sp(m)_N}{\mathbb{Z}_2^N} \times SO(3)_M \right], \mathbb{Z}_2 \right), \tag{171}$$

where $\widetilde{v}_2 \in H^2\left( B\left[ \frac{Sp(m)_N}{\mathbb{Z}_2^N} \times SO(3)_M \right], \mathbb{Z}_2 \right)$ is the characteristic class capturing obstruction of lifting $\frac{Sp(m)_N}{\mathbb{Z}_2^N} \times SO(3)_M$ bundles to $\frac{SU(2)_M \times Sp(m)_N}{\mathbb{Z}_2^{N,M}}$ bundles, and Bock is the Bockstein associated to the above short exact sequence (170). It can be argued that the Postnikov class is non-trivial as follows. We can write $\widetilde{v}_2 = v_2 + w_2$ where $v_2$ captures the obstruction of lifting

$PSp(m)_N \times SO(3)_M$ bundles to $Sp(m)_N \times SO(3)_M$ bundles and $w_2$ captures the obstruction of lifting $PSp(m)_N \times SO(3)_M$ bundles to $PSp(m)_N \times SU(2)_M$ bundles. It is known, for $m = 4k-2$, that Bock acting on $v_2$ and $w_2$ leads to two non-trivial elements $\text{Bock}(v_2)$ and $\text{Bock}(w_2)$ such that $\text{Bock}(v_2) \neq \text{Bock}(w_2)$. Thus $\text{Bock}(\widetilde{v}_2)$ is non-trivial.

On the other hand, for $n = 2k + 1$, we find that $\mathcal{E} \simeq \mathbb{Z}_4 \times \mathbb{Z}_2$ generated by

$$\left(0, \frac{1}{2}, 0, \frac{1}{2}, \cdots, 0, \frac{1}{2}, \frac{1}{4}, \frac{3}{4}, 1, 1\right) \in Z_G \times Z_F \simeq (\mathbb{R}/\mathbb{Z})^{n+1} \times (\mathbb{Z}/2\mathbb{Z})_M \times (\mathbb{Z}/2\mathbb{Z})_N \qquad (172)$$

and $\left(\frac{1}{2}, 0, \frac{1}{2}, 0, \cdots, \frac{1}{2}, 0, 0, 1\right) \in Z_G \times Z_F$. From this we see that $\mathcal{O} \simeq \mathbb{Z}_2$ generated by $\left(0, 0, \cdots, 0, \frac{1}{2}, \frac{1}{2}, 0, 0\right) \in Z_G \times Z_F$ which is identified with the 1-form symmetry group $\mathcal{O}_{\mathfrak{T}} \simeq \mathbb{Z}_2$ of $\mathfrak{T}$. We have $\mathcal{Z} = \mathbb{Z}_F$. Thus, the 0-form flavor symmetry group is

$$\mathcal{F}_{\mathfrak{T}} = \frac{Sp(m)_N}{\mathbb{Z}_2^N} \times SO(3)_M \,, \qquad (173)$$

where $\mathbb{Z}_2^N$ is the center of $Sp(m)_N$. The projection of $\mathcal{E}$ onto $Z_G$ is $\mathcal{E}' \simeq \mathbb{Z}_4 \times \mathbb{Z}_2$ generated by $\left(0, \frac{1}{2}, 0, \frac{1}{2}, \cdots, 0, \frac{1}{2}, \frac{1}{4}, \frac{3}{4}\right) \in Z_G$ and $\left(\frac{1}{2}, 0, \frac{1}{2}, 0, \cdots, \frac{1}{2}, 0\right) \in Z_G$, implying that the projection map is injective. The short exact sequence (13) becomes

$$0 \to \mathbb{Z}_2 \to \mathbb{Z}_4 \times \mathbb{Z}_2 \to \mathbb{Z}_2^N \times \mathbb{Z}_2^M \to 0 \,, \qquad (174)$$

where $\mathbb{Z}_2^M$ is the center of $SU(2)_M$. The non-trivial part of the above short exact sequence is

$$0 \to \mathbb{Z}_2 \to \mathbb{Z}_4 \to \mathbb{Z}_2^M \to 0 \,. \qquad (175)$$

Thus, we learn that $\mathcal{O}_{\mathfrak{T}}$ and $\mathcal{F}_{\mathfrak{T}}$ form a non-trivial 2-group with the Postnikov class being

$$\text{Bock}(\widetilde{v}_2) \in H^3\left(B\left[\frac{Sp(m)_N}{\mathbb{Z}_2^N} \times SO(3)_M\right], \mathbb{Z}_2\right), \qquad (176)$$

where $\widetilde{v}_2 \in H^2\left(B\left[\frac{Sp(m)_N}{\mathbb{Z}_2^N} \times SO(3)_M\right], \mathbb{Z}_2\right)$ is the characteristic class capturing obstruction of lifting $\frac{Sp(m)_N}{\mathbb{Z}_2^N} \times SO(3)_M$ bundles to $\frac{Sp(m)_N}{\mathbb{Z}_2^N} \times SU(2)_M$ bundles, and Bock is the Bockstein associated to the above short exact sequence (175).

**$\underline{\text{Spin}(2n+1) + mF; \; 0 \leq m \leq 2n-4}$ :** Consider the 5d SCFT $\mathfrak{T}$ which is the UV completion of 5d $\text{Spin}(2n+1)$ gauge theory having $0 \leq m \leq 2n-4$ hypers in vector irrep. The non-abelian part of the flavor symmetry is $\mathfrak{f}_{\mathfrak{T}} = \mathfrak{sp}(m)$ which is perturbative. This theory has 1-form symmetry group $\mathcal{O}_{\mathfrak{T}} \simeq \mathbb{Z}_2$. The analysis of section 2.2.3 implies that

$$\mathcal{F}_{\mathfrak{T}} = Sp(m) \qquad (177)$$

and that there is no 2-group structure between $\mathcal{O}_{\mathfrak{T}}$ and $\mathcal{F}_{\mathfrak{T}}$.

**$\underline{\text{Spin}(2n+2) + mF; \; 0 \leq m \leq 2n-3}$ :** Consider the 5d SCFT $\mathfrak{T}$ which is the UV completion of 5d $\text{Spin}(2n+2)$ gauge theory having $0 \leq m \leq 2n-3$ hypers in vector irrep. The non-abelian part of the flavor symmetry is $\mathfrak{f}_{\mathfrak{T}} = \mathfrak{sp}(m)$ which is perturbative. This theory has 1-form symmetry group $\mathcal{O}_{\mathfrak{T}} \simeq \mathbb{Z}_2$.

Let us now apply the analysis of section 2.2.3. For $m$ odd, we find that the 0-form symmetry group is

$$\mathcal{F}_{\mathfrak{T}} = Sp(m) \qquad (178)$$

and that there is no 2-group structure between $\mathcal{O}_{\mathfrak{T}}$ and $\mathcal{F}_{\mathfrak{T}}$.

For $m$ even and $n$ odd, we find that the 0-form symmetry group is

$$\mathcal{F}_{\mathfrak{T}} = PSp(m) \tag{179}$$

and there is no 2-group structure between $\mathcal{O}_{\mathfrak{T}}$ and $\mathcal{F}_{\mathfrak{T}}$.

For $m$ even and $n$ even, we find that the 0-form symmetry group is

$$\mathcal{F}_{\mathfrak{T}} = PSp(m) \tag{180}$$

and there is a potential 2-group structure between $\mathcal{O}_{\mathfrak{T}}$ and $\mathcal{F}_{\mathfrak{T}}$ whose Postnikov class is

$$\mathrm{Bock}(\widetilde{\nu}_2) \in H^3\left(BPSp(m), \mathbb{Z}_2\right), \tag{181}$$

where $\widetilde{\nu}_2 \in H^2\left(BPSp(m), \mathbb{Z}_2\right)$ is the characteristic class capturing obstruction of lifting $PSp(m)$ bundles to $Sp(m)$ bundles, and Bock is the Bockstein associated to the short exact sequence

$$0 \to \mathbb{Z}_2 \to \mathbb{Z}_4 \to \mathbb{Z}_2 \to 0. \tag{182}$$

For $m = 4k + 2$, the Postnikov class is non-trivial [54] and hence we have a non-trivial 2-group structure. For $m = 4k$, we do not know if the Postnikov class is non-trivial. If it is, then the 2-group is non-trivial.

## 4.2 Rank 1

$\underline{SU(2)_0}$ : Consider the 5d SCFT $\mathfrak{T}$ which is the UV completion of 5d pure $SU(2)$ gauge theory with vanishing discrete theta angle. The corresponding geometry is

$$\boldsymbol{N} \xrightarrow{\quad f \qquad\qquad e \quad} \boldsymbol{1}_2 \tag{183}$$

$\boldsymbol{N}$ describes an $\mathfrak{su}(2)$ flavor symmetry of $\mathfrak{T}$. Thus, $Z_F \simeq \mathbb{Z}/2\mathbb{Z}$. We find that $\mathcal{E} \simeq \mathbb{Z}_4$ generated by $\left(\frac{1}{4}, 1\right) \in Z_G \times Z_F \simeq \mathbb{R}/\mathbb{Z} \times \mathbb{Z}/2\mathbb{Z}$. From this, we compute $\mathcal{O} \simeq \mathbb{Z}_2$ generated by $\left(\frac{1}{2}, 0\right) \in Z_G \times Z_F$ which is identified with the 1-form symmetry group $\mathcal{O}_{\mathfrak{T}}$ of $\mathfrak{T}$. We also have $\mathcal{Z} \simeq \mathbb{Z}_2$ generated by $1 \in Z_F$. Thus, the 0-form flavor symmetry group $\mathcal{F}_{\mathfrak{T}}$ of $\mathfrak{T}$ is

$$\mathcal{F}_{\mathfrak{T}} = SO(3). \tag{184}$$

The short exact sequence (13) becomes

$$0 \to \mathbb{Z}_2 \to \mathbb{Z}_4 \to \mathbb{Z}_2 \to 0. \tag{185}$$

The projection of $\mathcal{E}$ onto $Z_G$ is $\mathcal{E}' \simeq \mathbb{Z}_4$ generated by $\frac{1}{4} \in Z_G \simeq \mathbb{R}/\mathbb{Z}$, implying that the projection map is injective. Thus, following the arguments of section 2.1, we find that $\mathcal{F}_{\mathfrak{T}}$ and $\mathcal{O}_{\mathfrak{T}}$ form a non-trivial 2-group with the Postnikov class being

$$\mathrm{Bock}(\widehat{\nu}_2) \in H^3(BSO(3), \mathbb{Z}_2), \tag{186}$$

where $\widehat{\nu}_2 \in H^2(BSO(3), \mathbb{Z}_2)$ is the characteristic class capturing obstruction of lifting $\mathcal{F}_{\mathfrak{T}} = SO(3)$ bundles to $SU(2)$ bundles, and Bock is the Bockstein corresponding to the above short exact sequence (185).

## 4.3  Rank 2

$\underline{SU(3)_6}$:  Consider the 5d SCFT $\mathfrak{T}$ which is the UV completion of 5d pure $SU(3)$ gauge theory with CS level 6. The corresponding geometry is

$$
\begin{array}{c}
\mathbf{1}_7 \xrightarrow[\hspace{2cm}]{e \qquad h+2f} \mathbf{2}_1 \\
f,f \Big| \quad\qquad\qquad e \\
2 \\
f\text{-}x_1\text{-}x_2, x_1\text{-}x_2 \Big| \quad x_2 \\
\mathbf{N}^2
\end{array}
\tag{187}
$$

$N$ describes an $\mathfrak{su}(2)$ flavor symmetry of $\mathfrak{T}$. Thus, $Z_F \simeq \mathbb{Z}/2\mathbb{Z}$. We find that $\mathcal{E} \simeq \mathbb{Z}_3$ generated by $\left(\frac{1}{3}, \frac{2}{3}, 0\right) \in Z_G \times Z_F \simeq (\mathbb{R}/\mathbb{Z})^2 \times \mathbb{Z}/2\mathbb{Z}$. The 1-form symmetry group $\mathcal{O}_\mathfrak{T}$ of $\mathfrak{T}$ is $\mathcal{O}_\mathfrak{T} \simeq \mathcal{E} \simeq \mathbb{Z}_3$. This implies that $\mathcal{Z} = 0$ and thus the 0-form flavor symmetry group $\mathcal{F}_\mathfrak{T}$ of $\mathfrak{T}$ is

$$
\mathcal{F}_\mathfrak{T} = SU(2). \tag{188}
$$

The projection of $\mathcal{E}$ onto $Z_G$ is $\mathcal{E}' \simeq \mathbb{Z}_3$ generated by $\left(\frac{1}{3}, \frac{2}{3}\right) \in Z_G \simeq (\mathbb{R}/\mathbb{Z})^2$, implying that the projection map is injective. Thus, following the arguments of section 2.1, we find that $\mathfrak{T}$ does *not* admit a non-trivial 2-group structure since the extension of $\mathcal{Z}$ by $\mathcal{O}$ is trivial.

$\underline{Sp(2)_0 + 2\mathbf{\Lambda}^2}$:  Consider the 5d SCFT $\mathfrak{T}$ which is the UV completion of 5d $Sp(2)$ gauge theory having 2 hypers in 2-index antisymmetric irrep and vanishing discrete theta angle. The corresponding geometry is

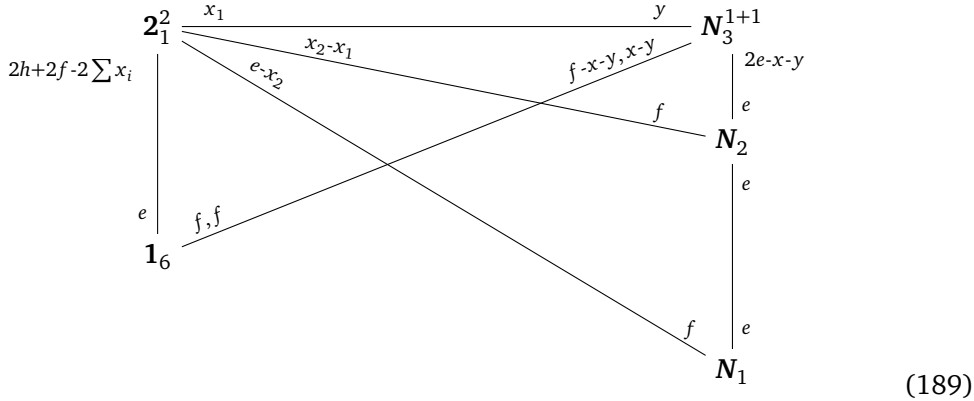

$$
\tag{189}
$$

$N_i$ describe an $\mathfrak{sp}(3)$ flavor symmetry of $\mathfrak{T}$. Thus, $Z_F \simeq \mathbb{Z}/2\mathbb{Z}$. We find that $\mathcal{E} \simeq \mathbb{Z}_2$ generated by $\left(\frac{1}{2}, 0, 0\right) \in Z_G \times Z_F \simeq (\mathbb{R}/\mathbb{Z})^2 \times \mathbb{Z}/2\mathbb{Z}$. The 1-form symmetry group $\mathcal{O}$ of $\mathfrak{T}$ is $\mathcal{O} \simeq \mathcal{E} \simeq \mathbb{Z}_2$. This implies that $\mathcal{Z} = 0$ and thus the 0-form flavor symmetry group $\mathcal{F}_\mathfrak{T}$ of $\mathfrak{T}$ is

$$
\mathcal{F}_\mathfrak{T} = Sp(3). \tag{190}
$$

The projection of $\mathcal{E}$ onto $Z_G$ is $\mathcal{E}' \simeq \mathbb{Z}_2$ generated by $\left(\frac{1}{2}, 0\right) \in Z_G \simeq (\mathbb{R}/\mathbb{Z})^2$, implying that the projection map is injective. Thus, following the arguments of section 2.1, we find that $\mathfrak{T}$ does *not* admit a non-trivial 2-group structure since the extension of $\mathcal{Z}$ by $\mathcal{O}$ is trivial.

## 4.4  Rank 3

$SU(4)_4 + \Lambda^2$ :  Consider the 5d SCFT $\mathfrak{T}$ which is the UV completion of 5d $SU(4)$ gauge theory having a hyper in 2-index antisymmetric irrep and CS level 4. The corresponding geometry is

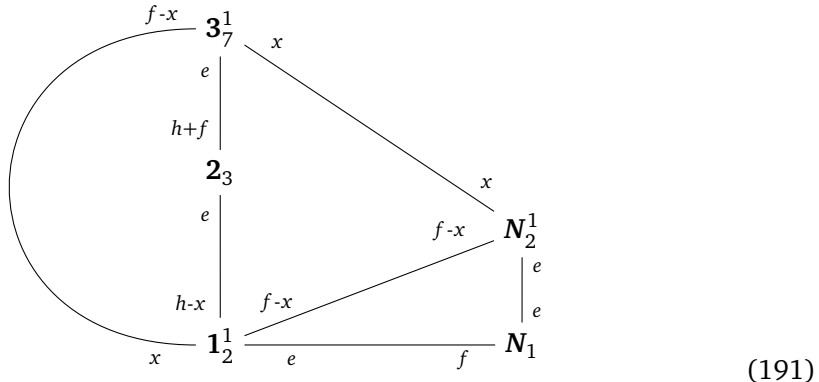

$$\tag{191}$$

$N_i$ describe the $\mathfrak{su}(3)$ flavor symmetry of $\mathfrak{T}$. Thus, $Z_F \simeq \mathbb{Z}/3\mathbb{Z}$. We find that $\mathcal{E} \simeq \mathbb{Z}_3 \times \mathbb{Z}_2$ generated by $\left(0, \frac{2}{3}, \frac{1}{3}, 1\right) \in Z_G \times Z_F \simeq (\mathbb{R}/\mathbb{Z})^3 \times \mathbb{Z}/3\mathbb{Z}$ and $\left(\frac{1}{2}, 0, \frac{1}{2}, 0\right) \in Z_G \times Z_F$. From this we see that $\mathcal{O} \simeq \mathbb{Z}_2$ is the $\mathbb{Z}_2$ subfactor of $\mathcal{E}$, which is identified with the 1-form symmetry group $\mathcal{O}_{\mathfrak{T}} \simeq \mathbb{Z}_2$ of $\mathfrak{T}$. We have $\mathcal{Z} \simeq \mathbb{Z}_3$ generated by $1 \in Z_F$. Thus, the 0-form flavor symmetry group is

$$\mathcal{F}_{\mathfrak{T}} = PSU(3) \,. \tag{192}$$

The projection of $\mathcal{E}$ onto $Z_G$ is $\mathcal{E}' \simeq \mathbb{Z}_3 \times \mathbb{Z}_2$ generated by $\left(0, \frac{2}{3}, \frac{1}{3}\right) \in Z_G$ and $\left(\frac{1}{2}, 0, \frac{1}{2}\right) \in Z_G$, implying that the projection map is injective. The short exact sequence (13) becomes

$$0 \to \mathbb{Z}_3 \to \mathbb{Z}_3 \times \mathbb{Z}_2 \to \mathbb{Z}_2 \to 0 \,, \tag{193}$$

which splits, and hence $\mathfrak{T}$ does not admit a non-trivial 2-group structure.

$SU(4)_0 + \Lambda^2$ :  Consider the 5d SCFT $\mathfrak{T}$ which is the UV completion of 5d $SU(4)$ gauge theory having a hyper in 2-index antisymmetric irrep and CS level 0. This theory has 1-form symmetry group $\mathcal{O}_{\mathfrak{T}} = \mathbb{Z}_2$. The non-abelian part of the flavor symmetry of $\mathfrak{T}$ is $\mathfrak{su}(2)$ which is the symmetry rotating the $\Lambda^2$ hyper in the $SU(4)$ gauge theory description. Thus, we can apply the analysis of section 2.2.3, using which we find that the 0-form flavor symmetry group of $\mathfrak{T}$ is

$$\mathcal{F}_{\mathfrak{T}} = SU(2) \tag{194}$$

and that there is no 2-group between $\mathcal{O}_{\mathfrak{T}}$ and $\mathcal{F}_{\mathfrak{T}}$.

$SU(4)_2 + \Lambda^2$ :  Consider the 5d SCFT $\mathfrak{T}$ which is the UV completion of 5d $SU(4)$ gauge theory having a hyper in 2-index antisymmetric irrep and CS level 2. This theory has 1-form symmetry group $\mathcal{O}_{\mathfrak{T}} = \mathbb{Z}_2$. The non-abelian part of the flavor symmetry of $\mathfrak{T}$ is $\mathfrak{su}(2)$ which is the symmetry rotating the $\Lambda^2$ hyper in the $SU(4)$ gauge theory description. Thus, we can apply the analysis of section 2.2.3, using which we find that the 0-form flavor symmetry group of $\mathfrak{T}$ is

$$\mathcal{F}_{\mathfrak{T}} = SO(3) \tag{195}$$

and there is a 2-group structure between $\mathcal{O}_{\mathfrak{T}}$ and $\mathcal{F}_{\mathfrak{T}}$ whose Postnikov class is

$$\text{Bock}(\widetilde{\nu}_2) \in H^3(BSO(3), \mathbb{Z}_2) \,, \tag{196}$$

where $\widetilde{\nu}_2 \in H^2(BSO(3), \mathbb{Z}_2)$ is the characteristic class capturing obstruction of lifting $SO(3)$ bundles to $SU(2)$ bundles, and Bock is the Bockstein associated to the short exact sequence

$$0 \to \mathbb{Z}_2 \to \mathbb{Z}_4 \to \mathbb{Z}_2 \to 0 \,. \tag{197}$$

$SU(4)_4 + 2\Lambda^2$ : Consider 5d SCFT $\mathfrak{T}$ which is the UV completion of 5d $SU(4)$ gauge theory having 2 hypers in 2-index antisymmetric irrep and CS level 4. The corresponding geometry is

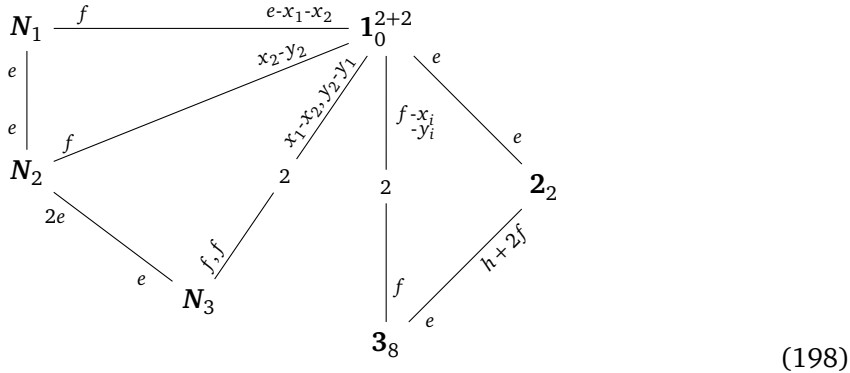

$$\text{(198)}$$

$N_i$ describe an $\mathfrak{so}(7)$ flavor symmetry of $\mathfrak{T}$. Thus, $Z_F \simeq \mathbb{Z}/2\mathbb{Z}$. We find that $\mathcal{E} \simeq \mathbb{Z}_4$ generated by $\left(\frac{3}{4}, \frac{1}{2}, \frac{1}{4}, 1\right) \in Z_G \times Z_F \simeq (\mathbb{R}/\mathbb{Z})^3 \times \mathbb{Z}/2\mathbb{Z}$. From this we see that $\mathcal{O} \simeq \mathbb{Z}_2$ generated by $\left(\frac{1}{2}, 0, \frac{1}{2}, 0\right) \in Z_G \times Z_F$ which is identified with the 1-form symmetry group $\mathcal{O}_{\mathfrak{T}} \simeq \mathbb{Z}_2$ of $\mathfrak{T}$. We have $\mathcal{Z} \simeq \mathbb{Z}_2$ generated by $1 \in Z_F$. Thus, the 0-form flavor symmetry group $\mathcal{F}_{\mathfrak{T}}$ of $\mathfrak{T}$ is

$$\mathcal{F}_{\mathfrak{T}} = SO(7). \tag{199}$$

The projection of $\mathcal{E}$ onto $Z_G$ is $\mathcal{E}' \simeq \mathbb{Z}_4$ generated by $\left(\frac{3}{4}, \frac{1}{2}, \frac{1}{4}\right) \in Z_G$ implying that the projection map is injective. The short exact sequence (13) becomes

$$0 \to \mathbb{Z}_2 \to \mathbb{Z}_4 \to \mathbb{Z}_2 \to 0. \tag{200}$$

Thus, we learn that $\mathcal{O}_{\mathfrak{T}} \simeq \mathbb{Z}_2$ and $\mathcal{F}_{\mathfrak{T}} = SO(7)$ form a non-trivial 2-group with the Postnikov class being

$$\text{Bock}(\widetilde{\nu}_2) \in H^3(BSO(7), \mathbb{Z}_2), \tag{201}$$

where $\widetilde{\nu}_2 \in H^2(BSO(7), \mathbb{Z}_2)$ is the characteristic class capturing obstruction of lifting $SO(7)$ bundles to $\text{Spin}(7)$ bundles, and Bock is the Bockstein associated to the above short exact sequence. Note that the Postnikov class is trivial since $\widetilde{\nu}_2$ is the second Stiefel-Whitney class $w_2$ and then $\text{Bock}(\widetilde{\nu}_2)$ is the third Stiefel-Whitney class $w_3$.

$SU(4)_0 + 2\Lambda^2$ : Consider 5d SCFT $\mathfrak{T}$ which is the UV completion of 5d $SU(4)$ gauge theory having 2 hypers in 2-index antisymmetric irrep and CS level 0. The corresponding geometry is

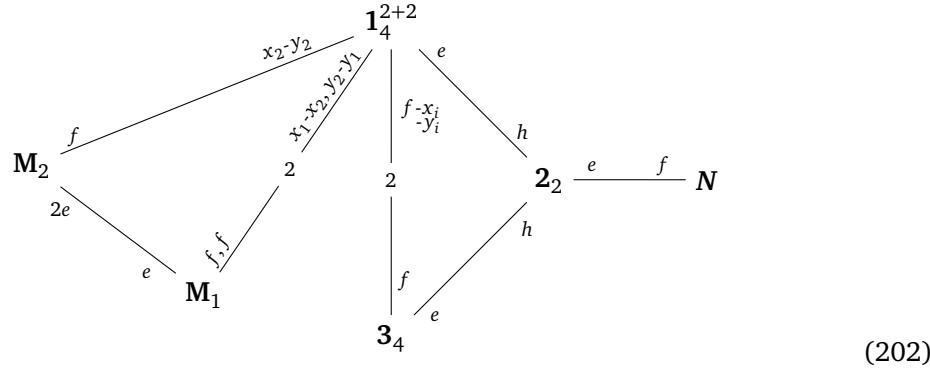

$$\text{(202)}$$

$M_i$ describe an $\mathfrak{sp}(2)$ flavor symmetry of $\mathfrak{T}$ and $N$ describes an $\mathfrak{su}(2)$ flavor symmetry of $\mathfrak{T}$. Thus, $Z_F \simeq (\mathbb{Z}/2\mathbb{Z})_N \times (\mathbb{Z}/2\mathbb{Z})_M$. We find that $\mathcal{E} \simeq \mathbb{Z}_4 \times \mathbb{Z}_2$ generated by $\left(\frac{3}{4}, \frac{1}{2}, \frac{3}{4}, 1, 0\right) \in Z_G \times Z_F \simeq (\mathbb{R}/\mathbb{Z})^3 \times (\mathbb{Z}/2\mathbb{Z})_N \times (\mathbb{Z}/2\mathbb{Z})_M$ and $\left(0, 0, \frac{1}{2}, 1, 1\right) \in Z_G \times Z_F$. From this

we see that $\mathcal{O} \simeq \mathbb{Z}_2$ generated by $\left(\frac{1}{2}, 0, \frac{1}{2}, 0, 0\right) \in Z_G \times Z_F$ which is identified with the 1-form symmetry group $\mathcal{O}_{\mathfrak{T}} \simeq \mathbb{Z}_2$ of $\mathfrak{T}$. We have $\mathcal{Z} = Z_F$. Thus, the 0-form flavor symmetry group $\mathcal{F}_{\mathfrak{T}}$ of $\mathfrak{T}$ is

$$\mathcal{F}_{\mathfrak{T}} = \frac{Sp(2)_M}{\mathbb{Z}_2} \times SO(3)_N \,. \tag{203}$$

The projection of $\mathcal{E}$ onto $Z_G$ is $\mathcal{E}' \simeq \mathbb{Z}_4 \times \mathbb{Z}_2$ generated by $\left(\frac{3}{4}, \frac{1}{2}, \frac{3}{4}\right) \in Z_G$ and $\left(0, 0, \frac{1}{2}\right) \in Z_G$, implying that the projection map is injective. The short exact sequence (13) becomes

$$0 \to \mathbb{Z}_2 \to \mathbb{Z}_4 \times \mathbb{Z}_2 \to \mathbb{Z}_2^N \times \mathbb{Z}_2^{N,M} \to 0 \,, \tag{204}$$

where $\mathbb{Z}_2^N$ is the center of $SU(2)_N$, and $\mathbb{Z}_2^{N,M}$ is the diagonal combination of the centers of $SU(2)_N$ and $Sp(2)_M$. The non-trivial part of the above short exact sequence is

$$0 \to \mathbb{Z}_2 \to \mathbb{Z}_4 \to \mathbb{Z}_2^N \to 0 \,. \tag{205}$$

Thus, we learn that $\mathcal{O}_{\mathfrak{T}}$ and $\mathcal{F}_{\mathfrak{T}}$ form a potentially non-trivial 2-group with the Postnikov class being

$$\text{Bock}(\widetilde{\nu}_2) \in H^3\left(B\left[\frac{Sp(2)_M}{\mathbb{Z}_2} \times SO(3)_N\right], \mathbb{Z}_2\right) \,, \tag{206}$$

where $\widetilde{\nu}_2 \in H^2\left(B\left[\frac{Sp(2)_M}{\mathbb{Z}_2} \times SO(3)_N\right], \mathbb{Z}_2\right)$ is the characteristic class capturing obstruction of lifting $\frac{Sp(2)_M}{\mathbb{Z}_2} \times SO(3)_N$ bundles to $\frac{SU(2)_N \times Sp(2)_M}{\mathbb{Z}_2^{N,M}}$ bundles, and Bock is the Bockstein associated to the above short exact sequence (205). The 2-group is non-trivial if the Postnikov class $\text{Bock}(\widetilde{\nu}_2)$ is a non-trivial element of $H^3\left(B\left[\frac{Sp(2)_M}{\mathbb{Z}_2} \times SO(3)_N\right], \mathbb{Z}_2\right)$.

$\underline{SU(4)_2 + 2\Lambda^2}$ : Consider 5d SCFT $\mathfrak{T}$ which is the UV completion of 5d $SU(4)$ gauge theory having 2 hypers in 2-index antisymmetric irrep and CS level 2. This theory has 1-form symmetry group $\mathcal{O}_{\mathfrak{T}} = \mathbb{Z}_2$. The non-abelian part of the flavor symmetry of $\mathfrak{T}$ is $\mathfrak{sp}(2)$ which is the symmetry rotating the two $\Lambda^2$ hypers in the $SU(4)$ gauge theory description. Thus, we can apply the analysis of section 2.2.3, using which we find that the 0-form flavor symmetry group of $\mathfrak{T}$ is

$$\mathcal{F}_{\mathfrak{T}} = Sp(2) \tag{207}$$

and that there is no 2-group between $\mathcal{O}_{\mathfrak{T}}$ and $\mathcal{F}_{\mathfrak{T}}$.

$\underline{SU(4)_0 + 3\Lambda^2}$ : Consider the 5d SCFT $\mathfrak{T}$ which is the UV completion of 5d $SU(4)$ gauge theory having 3 hypers in 2-index antisymmetric irrep and vanishing CS level. The corresponding geometry is

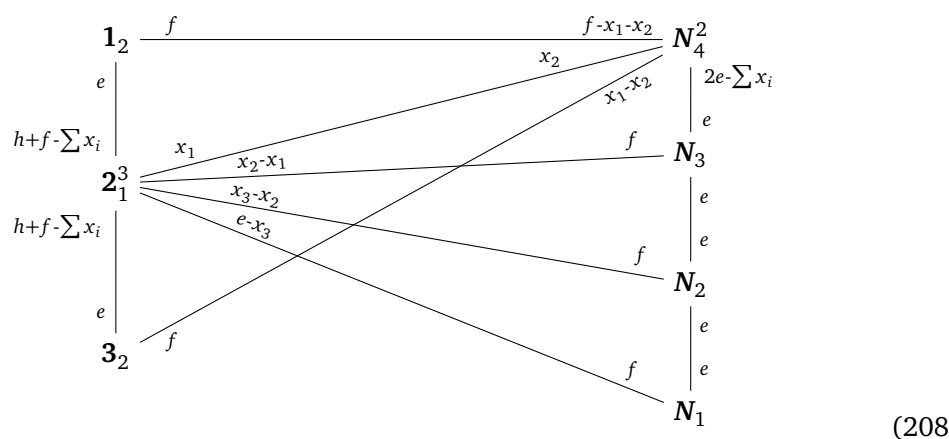

$$\tag{208}$$

$N_i$ describe an $\mathfrak{sp}(4)$ flavor symmetry of $\mathfrak{T}$. Thus, $Z_F \simeq \mathbb{Z}/2\mathbb{Z}$. We find that $\mathcal{E} \simeq \mathbb{Z}_2 \times \mathbb{Z}_2$ generated by $\left(\frac{1}{2}, 0, 0, 1\right) \in Z_G \times Z_F \simeq (\mathbb{R}/\mathbb{Z})^3 \times \mathbb{Z}/2\mathbb{Z}$ and $\left(0, 0, \frac{1}{2}, 1\right) \in Z_G \times Z_F$. From this we compute that $\mathcal{O} \simeq \mathbb{Z}_2$ generated by $\left(\frac{1}{2}, 0, \frac{1}{2}, 0\right) \in Z_G \times Z_F$ which is identified with the 1-form symmetry group $\mathcal{O}_{\mathfrak{T}} \simeq \mathbb{Z}_2$ of $\mathfrak{T}$. We also have $\mathcal{Z} \simeq \mathbb{Z}_2$ generated by $1 \in Z_F$. Thus the 0-form flavor symmetry group $\mathcal{F}_{\mathfrak{T}}$ of $\mathfrak{T}$ is

$$\mathcal{F}_{\mathfrak{T}} = \frac{Sp(4)}{\mathbb{Z}_2}. \tag{209}$$

The projection of $\mathcal{E}$ onto $Z_G$ is $\mathcal{E}' \simeq \mathbb{Z}_2 \times \mathbb{Z}_2$ generated by $\left(\frac{1}{2}, 0, 0\right) \in Z_G$ and $\left(0, 0, \frac{1}{2}\right) \in Z_G$, implying that the projection map is injective. The short exact sequence (13) becomes

$$0 \to \mathbb{Z}_2 \to \mathbb{Z}_2 \times \mathbb{Z}_2 \to \mathbb{Z}_2 \to 0, \tag{210}$$

which splits, and hence $\mathfrak{T}$ does not admit a non-trivial 2-group structure.

$\underline{SU(4)_2 + 3\Lambda^2:}$ Consider the 5d SCFT $\mathfrak{T}$ which is the UV completion of 5d $SU(4)$ gauge theory having 3 hypers in 2-index antisymmetric irrep and CS level 2. The corresponding geometry is

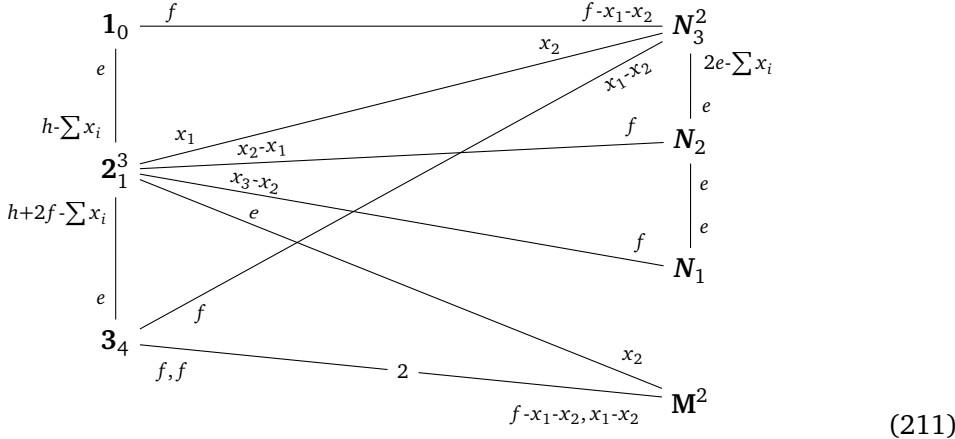

$$\tag{211}$$

$N_i$ describe an $\mathfrak{sp}(3)$ flavor symmetry of $\mathfrak{T}$ and $\mathbf{M}$ describes an $\mathfrak{su}(2)$ flavor symmetry of $\mathfrak{T}$. Thus, $Z_F \simeq (\mathbb{Z}/2\mathbb{Z})_N \times (\mathbb{Z}/2\mathbb{Z})_M$. We find that $\mathcal{E} \simeq \mathbb{Z}_4$ generated by $\left(\frac{1}{4}, \frac{1}{2}, \frac{3}{4}, 1, 0\right) \in Z_G \times Z_F \simeq (\mathbb{R}/\mathbb{Z})^3 \times (\mathbb{Z}/2\mathbb{Z})_N \times (\mathbb{Z}/2\mathbb{Z})_M$. From this we compute that $\mathcal{O} \simeq \mathbb{Z}_2$ generated by $\left(\frac{1}{2}, 0, \frac{1}{2}, 0, 0\right) \in Z_G \times Z_F$ which is identified with the 1-form symmetry group $\mathcal{O}_{\mathfrak{T}} \simeq \mathbb{Z}_2$ of $\mathfrak{T}$. We also have $\mathcal{Z} \simeq \mathbb{Z}_2$ generated by $(1, 0) \in Z_F$. Thus the 0-form flavor symmetry group $\mathcal{F}_{\mathfrak{T}}$ of $\mathfrak{T}$ is

$$\mathcal{F}_{\mathfrak{T}} = \frac{Sp(3)}{\mathbb{Z}_2} \times SU(2). \tag{212}$$

The projection of $\mathcal{E}$ onto $Z_G$ is $\mathcal{E}' \simeq \mathbb{Z}_4$ generated by $\left(\frac{1}{4}, \frac{1}{2}, \frac{3}{4}\right) \in Z_G$, implying that the projection map is injective. The short exact sequence (13) becomes

$$0 \to \mathbb{Z}_2 \to \mathbb{Z}_4 \to \mathbb{Z}_2 \to 0. \tag{213}$$

Thus, the 1-form symmetry group $\mathcal{O}_{\mathfrak{T}} \simeq \mathbb{Z}_2$ and $Sp(3)/\mathbb{Z}_2$ part of 0-form symmetry group $\mathcal{F}_{\mathfrak{T}}$ form a potentially non-trivial 2-group with the Postnikov class being

$$\text{Bock}(\widetilde{\nu}_2) \in H^3\left(B\frac{Sp(3)}{\mathbb{Z}_2}, \mathbb{Z}_2\right), \tag{214}$$

where $\widetilde{\nu}_2 \in H^2\left(B\frac{Sp(3)}{\mathbb{Z}_2}, \mathbb{Z}_2\right)$ is the characteristic class capturing obstruction of lifting $Sp(3)/\mathbb{Z}_2$ bundles to $Sp(3)$ bundles, and Bock is the Bockstein associated to the above short exact sequence. The 2-group is non-trivial if the Postnikov class $\text{Bock}(\widetilde{\nu}_2)$ is a non-trivial element of $H^3\left(B\frac{Sp(3)}{\mathbb{Z}_2}, \mathbb{Z}_2\right)$.

Note that this 2-group is perturbative in the sense that it is already visible at the level of the $SU(4)_2 + 3\Lambda^2$ gauge theory. The instantonic non-perturbative part of the flavor symmetry group is $SU(2)$ and does not contribute to the 2-group structure.

$\underline{SU(4)_4 + 3\Lambda^2}$ :  Consider the 5d SCFT $\mathfrak{T}$ which is the UV completion of 5d $SU(4)$ gauge theory having 3 hypers in 2-index antisymmetric irrep and CS level 4. It is known that this theory has a 1-form symmetry group $\mathcal{O}_{\mathfrak{T}} \simeq \mathbb{Z}_2$ and 0-form flavor symmetry algebra $\mathfrak{f}_4$. Since $F_4$ does not have center, the 0-form flavor symmetry group $\mathcal{F}_{\mathfrak{T}}$ must be

$$\mathcal{F}_{\mathfrak{T}} = F_4 \,. \tag{215}$$

Furthermore, there cannot be a non-trivial 2-group structure between $\mathcal{F}_{\mathfrak{T}}$ and $\mathcal{O}_{\mathfrak{T}}$.

$\underline{\mathbf{Spin}(7) + 3F}$ :  Consider the 5d SCFT $\mathfrak{T}$ which is the UV completion of 5d Spin(7) gauge theory having 3 hypers in vector irrep. The corresponding geometry is

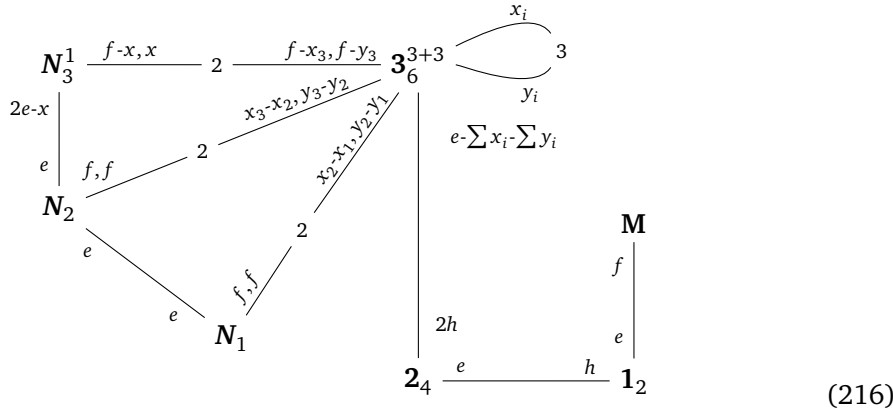

$$\tag{216}$$

$N_i$ describe an $\mathfrak{sp}(3)$ flavor symmetry of $\mathfrak{T}$ and $\mathbf{M}$ describes an $\mathfrak{su}(2)$ flavor symmetry of $\mathfrak{T}$. Thus, $Z_F \simeq (\mathbb{Z}/2\mathbb{Z})_N \times (\mathbb{Z}/2\mathbb{Z})_M$. We find that $\mathcal{E} \simeq \mathbb{Z}_4$ generated by $\left(\frac{1}{2}, \frac{1}{2}, \frac{1}{4}, 0, 1\right) \in Z_G \times Z_F \simeq (\mathbb{R}/\mathbb{Z})^3 \times (\mathbb{Z}/2\mathbb{Z})_N \times (\mathbb{Z}/2\mathbb{Z})_M$. From this we compute that $\mathcal{O} \simeq \mathbb{Z}_2$ generated by $\left(0, 0, \frac{1}{2}, 0, 0\right) \in Z_G \times Z_F$ which is identified with the 1-form symmetry group $\mathcal{O}_{\mathfrak{T}} \simeq \mathbb{Z}_2$ of $\mathfrak{T}$. We also have $\mathcal{Z} \simeq \mathbb{Z}_2$ generated by $(0, 1) \in Z_F$. Thus the 0-form flavor symmetry group $\mathcal{F}_{\mathfrak{T}}$ of $\mathfrak{T}$ is

$$\mathcal{F}_{\mathfrak{T}} = Sp(3) \times SO(3) \,. \tag{217}$$

The projection of $\mathcal{E}$ onto $Z_G$ is $\mathcal{E}' \simeq \mathbb{Z}_4$ generated by $\left(\frac{1}{2}, \frac{1}{2}, \frac{1}{4}\right) \in Z_G$, implying that the projection map is injective. The short exact sequence (13) becomes

$$0 \to \mathbb{Z}_2 \to \mathbb{Z}_4 \to \mathbb{Z}_2 \to 0 \,. \tag{218}$$

Thus, the 1-form symmetry group $\mathcal{O}_{\mathfrak{T}} \simeq \mathbb{Z}_2$ and $SO(3)$ part of 0-form symmetry group $\mathcal{F}_{\mathfrak{T}}$ form a non-trivial 2-group with the Postnikov class being

$$\text{Bock}(\widetilde{\nu}_2) \in H^3\left(BSO(3), \mathbb{Z}_2\right) \,, \tag{219}$$

where $\widetilde{\nu}_2 \in H^2\left(BSO(3), \mathbb{Z}_2\right)$ is the characteristic class capturing obstruction of lifting $SO(3)$ bundles to $SU(2)$ bundles, and Bock is the Bockstein associated to the above short exact sequence.

**Spin**$(7) + nF$; $0 \leq n \leq 4$; $n \neq 3$ : For these cases, the non-abelian part of the 0-form symmetry algebra is $\mathfrak{sp}(n)$ and it is purely perturbative. So we can follow the analysis of section 2.2.3. The center $\mathbb{Z}_2$ of the simply connected group $Sp(n)$ associated to the non-abelian part of flavor algebra acts non-trivially on the perturbative matter. Thus, we must have

$$\mathcal{F}_{\mathfrak{T}} = Sp(n). \tag{220}$$

The 1-form symmetry group of $\mathfrak{T}$ is $\mathcal{O}_{\mathfrak{T}} \simeq \mathbb{Z}_2$. These facts imply that we must have $\mathcal{E} = \mathcal{O} \simeq \mathbb{Z}_2 \simeq \mathcal{E}'$ and $\mathcal{Z} = 0$. Thus, following section 2.1, there is no non-trivial 2-group structure between $\mathcal{F}_{\mathfrak{T}}$ and $\mathcal{O}_{\mathfrak{T}}$.

### 4.5 Rank 5

$\underline{SU(6)_{\frac{15}{2}} + \frac{1}{2}\Lambda^3}$ : Consider the 5d SCFT $\mathfrak{T}$ which is the UV completion of 5d $SU(6)$ gauge theory having a half-hyper in 3-index antisymmetric irrep and CS level $\frac{15}{2}$. The corresponding geometry is

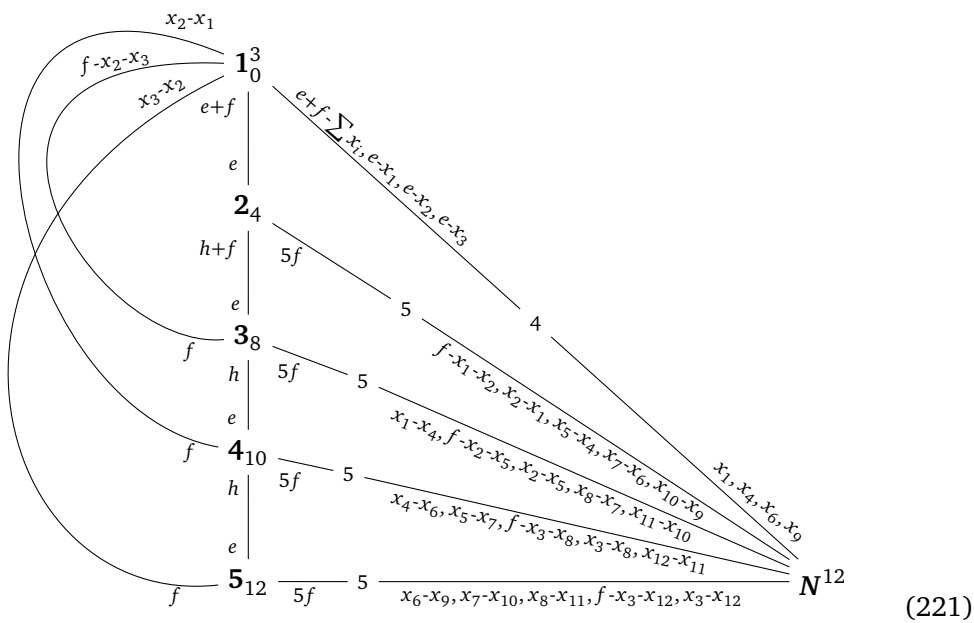

$$\tag{221}$$

$N$ describes an $\mathfrak{su}(2)$ flavor symmetry of $\mathfrak{T}$. Thus, $Z_F \simeq \mathbb{Z}/2\mathbb{Z}$. We find that $\mathcal{E} \simeq \mathbb{Z}_3$ generated by $\left(\frac{2}{3}, \frac{1}{3}, 0, \frac{2}{3}, \frac{1}{3}, 0\right) \in Z_G \times Z_F \simeq (\mathbb{R}/\mathbb{Z})^5 \times \mathbb{Z}/2\mathbb{Z}$. From this we see that $\mathcal{O} = \mathcal{E} \simeq \mathbb{Z}_3$ which is identified with the 1-form symmetry group $\mathcal{O}_{\mathfrak{T}} \simeq \mathbb{Z}_3$ of $\mathfrak{T}$. We have $\mathcal{Z} = 0$, and thus the 0-form flavor symmetry group $\mathcal{F}_{\mathfrak{T}}$ of $\mathfrak{T}$ is

$$\mathcal{F}_{\mathfrak{T}} = SU(2). \tag{222}$$

The projection of $\mathcal{E}$ onto $Z_G$ is $\mathcal{E}' \simeq \mathbb{Z}_3$ generated by $\left(\frac{2}{3}, \frac{1}{3}, 0, \frac{2}{3}, \frac{1}{3}\right) \in Z_G$, implying that the projection map is injective. Thus, following section 2.1, there is no non-trivial 2-group structure between $\mathcal{F}_{\mathfrak{T}}$ and $\mathcal{O}_{\mathfrak{T}}$.

$\underline{SU(6)_3 + \Lambda^3}$ : Consider the 5d SCFT $\mathfrak{T}$ which is the UV completion of 5d $SU(6)$ gauge theory having a full hyper in 3-index antisymmetric irrep and CS level 3. The corresponding geometry

is

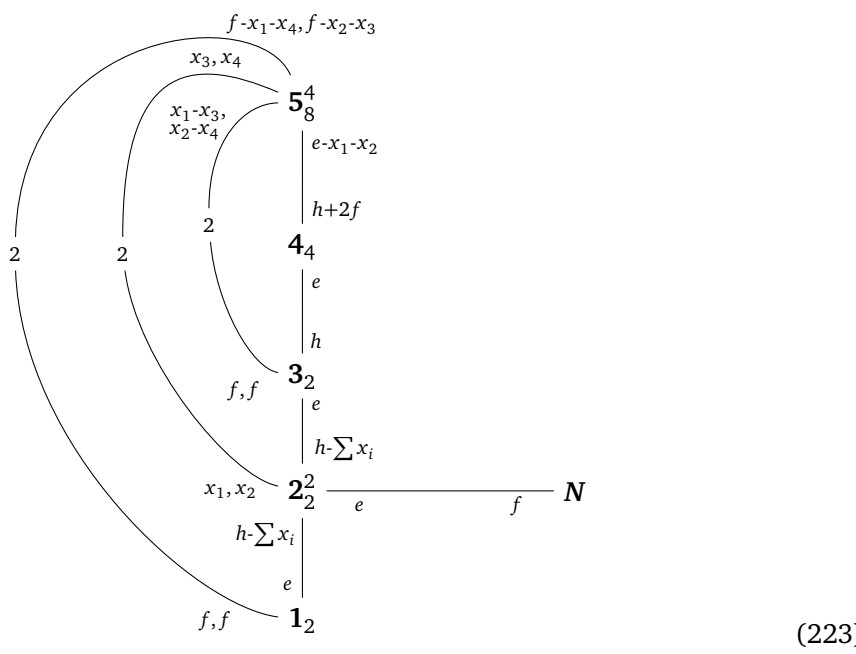

(223)

$N$ describes an $\mathfrak{su}(2)$ flavor symmetry of $\mathfrak{T}$. Thus, $Z_F \simeq \mathbb{Z}/2\mathbb{Z}$. We find that $\mathcal{E} \simeq \mathbb{Z}_3 \times \mathbb{Z}_2$ generated by $\left(\frac{1}{3}, \frac{2}{3}, 0, \frac{1}{3}, \frac{2}{3}, 0\right) \in Z_G \times Z_F \simeq (\mathbb{R}/\mathbb{Z})^5 \times \mathbb{Z}/2\mathbb{Z}$ and $\left(0, 0, \frac{1}{2}, 0, \frac{1}{2}, 1\right) \in Z_G \times Z_F$. From this we see that $\mathcal{O} \simeq \mathbb{Z}_3$ is the first subfactor in $\mathcal{E}$, which is identified with the 1-form symmetry group $\mathcal{O}_{\mathfrak{T}} \simeq \mathbb{Z}_3$ of $\mathfrak{T}$. We have $\mathcal{Z} \simeq \mathbb{Z}_2$ generated by $1 \in Z_F$. Thus, non-abelian part of the 0-form flavor symmetry group $\mathcal{F}_{\mathfrak{T}}$ of $\mathfrak{T}$ is

$$\mathcal{F}_{\mathfrak{T}} = SO(3). \tag{224}$$

The projection of $\mathcal{E}$ onto $Z_G$ is $\mathcal{E}' \simeq \mathbb{Z}_3 \times \mathbb{Z}_2$ generated by $\left(\frac{1}{3}, \frac{2}{3}, 0, \frac{1}{3}, \frac{2}{3}\right) \in Z_G$ and $\left(0, 0, \frac{1}{2}, 0, \frac{1}{2}\right) \in Z_G$, implying that the projection map is injective. The short exact sequence (13) becomes

$$0 \to \mathbb{Z}_3 \to \mathbb{Z}_3 \times \mathbb{Z}_2 \to \mathbb{Z}_2 \to 0, \tag{225}$$

which splits, and hence $\mathfrak{T}$ does not admit a non-trivial 2-group structure.

$\underline{SU(6)_6 + \Lambda^3}$ :  Consider the 5d SCFT $\mathfrak{T}$ which is the UV completion of 5d $SU(6)$ gauge theory having a full hyper in 3-index antisymmetric irrep and CS level 6. We have $\mathcal{O}_{\mathfrak{T}} \simeq \mathbb{Z}_3$ and

$$\mathcal{F}_{\mathfrak{T}} = G_2, \tag{226}$$

which has trivial center. Thus, there can be no 2-group structure between $\mathcal{O}_{\mathfrak{T}}$ and $\mathcal{F}_{\mathfrak{T}}$.

$\underline{SU(6)_{\frac{3}{2}} + \frac{3}{2}\Lambda^3}$ :  Consider the 5d SCFT $\mathfrak{T}$ which is the UV completion of 5d $SU(6)$ gauge theory having 3 half-hypers in 3-index antisymmetric irrep and CS level $\frac{3}{2}$. The corresponding

geometry is

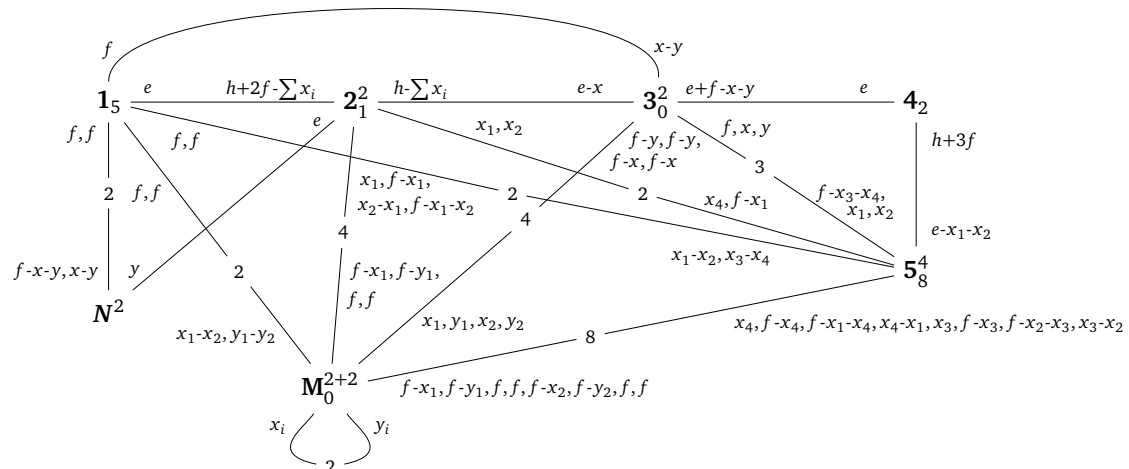

(227)

$N$ and $M$ describe $\mathfrak{su}(2) \oplus \mathfrak{su}(2)$ flavor symmetry of $\mathfrak{T}$. Thus, $Z_F \simeq (\mathbb{Z}/2\mathbb{Z})_N \times (\mathbb{Z}/2\mathbb{Z})_M$. We find that $\mathcal{E} \simeq \mathbb{Z}_3 \times \mathbb{Z}_2$ generated by $\left(\frac{1}{3}, \frac{2}{3}, 0, \frac{1}{3}, \frac{2}{3}, 0, 0\right) \in Z_G \times Z_F \simeq (\mathbb{R}/\mathbb{Z})^5 \times (\mathbb{Z}/2\mathbb{Z})_N \times (\mathbb{Z}/2\mathbb{Z})_M$ and $(0, 0, 0, 0, 0, 0, 1) \in Z_G \times Z_F$. From this we see that $\mathcal{O} \simeq \mathbb{Z}_3$ is the first subfactor in $\mathcal{E}$, which is identified with the 1-form symmetry group $\mathcal{O}_{\mathfrak{T}} \simeq \mathbb{Z}_3$ of $\mathfrak{T}$. We have $\mathcal{Z} \simeq \mathbb{Z}_2$ generated by $(0, 1) \in Z_F$. Thus the 0-form flavor symmetry group $\mathcal{F}_{\mathfrak{T}}$ of $\mathfrak{T}$ is

$$\mathcal{F}_{\mathfrak{T}} = SU(2)_N \times SO(3)_M, \tag{228}$$

where $SO(3)_M$ is the flavor symmetry rotating the 3 half-hypers, and $SU(2)_N$ is the instantonic symmetry. The projection of $\mathcal{E}$ onto $Z_G$ is $\mathcal{E}' \simeq \mathbb{Z}_3$ generated by $\left(\frac{1}{3}, \frac{2}{3}, 0, \frac{1}{3}, \frac{2}{3}\right) \in Z_G$, implying that the projection map is *not* injective. However, let us define $\mathcal{E}_1 \simeq \mathcal{Z}_3$ to be the first subfactor of $\mathcal{E}$ and $\mathcal{E}_2 \simeq \mathbb{Z}_2$ to be the second subfactor of $\mathcal{E}$. Then $\mathcal{O} \subseteq \mathcal{E}_1$ and the elements of $\mathcal{E}_2$ are of the form $(0, *) \in Z_G \times Z_F$. Also, the projection of $\mathcal{E}_1$ onto $Z_G$ is $\mathcal{E}'_1 = \mathcal{E}' \simeq \mathbb{Z}_3$, implying that the projection map $\mathcal{E}_1 \to \mathcal{E}'_1$ is injective. Now, following section 2.1, we learn that $\mathfrak{T}$ does not admit a non-trivial 2-group structure.

## 4.6 Rank 6

**Spin**$(12) + 2S$ : Consider the 5d SCFT $\mathfrak{T}$ which is the UV completion of 5d Spin(12) gauge theory having 2 hypers in spinor irrep. The corresponding geometry is

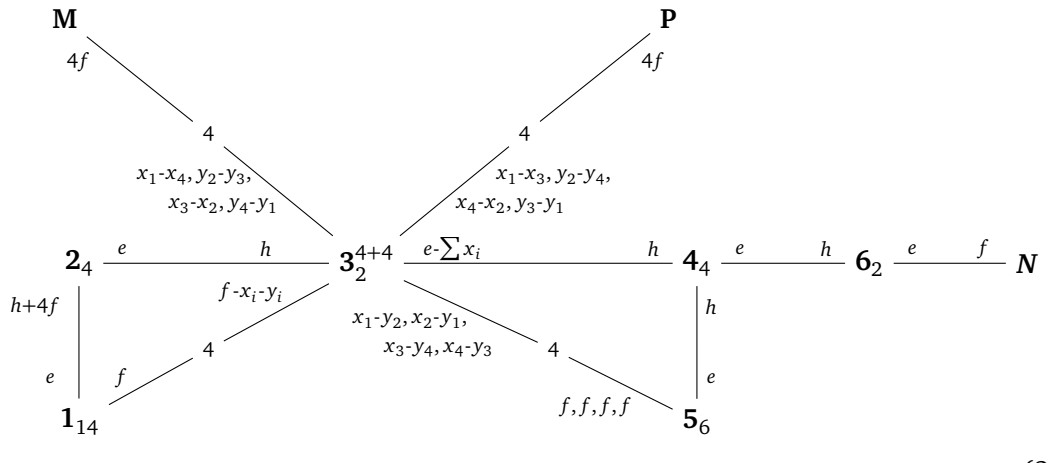

(229)

$\mathbf{M}, \mathbf{N}, \mathbf{P}$ describe three $\mathfrak{su}(2)$ flavor symmetries of $\mathfrak{T}$. Thus, $Z_F \simeq (\mathbb{Z}/2\mathbb{Z})_M \times (\mathbb{Z}/2\mathbb{Z})_N \times (\mathbb{Z}/2\mathbb{Z})_P$. We find that $\mathcal{E} \simeq \mathbb{Z}_4 \times \mathbb{Z}_2 \times \mathbb{Z}_2$ generated by $\left(\frac{3}{4}, \frac{1}{2}, \frac{1}{4}, 0, \frac{1}{2}, \frac{1}{4}, 0, 1, 0\right) \in Z_G \times Z_F \simeq (\mathbb{R}/\mathbb{Z})^6 \times (\mathbb{Z}/2\mathbb{Z})_M$ $\times (\mathbb{Z}/2\mathbb{Z})_N \times (\mathbb{Z}/2\mathbb{Z})_P$, $\left(\frac{1}{2}, 0, \frac{1}{2}, 0, \frac{1}{2}, 0, 1, 0, 0\right) \in Z_G \times Z_F$ and $(0, 0, 0, 0, 0, 0, 1, 0, 1) \in Z_G \times Z_F$. From this we see that $\mathcal{O} \simeq \mathbb{Z}_2$ generated by $\left(\frac{1}{2}, 0, \frac{1}{2}, 0, 0, \frac{1}{2}, 0, 0, 0\right) \in Z_G \times Z_F$ which is identified with the 1-form symmetry group $\mathcal{O}_{\mathfrak{T}} \simeq \mathbb{Z}_2$ of $\mathfrak{T}$. We have $\mathcal{Z} \simeq \mathbb{Z}_2^3$ generated by $(1, 0, 0) \in Z_F$, $(0, 1, 0) \in Z_F$ and $(0, 0, 1) \in Z_F$. Thus, non-abelian part of the 0-form flavor symmetry group $\mathcal{F}_{\mathfrak{T}}$ of $\mathfrak{T}$ is

$$\mathcal{F}_{\mathfrak{T}} = SO(3)_M \times SO(3)_N \times SO(3)_P. \tag{230}$$

Here $SO(3)_M, SO(3)_P$ can be seen perturbatively, but $SO(3)_N$ is instantonic. The projection of $\mathcal{E}$ onto $Z_G$ is not injective, but we can define $\mathcal{E}_1 \simeq \mathbb{Z}_4 \times \mathbb{Z}_2$ as the subgroup of $\mathcal{E}$ generated by its first two subfactors, and $\mathcal{E}_2 \simeq \mathbb{Z}_2$ as the subgroup of $\mathcal{E}$ generated by its third subfactor. Then $\mathcal{O} \subseteq \mathcal{E}_1$, the elements of $\mathcal{E}_2$ are of the form $(0, *) \in Z_G \times Z_F$, and the projection of $\mathcal{E}_1$ onto $Z_G$ is $\mathcal{E}_1' \simeq \mathbb{Z}_4 \times \mathbb{Z}_2$ generated by $\left(\frac{3}{4}, \frac{1}{2}, \frac{1}{4}, 0, \frac{1}{2}, \frac{1}{4}\right) \in Z_G$ and $\left(\frac{1}{2}, 0, \frac{1}{2}, 0, \frac{1}{2}, 0\right) \in Z_G$, implying that the projection map $\mathcal{E}_1 \to \mathcal{E}_1'$ is injective. The short exact sequence (22) becomes

$$0 \to \mathbb{Z}_2 \to \mathbb{Z}_4 \times \mathbb{Z}_2 \to \mathbb{Z}_2 \times \mathbb{Z}_2 \to 0, \tag{231}$$

whose non-trivial part is only

$$0 \to \mathbb{Z}_2 \to \mathbb{Z}_4 \to \mathbb{Z}_2 \to 0, \tag{232}$$

where the first $\mathbb{Z}_2$ is the 1-form symmetry, the $\mathbb{Z}_4$ is a sub-factor of $\mathcal{E}$ and the second $\mathbb{Z}_2$ is $(\mathbb{Z}/2\mathbb{Z})_N$. Thus, we learn that $\mathcal{O}_{\mathfrak{T}}$ and $\mathcal{F}_{\mathfrak{T}}$ form a non-trivial 2-group with the Postnikov class being

$$\text{Bock}(\widetilde{\nu}_2) \in H^3\left(B[SO(3)_M \times SO(3)_N \times SO(3)_P], \mathbb{Z}_2\right), \tag{233}$$

where $\widetilde{\nu}_2 \in H^2\left(B[SO(3)_M \times SO(3)_N \times SO(3)_P], \mathbb{Z}_2\right)$ is the characteristic class capturing obstruction of lifting $SO(3)_M \times SO(3)_N \times SO(3)_P$ bundles to $SO(3)_M \times SU(2)_N \times SO(3)_P$ bundles, and Bock is the Bockstein associated to the above short exact sequence (232).

# 5 Conclusions and Outlook

In this paper, we explored 2-group symmetries and global form of 0-form symmetry groups in 5d SCFTs and related 5d SQFTs obtained by mass deforming the 5d SCFTs. Our analysis used information about the Coulomb branch of vacua of these theories which can be read from M-theory geometric constructions of these theories. Central to this analysis is the identification of the structure group, which is the maximal group acting faithfully on the particle spectrum.

As a concrete application of our method, we explicitly studied 2-group symmetries and global form of 0-form symmetry groups in a large class of 5d SCFTs which admit a mass deformation such that the low-energy theory after the mass deformation becomes some 5d $\mathcal{N} = 1$ gauge theory with a simply connected gauge group $G$ based on a simple Lie algebra $\mathfrak{g}$. Such theories were geometrically classified in [43] using the description in M-theory on Calabi-Yau, retaining only the information about the surface geometry. Since we are interested in 2-group symmetries, we only study the sub-class of theories (in the above class of theories) that have in addition a 1-form symmetry, which again is encoded in the geometry of the surfaces [20, 21]. We collect our results in table 1 where some of the 2-groups are unconfirmed since we do not know an argument to show that the associated Postnikov class is non-trivial. It would be very interesting to determine whether these Postnikov classes are non-trivial.

Our method determines the global form of 0-form symmetry group of any 5d SCFT/SQFT irrespective of whether or not the theory admits a 1-form symmetry. This was exemplified by

studying the global form of the 0-form symmetry group of rank 1 Seiberg theories which have $\mathfrak{e}_{N_f+1}$ 0-form symmetry algebra. We found that for $N_f \neq 1$, the 0-form symmetry group is the maximally center-quotiented form of the group $E_{N_F+1}/Z_{E_{N_F+1}}$. We also explain how this is consistent with the ray indices for these theories [19] which contains representations carrying non-trivial charges under the flavor center $Z_{E_{N_F+1}}$.

It would also be interesting to find the connection to the work in [55], where global symmetries of the $E_{N_F+1}$ theories in 3d were computed, in particular the connection with the groups $(E_1)_1$ therein and the 5d global symmetries.

For the $E_1$ theory, we also discuss the anomaly discussed in [18], and propose that it most likely lifts to an anomaly of the 2group symmetry, but we leave the determination of the 2-group anomaly to future work. We also propose a mechanism for matching the anomaly of [18] on the Higgs branch of vacua of the $E_1$ SCFT. The proposal involves a coupling between the Higgs branch sigma model capturing the 0-form symmetry and a 5d $\mathbb{Z}_2$ gauge theory capturing the 1-form symmetry on the Higgs branch.

Another question that arose in the context of global flavor symmetries is the superconformal index in the case of non-simply connected gauge groups, such as $SO(3)$. We have argued using the results of [53] that the index for $SU(2)_0$ and $SO(3)_0$ should agree [31]. It would be interesting to provide more physical arguments explaining the matching of the two indices.

# Acknowledgements

We thank Oren Bergman, Cyril Closset, Pietro Benetti-Genolini, Federico Bonetti, Simone Giacomelli, Hee-Cheol Kim, Joonho Kim, Miguel Montero, Kantaro Ohmori, Luigi Tizzano and Irene Valenzuela for comments and discussions. We in particular thank Pietro Benetti-Genolini and Luigi Tizzano for discussions and comments on this draft. LB and SSN are particularly grateful to Yasunori Lee, Kantaro Ohmori and Yuji Tachikawa for sharing a draft of their unpublished work which formed a core inspiration for this paper, and some of the analysis in sections 2.1 and 2.3 is a generalization of the analysis presented in their work. We are grateful to Cyril Closset for pointing out an error in the analysis of section 3.5 in a previous version of this paper.
The work is supported by the European Union's Horizon 2020 Framework: ERC grants 682608 (FA, LB, JO, and SSN), 787185 (LB), and 864828 (JO). SSN is also supported in part by the "Simons Collaboration on Special Holonomy in Geometry, Analysis and Physics".

# A  Geometric Computations

## A.1  Background Material

In this paper, we discuss non-compact Calabi-Yau threefolds which carry intersecting compact and non-compact Kahler surfaces. We represent the intersection properties of these surfaces in terms of graphs whose nodes represent different surfaces and edges represent various in-

---

[31]One might think this is intuitively obvious since the index only counts local operators and so should be insensitive to the choice of line operators. However, there can be twisted sectors giving rise to additional local operators after gauging the 1-form symmetry. One such source is provided by local operators living at the ends of the topological defect dual to the Pontryagin square $\mathfrak{P}(B)$ of the 1-form symmetry background $B$ in the $SU(2)_0$ SCFT. Such operators do not contribute to the $SU(2)_0$ index. But they become genuine local operators (i.e. not attached to any higher-dimensional defect) in the $SO(3)_0$ SCFT upon gauging the 1-form symmetry, and hence contribute to $SO(3)_0$ index. Such operators might carry representations under the flavor $\mathfrak{su}(2)$ algebra that are not carried by the genuine local operators in the $SU(2)_0$ SCFT.

tersections between the surfaces. A node of the form

$$\mathbf{m}_n^b \tag{234}$$

denotes a surface obtained by taking a Hirzebruch surface of degree[32] $n$ and performing $b$ blowups on it[33]. The variable $\mathbf{m}$ is a label used to distinguish different surfaces, when one has multiple surfaces. It denotes that the surface (234) is the $m$-th surface in the geometry and the surface (234) is also denoted as $S_m$. On the other hand, a node of the form[34]

$$\mathbf{N}_m^b \qquad \text{or} \qquad \mathbf{M}_m^b \qquad \text{or} \qquad \mathbf{P}_m^b \tag{235}$$

denotes a $\mathbb{P}^1$ fibered non-compact surface carrying $b$ blowups and $m$ is a label differentiating different non-compact surfaces in the geometry. The $\mathbb{P}^1$ fiber of any surface $D$ of type (234) or (235) is denoted as $f$ and has the intersection properties

$$\begin{aligned} f \cdot_D f &= 0 \\ f \cdot_D x &= 0 \quad, \\ f \cdot_D K_D &= -2 \end{aligned} \tag{236}$$

where $K_D$ is dual of the canonical class of $D$, $x$ is any blowup out of the $b$ blowups and $\cdot_D$ denotes intersection number inside the surface $D$. For Hirzebruch surface $D$ of type (234) we denote the base $\mathbb{P}^1$ as $e$ which satisfies

$$\begin{aligned} e \cdot_D f &= 1 \\ e \cdot_D x &= 0 \\ e \cdot_D e &= -n \\ e \cdot_D K_D &= n-2 \end{aligned} \tag{237}$$

For a Hirzebruch surface we also define a curve $h := e + nf$. For a blowup $x$ on a surface $D$, we have

$$\begin{aligned} x \cdot_D x &= -1 \\ x \cdot_D K_D &= -1. \end{aligned} \tag{238}$$

For two different blowups $x_i, x_j$ on a surface $D$, we have

$$x_i \cdot_D x_j = 0 \tag{239}$$

For a non-compact surface $D$ of type (235), we denote by $ne + \sum_i n_i x_i$ a section of the $\mathbb{P}^1$ fibration which has the following intersection properties with other curves in $D$

$$\begin{aligned} \left(ne + \sum_{i=1}^b n_i x_i\right) \cdot_D f &= n \\ \left(ne + \sum_{i=1}^b n_i x_i\right) \cdot_D x_j &= -n_j. \end{aligned} \tag{240}$$

The blowups $b$ are often divided as $b = b_1 + b_2 + b_3 + b_4$ and the corresponding surfaces represented as

$$\mathbf{m}_n^{b_1+b_2+b_3+b_4} \quad \text{or} \quad \mathbf{N}_m^{b_1+b_2+b_3+b_4} \quad \text{or} \quad \mathbf{M}_m^{b_1+b_2+b_3+b_4} \quad \text{or} \quad \mathbf{P}_m^{b_1+b_2+b_3+b_4} \tag{241}$$

---

[32]We allow $n$ to become negative for ease in providing a uniform description in some cases.

[33]It should be noted that the blowups are in general non-generic.

[34]Unlike $\mathbf{m}$ which is a number, $\mathbf{N}$, $\mathbf{M}$ and $\mathbf{P}$ are just letters.

Then, the $b_1$ blowups are labeled as $x_i$ for $i = 1, \cdots, b_1$, the $b_2$ blowups are labeled as $y_i$ for $i = 1, \cdots, b_2$, the $b_3$ blowups are labeled as $z_i$ for $i = 1, \cdots, b_3$, and the $b_4$ blowups are labeled as $w_i$ for $i = 1, \cdots, b_4$. $\sum x_i$ represents the sum of all the $b_1$ blowups etc.

An edge between two nodes $D_1$ and $D_2$ of the form

$$D_1 \ \overset{C_{12} \qquad\quad C_{21}}{\rule{6cm}{0.4pt}} \ D_2 \tag{242}$$

describes an intersection between $D_1$ and $D_2$ such that the intersection locus when seen from the point of view of divisor $D_1$ describes a curve $C_{12}$ inside $D_1$, and when seen from the point of view of divisor $D_2$ describes a curve $C_{21}$ inside $D_2$. We also say that the edge describes a "gluing" of $D_1$ and $D_2$ such that $C_{12} \in D_1$ is glued to $C_{21} \in D_2$. Extending this terminology, $C_{12}$ and $C_{21}$ are also referred to as "gluing curves". An edge between two nodes $D_1$ and $D_2$ of the form

$$D_1 \ \overset{C_{12,1}, C_{12,2}, \cdots, C_{12,m}}{\rule{4cm}{0.4pt}} \ m \ \overset{C_{21,1}, C_{21,2}, \cdots, C_{21,m}}{\rule{4cm}{0.4pt}} \ D_2 \tag{243}$$

describes $m$ number of intersections between $D_1$ and $D_2$. The $i$-th intersection is described by gluing $C_{12,i} \in D_1$ to $C_{21,i} \in D_2$. We also define "total gluing curves" $C_{12}$ and $C_{21}$ in this situation as

$$\begin{aligned} C_{12} &:= \sum_{i=1}^{m} C_{12,i} \\ C_{21} &:= \sum_{i=1}^{m} C_{21,i} \end{aligned} \tag{244}$$

Finally, an edge of the form

$$
\begin{array}{c}
C_{1,1}, C_{1,2}, \cdots, C_{1,m} \\
m \quad \bigcirc \quad D_1 \\
C'_{1,1}, C'_{1,2}, \cdots, C'_{1,m}
\end{array}
\tag{245}
$$

describes $m$ number of self-intersections of $D$. The $i$-th self-intersection is described by gluing $C_{1,i} \in D_1$ to $C'_{1,i} \in D_1$. We also define "total self-gluing curve" $C_1$ as

$$C_1 := \sum_{i=1}^{m} \left( C_{1,i} + C'_{1,i} \right). \tag{246}$$

An intersection number between a surface $D_2$ and a compact curve $C$ living in a different surface $D_1$ is computed as

$$D_2 \cdot C = C_{12} \cdot_{D_1} C, \tag{247}$$

where $C_{12}$ is the total gluing curve defined in (244). On the other hand, an intersection number between a surface $D_1$ and a compact curve $C$ living in the same surface $D_1$ is computed as

$$D_1 \cdot C = (K_1 + C_1) \cdot_{D_1} C, \tag{248}$$

where $K_1$ is the dual of canonical class of $D_1$ and $C_1$ is the total self-gluing curve defined in (246).

In the computations of section 4, each compact surface $D$ corresponds to a $U(1)_D$ gauge group. The charge $q_D(C)$ of matter content associated to M2-brane wrapping a compact curve $C$ under the $U(1)_D$ gauge group is computed as

$$q_D(C) = -D \cdot C. \tag{249}$$

In addition to these charges, we also need flavor center charges of $C$ which are discussed in next subsection.

## A.2 Center Symmetry from Geometry

Some particular configurations of intersecting $\mathbb{P}^1$ fibered surfaces can be associated to non-abelian gauge or flavor algebras, depending on whether the surfaces are compact or non-compact [9, 48]. If the surfaces are compact, and the associated algebra is a gauge algebra $\mathfrak{g}$, then the encoding of the center symmetry of the simply connected group $G$ associated to $\mathfrak{g}$ was discussed in [21]. A straightforward extension of their analysis can be used to describe the encoding of the center symmetry of the simply connected group $F$ associated to a flavor algebra $\mathfrak{f}$ if the surfaces are non-compact, which is needed for the analysis of this paper.

For $\mathfrak{f} = \mathfrak{su}(n)$, the geometric configuration is

$$\mathbf{N}_1 \;\overset{e}{\rule{1.5cm}{0.4pt}}\overset{e}{\rule{1cm}{0.4pt}}\; \mathbf{N}_2 \;\overset{e}{\rule{1.2cm}{0.4pt}}\; \cdots \;\overset{e}{\rule{1cm}{0.4pt}}\; \mathbf{N}_{n-1} \tag{250}$$

and the charge $q_f(C)$ under the center $\mathbb{Z}_n$ of $F = SU(n)$ of a compact curve $C$ can be computed as

$$q_f(C) = -\left(\sum_{i=1}^{n-1} i \mathbf{N}_i\right) \cdot C \pmod{n}. \tag{251}$$

For $\mathfrak{f} = \mathfrak{so}(2n+1)$, the geometric configuration is

$$\mathbf{N}_1 \;\overset{e}{\rule{1.2cm}{0.4pt}}\overset{e}{\rule{1cm}{0.4pt}}\; \mathbf{N}_2 \;\overset{e}{\rule{1cm}{0.4pt}}\; \cdots \;\overset{e}{\rule{1cm}{0.4pt}}\; \mathbf{N}_{n-1} \;\overset{2e}{\rule{1.2cm}{0.4pt}}\overset{e}{\rule{0.8cm}{0.4pt}}\; \mathbf{N}_n \tag{252}$$

and the charge $q_f(C)$ under the center $\mathbb{Z}_2$ of $F = \text{Spin}(2n+1)$ of a compact curve $C$ can be computed as

$$q_f(C) = -\mathbf{N}_n \cdot C \pmod{2}. \tag{253}$$

For $\mathfrak{f} = \mathfrak{sp}(n)$, the geometric configuration is

$$\mathbf{N}_1 \;\overset{e}{\rule{1.2cm}{0.4pt}}\overset{e}{\rule{1cm}{0.4pt}}\; \mathbf{N}_2 \;\overset{e}{\rule{1cm}{0.4pt}}\; \cdots \;\overset{e}{\rule{1cm}{0.4pt}}\; \mathbf{N}_{n-1} \;\overset{e}{\rule{1.2cm}{0.4pt}}\overset{2e}{\rule{0.8cm}{0.4pt}}\; \mathbf{N}_n \tag{254}$$

and the charge $q_f(C)$ under the center $\mathbb{Z}_2$ of $F = Sp(n)$ of a compact curve $C$ can be computed as

$$q_f(C) = -\left(\sum_{i=1}^{n} \frac{1-(-1)^i}{2} \mathbf{N}_i\right) \cdot C \pmod{2}. \tag{255}$$

For $\mathfrak{f} = \mathfrak{so}(4n+2)$, the geometric configuration is

$$
\begin{array}{c}
\mathbf{N}_{2n} \\[4pt]
\Big| e \\
\Big| e \\[4pt]
\mathbf{N}_1 \;\overset{e}{\rule{1.2cm}{0.4pt}}\overset{e}{\rule{1cm}{0.4pt}}\; \mathbf{N}_2 \;\overset{e}{\rule{1cm}{0.4pt}}\; \cdots \;\overset{e}{\rule{1cm}{0.4pt}}\; \mathbf{N}_{2n-1} \;\overset{e}{\rule{1.2cm}{0.4pt}}\overset{e}{\rule{1cm}{0.4pt}}\; \mathbf{N}_{2n+1}
\end{array}
\tag{256}
$$

and the charge $q_f(C)$ under the center $\mathbb{Z}_4$ of $F = \text{Spin}(4n+2)$ of a compact curve $C$ can be computed as

$$q_f(C) = -\left(\sum_{i=1}^{2n-1} \left(1-(-1)^i\right) \mathbf{N}_i + 3\mathbf{N}_{n-1} + \mathbf{N}_n\right) \cdot C \pmod{4}. \tag{257}$$

For $\mathfrak{f} = \mathfrak{so}(4n)$, the geometric configuration is

$$
\begin{array}{c}
\mathbf{N}_{2n} \\
\Big| e \\
\Big| e \\
\mathbf{N}_1 \xrule{e}{e} \mathbf{N}_2 \xrule{e}{} \cdots \xrule{}{e} \mathbf{N}_{2n-2} \xrule{e}{e} \mathbf{N}_{2n-1}
\end{array}
\tag{258}
$$

The charge $q_f^C(C)$ under the subgroup $\mathbb{Z}_2^C$ of the center $\mathbb{Z}_2^S \times \mathbb{Z}_2^C$ of $F = \mathrm{Spin}(4n)$ of a compact curve $C$ can be computed as

$$
q_f^C(C) = -\left( \sum_{i=1}^{2n-1} \frac{1-(-1)^i}{2} \mathbf{N}_i \right) \cdot C \pmod 2
\tag{259}
$$

and the charge $q_f^S(C)$ under the subgroup $\mathbb{Z}_2^S$ of the center $\mathbb{Z}_2^S \times \mathbb{Z}_2^C$ of $F = \mathrm{Spin}(4n)$ of a compact curve $C$ can be computed as

$$
q_f^S(C) = -\left( \sum_{i=1}^{2n-2} \frac{1-(-1)^i}{2} \mathbf{N}_i + \mathbf{N}_{2n} \right) \cdot C \pmod 2 .
\tag{260}
$$

For $\mathfrak{f} = \mathfrak{e}_6$, the geometric configuration is

$$
\begin{array}{c}
\mathbf{N}_6 \\
\Big| e \\
\Big| e \\
\mathbf{N}_1 \xrule{e}{e} \mathbf{N}_2 \xrule{e}{e} \mathbf{N}_3 \xrule{e}{e} \mathbf{N}_4 \xrule{e}{e} \mathbf{N}_5
\end{array}
\tag{261}
$$

and the charge $q_f(C)$ under the center $\mathbb{Z}_3$ of $F = E_6$ of a compact curve $C$ can be computed as

$$
q_f(C) = -\left( \sum_{i=1}^{5} i\mathbf{N}_i \right) \cdot C \pmod 3 .
\tag{262}
$$

For $\mathfrak{f} = \mathfrak{e}_7$, the geometric configuration is

$$
\begin{array}{c}
\mathbf{N}_7 \\
\Big| e \\
\Big| e \\
\mathbf{N}_1 \xrule{e}{e} \mathbf{N}_2 \xrule{e}{e} \mathbf{N}_3 \xrule{e}{e} \mathbf{N}_4 \xrule{e}{e} \mathbf{N}_5 \xrule{e}{e} \mathbf{N}_6
\end{array}
\tag{263}
$$

and the charge $q_f(C)$ under the center $\mathbb{Z}_2$ of $F = E_7$ of a compact curve $C$ can be computed as

$$
q_f(C) = -\left( \mathbf{N}_1 + \mathbf{N}_3 + \mathbf{N}_7 \right) \cdot C \pmod 2 .
\tag{264}
$$

## A.3  Flavor Center Charges of Instantons

If one has a geometry which can describe non-abelian gauge theory with simple gauge algebra in the IR, then every compact surface is $\mathbb{P}^1$ fibered and all the (total) gluing curves involve the $e$ curve. The charges of matter content associated to all the compact curves in the geometry can be accounted by accounting the charges of matter content associated to all $\mathbb{P}^1$ fibers $f$, all

blowups $x_i$, and a single $e$ curve living in one of the compact surfaces [21]. The fibers $f$ account for contributions of gauge bosons, the blowups $x_i$ account for contributions of hypermultiplet content of the gauge theory and the single $e$ curve accounts for the contribution of a massive instanton BPS particle. The contributions of gauge bosons and hypermultiplets can be easily computed from the gauge theory data and one does not need geometry to determine it (even though one can, but it is a harder computation than gauge theory). However, the contribution of instanton requires non-perturbative understanding of the 5d gauge theory which is much easier to deduce from the geometry rather than the gauge theory. The charge of instanton under the center of the gauge group was determined in [21] for all 5d gauge theories with simple gauge algebra that can arise in the IR of 5d SCFTs. Here we extend their analysis to determine the charge of instanton under the center of each simply connected group associated to each flavor symmetry algebra that can arise for such 5d gauge theories. It should be noted that we only discuss flavor symmetry algebras visible at the level of gauge theory and not their possible enhancements. The results of this sub-appendix are also collected in section 2.2.3.

First of all, if we have $n$ hypers in a complex representation $\boldsymbol{R}$ of the the gauge group $G$, then the instanton associated to $G$ has a trivial charge under the center of the simply connected group $SU(n)$ associated to the $\mathfrak{su}(n)$ flavor symmetry algebra rotating the $n$ hypers. This is because the $n$ hypers then form a single *full* hypermultiplet transforming in representation $\boldsymbol{R} \otimes \boldsymbol{F}$ of $G \times SU(n)$, and according to the arguments presented in [21], the instanton of $G$ cannot be charged under the center $\mathbb{Z}_n$ of $SU(n)$. Non-trivial charge for instanton arises only if the hypers are charged in a real representation of $G$. We now go over the geometric realization of all such possible cases one-by-one and determine how the instanton is charged. We pick the geometries from [10].

Consider the gauge theory $\text{Spin}(2n+1)$ with $m$ hypers in vector representation $\boldsymbol{F}$. The associated geometry can be represented as

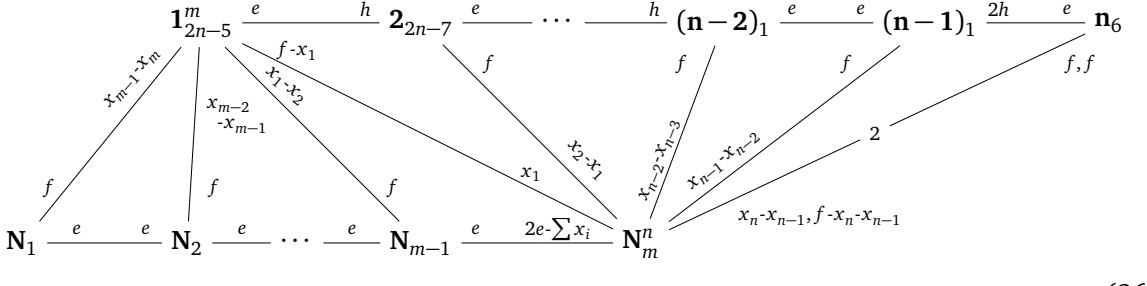

$$(265)$$

We can choose the instanton to be the curve $e$ in $\boldsymbol{S}_1$, which has to satisfy the consistency condition that its charge under the gauge center is as claimed in section 2.2.3. In this case, using (253) (or rather its compact analogue), we see that the gauge center charge of the chosen instanton is indeed 0 (mod 2), as required. Using (255), we see that its flavor center charge is $m$ (mod 2) under the $\mathbb{Z}_2$ center of the $Sp(m)$ simply connected group associated to the $\mathfrak{sp}(m)$ flavor symmetry algebra rotating the $m$ hypers. But the gauge center charge of a vector hyper is 0 (mod 2) and its flavor center charge is 1 (mod 2). So, by redefining the instanton contribution by combining it with the vector contribution, we can always choose the instanton to have both gauge and flavor center charges to be 0 (mod 2). Said another way, we could choose the instanton to be the curve $e - \sum x_i$ in $\boldsymbol{S}_1$, which also has gauge center charge 0 (mod 2), as required, and its flavor center is computed using (255) to also be 0 (mod 2).

Now, consider the gauge theory $\text{Spin}(2n+2)$ with $m$ hypers in vector representation $\boldsymbol{F}$.

The associated geometry can be represented as

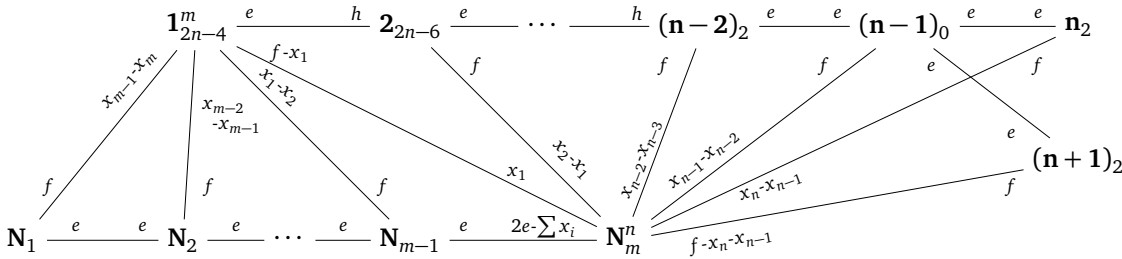

(266)

We can choose the instanton to be the curve $e$ in $S_1$, which satisfies the consistency condition that its charge under the gauge center is as claimed in section 2.2.3, i.e. 0 (mod 2). Using (255), we see that its flavor center charge is $m$ (mod 2), which is the final answer appearing in section 2.2.3. Notice that in this case we cannot use the vector hyper to neutralize the instanton's flavor center charge since the vector hyper carries a non-trivial gauge center charge but the instanton is required to carry trivial gauge center charge.

Now, consider the gauge theory $SU(4)_k$ with $m$ hypers in 2-index antisymmetric irrep $\Lambda^2$. The associated geometry can be represented as

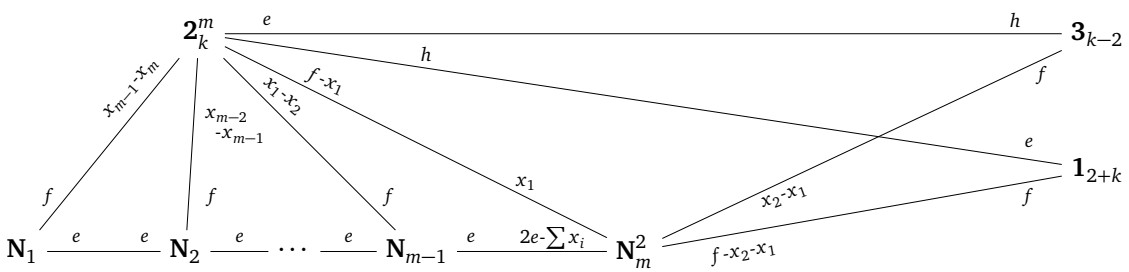

(267)

We can choose the instanton to be the curve $e$ in $S_1$, which satisfies the consistency condition that its charge under the gauge center is as claimed in section 2.2.3. Using (255), we see that its flavor center charge is $m$ (mod 2), which is the final answer appearing in section 2.2.3.

Now, consider the gauge theory $Sp(n)$ with $m$ hypers in fundamental representation $F$. The associated geometry can be represented as

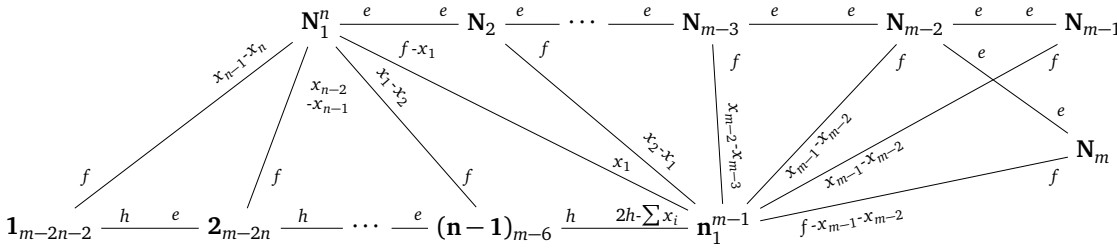

(268)

We can choose the instanton to be the curve $e - \frac{1-(-1)^n}{2} x_1$ in $S_1$, which satisfies the consistency condition that its charge under the gauge center is trivial, which is as required by section 2.2.3. We see that its flavor center charge is the same as that of a spinor/cospinor irrep of $\mathfrak{so}(2m)$ symmetry rotating the $m$ hypers, which is the final answer appearing in section 2.2.3, where we have chosen an outer-automorphism frame of $\mathfrak{so}(2m)$ such that the flavor center charge of instanton always coincides with the flavor center charge of a spinor irrep of $\mathfrak{so}(2m)$.

Now, consider the gauge theory $Sp(n)$ with $m$ hypers in 2-index antisymmetric irrep $\Lambda^2$. Since the hypers have trivial charge under gauge center and a non-trivial charge 1 (mod 2) under the flavor center, we can always choose an instanton which has a trivial charge under

the flavor center (and the charge under gauge center is determined by the discrete theta angle of $Sp(n)$ as discussed in section 2.2.3). The same argument can also be applied to $F_4$ and $G_2$ gauge theories carrying $m$ hypers in **27** and **7** dimensional representations respectively.

Now, consider the gauge theory $Sp(3)$ with a half-hyper in 3-index antisymmetric irrep $\Lambda^3$ and $2m+1$ half-hypers in fundamental representation $F$. The associated geometry can be represented as

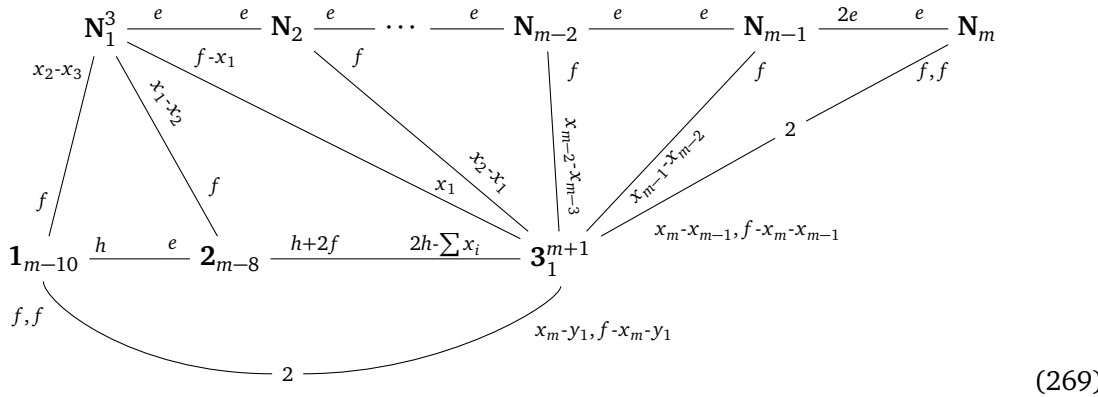

$$\tag{269}$$

We can choose the instanton to be the curve $e - x_1$ in $S_1$, which satisfies the consistency condition that its charge under the gauge center is trivial, which is as required by section 2.2.3. We see that its flavor center charge is 1 (mod 2) under the $\mathbb{Z}_2$ center of the $\mathrm{Spin}(2m+1)$ simply connected group associated to the $\mathfrak{so}(2m+1)$ flavor symmetry algebra rotating the $2m+1$ half-hypers, which is the final answer appearing in section 2.2.3.

Now, consider the gauge theory $\mathrm{Spin}(7)$ with $m$ hypers in spinor irrep $S$. The associated geometry can be represented as

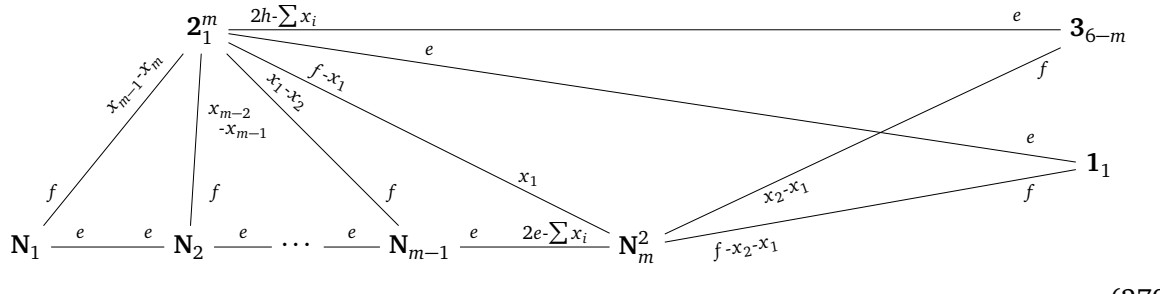

$$\tag{270}$$

We can choose the instanton to be the curve $e$ in $S_1$, which satisfies the consistency condition that its charge under the gauge center is trivial, which is as required by section 2.2.3. We see that its flavor center charge is $m$ (mod 2) under the $\mathbb{Z}_2$ center of the $Sp(m)$ simply connected group associated to the $\mathfrak{sp}(m)$ flavor symmetry algebra rotating the $m$ hypers, which is the final answer appearing in section 2.2.3.

Now, consider the gauge theory $\mathrm{Spin}(9)$ with $m$ hypers in spinor irrep $S$. The associated

geometry can be represented as

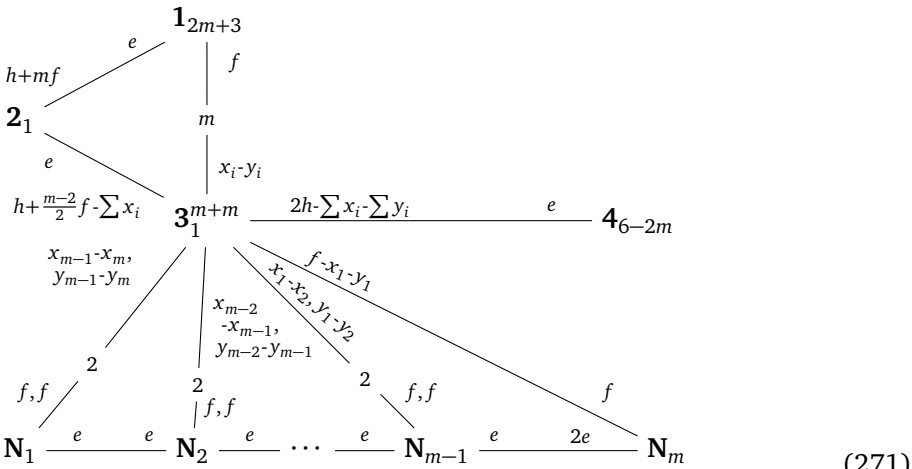

$$(271)$$

for $m$ even, and as

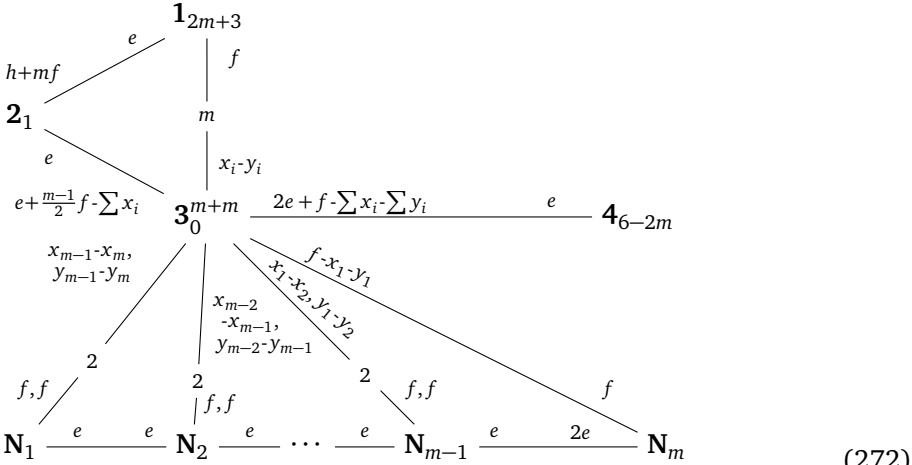

$$(272)$$

for $m$ odd. The $m$ intersections between $S_1$ and $S_3$ are such that $x_i - y_i \in S_3$ is glued to a copy of $f \in S_1$ for all $i = 1, \cdots, m$. We can choose the instanton to be the curve $e$ in $S_1$, which satisfies the consistency condition that its charge under the gauge center is trivial, which is as required by section 2.2.3. We see that its flavor center charge is 0 (mod 2) under the $\mathbb{Z}_2$ center of the $Sp(m)$ simply connected group associated to the $\mathfrak{sp}(m)$ flavor symmetry algebra rotating the $m$ hypers, which is the final answer appearing in section 2.2.3.

Now, consider the gauge theory Spin(11) with 2 hypers in spinor irrep $S$. The associated

geometry can be represented as

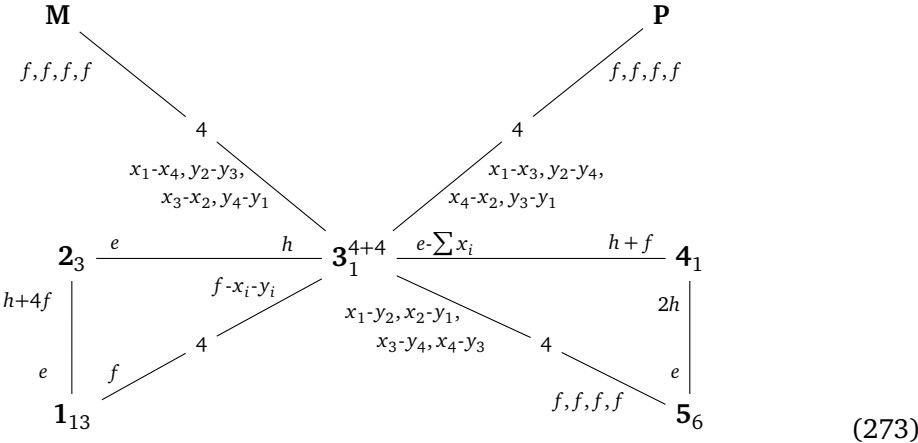

$$(273)$$

We can choose the instanton to be the curve $e$ in $S_1$, which satisfies the consistency condition that its charge under the gauge center is trivial, which is as required by section 2.2.3. We see that its flavor center charge is also trivial, which is the final answer appearing in section 2.2.3.

Now, consider the gauge theory Spin(11) with 3 half-hypers in spinor irrep $S$. The associated geometry can be represented as

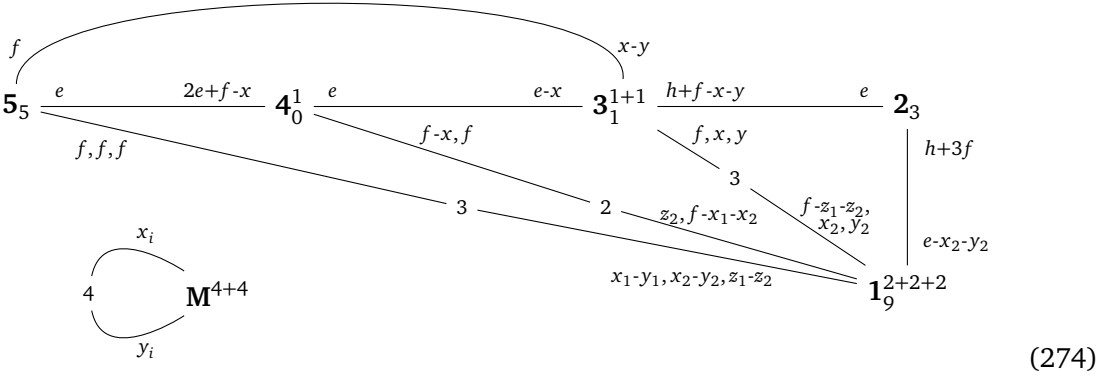

$$(274)$$

along with the following extra gluing rules:

- $x_1, x_1, y_1, y_1, z_2, f - z_2, f - z_2 - x_2, z_2 - x_2, z_1, f - z_1, f - z_1 - y_2, z_1 - y_2$ in $S_1$ are glued to $x_2, y_2, x_1, y_1, f - x_3, f - y_3, f, f, f - x_4, f - y_4, f, f$ in $M$.

- $f - y, f - y, f - x, f - x$ in $S_3$ are glued to $x_3, y_3, x_4, y_4$ in $M$.

- $f - x, x, x, f - x$ in $S_4$ are glued to $f - x_3, f - y_3, f - x_2, f - y_2$ in $M$.

- $f, f, f, f$ in $S_5$ are glued to $x_2 - x_1, y_2 - y_1, x_3 - x_4, y_3 - y_4$ in $M$.

We can choose the instanton to be the curve $e$ in $S_1$, which satisfies the consistency condition that its charge under the gauge center is trivial, which is as required by section 2.2.3. We see that its flavor center charge is also trivial, which is the final answer appearing in section 2.2.3.

Now, consider the gauge theory Spin(12) with 2 hypers in a spinor irrep $S$. The associated

geometry can be represented as

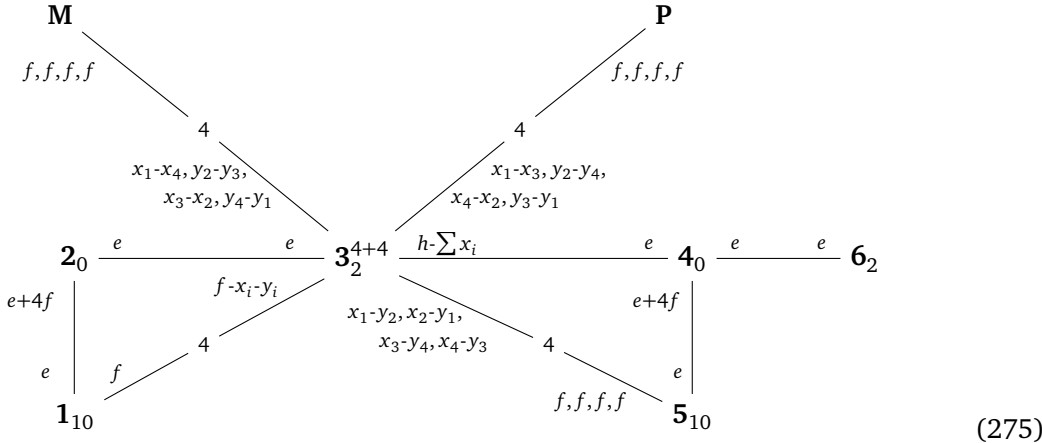

$$(275)$$

We can choose the instanton to be the curve $e$ in $\boldsymbol{S}_1$, which satisfies the consistency condition that its charge under the gauge center is trivial, which is as required by section 2.2.3. We see that its flavor center charge is also trivial, which is the final answer appearing in section 2.2.3.

Now, consider the gauge theory Spin(12) with 3 half-hypers in a spinor irrep $\boldsymbol{S}$. The associated geometry can be represented as

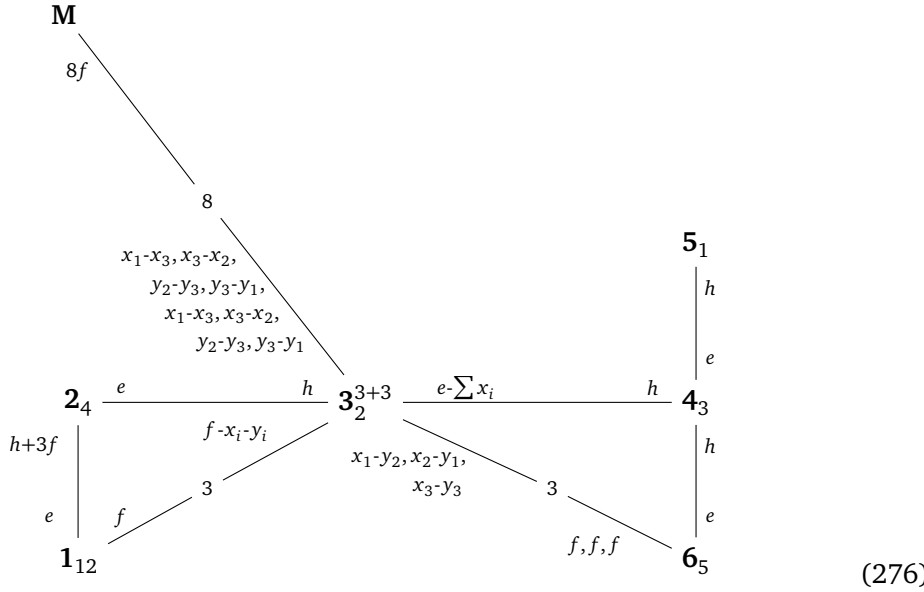

$$(276)$$

We can choose the instanton to be the curve $e$ in $\boldsymbol{S}_5$, which satisfies the consistency condition that its charge under the gauge center $\mathbb{Z}_2^S$ is trivial but its charge under the gauge center $\mathbb{Z}_2^C$ is non-trivial, which is as required by section 2.2.3. We see that its flavor center charge is trivial, which is the final answer appearing in section 2.2.3.

Finally, consider the gauge theory $SU(6)_{\frac{1}{2}+l}$ with 3 half-hypers in 3-index antisymmetric



irrep $\mathbf{\Lambda}^3$. The associated geometry can be represented as

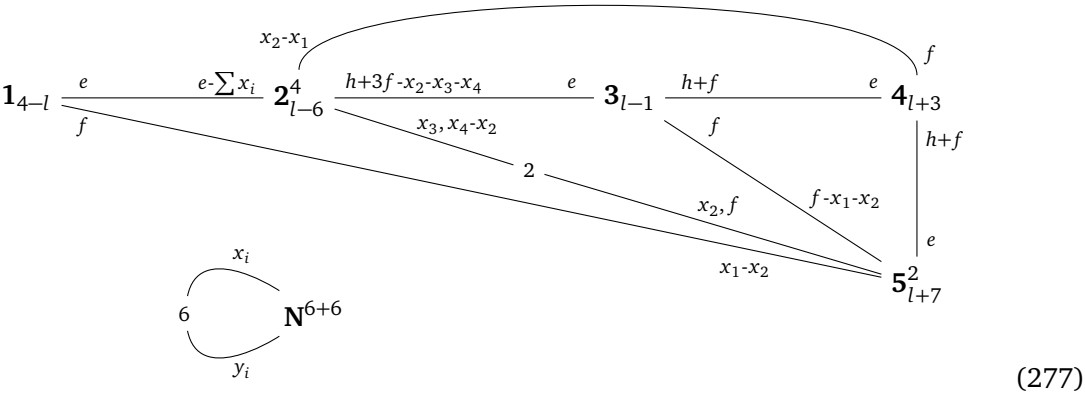

$$(277)$$

along with the following extra gluing rules:

- $f, f, f, f, f, f$ in $\boldsymbol{S}_1$ are glued to $x_4 - x_5, y_4 - y_5, x_6 - x_7, x_8 - x_9, y_6 - y_7, y_8 - y_9$ in $\mathbf{N}$.

- $x_3, f - x_3, x_2, x_2, x_1, x_1, x_4 - x_3, f - x_4 - x_3$ in $\boldsymbol{S}_2$ are glued to $f - x_4, f - y_4, x_7, y_7, x_9, y_9, f, f$ in $\mathbf{N}$.

- $f, f, f, f$ in $\boldsymbol{S}_3$ are glued to $x_5 - x_7, x_4 - x_6, y_5 - y_7, y_4 - y_6$ in $\mathbf{N}$.

- $f, f, f, f$ in $\boldsymbol{S}_4$ are glued to $x_7 - x_9, x_6 - x_8, y_7 - y_9, y_6 - y_8$ in $\mathbf{N}$.

- $f - x_1, x_1, f - x_2, x_2, x_1, f - x_1, x_2, f - x_2$ in $\boldsymbol{S}_5$ is glued to $f - x_6, f - x_5, f - x_7, f - x_4, f - y_6, f - y_5, f - y_7, f - y_4$ in $\mathbf{N}$.

We can choose the instanton to be the curve $e$ in $\boldsymbol{S}_1$, which satisfies the consistency condition that its charge under the gauge center is as required by section 2.2.3. We see that its flavor center charge is trivial, which is the final answer appearing in section 2.2.3.

## B  Fractionalization of Instanton Number and Mixed 't Hooft Anomalies of 5d Gauge Theories

In this appendix we review the details of coupling a gauge theory to background fields for the 1-form symmetry. The consequence of this operation is that the instanton density factionalizes, potentially leading to anomalies. In 5d gauge theories with a 1-form symmetry this is indeed the case. When the instanton number becomes fractional, a generic 5d gauge theory potentially has two types of 't Hooft anomalies. The first one is a mixed anomaly between the instanton $U(1)_I$, whose current is $J_I = \frac{1}{8\pi^2} * \mathrm{Tr}(F \wedge F)$, and the 1-form symmetry corresponding to the center of the simply connected gauge group $Z(G)$. This is due to the 5d coupling between the instanton density and the background gauge connection for $U(1)_I$, [18]. The second anomaly is a 1-form symmetry 't Hooft anomaly, which is only present when the 5d bare Chern-Simons coupling is non-trivial [24]. In the second part of this appendix we will review how and when these anomalies arise.

Let us first discuss the case when $\mathcal{O} = Z(G)$. Activating a background 2-form $B$ for the 1-form symmetry has a non-trivial effect on the instanton density,

$$I_4 = \frac{1}{8\pi^2} \mathrm{Tr}(F \wedge F). \tag{278}$$

Table 2: Coefficients of the fractional instanton density for non-trivial background $B$. For Spin($4N$) we have two contributions given by $\mathfrak{P}(B^{(L)} + B^{(R)})$ and $B^{(L)} \cup B^{(R)}$, respectively.

| $G$ | $Z(G)$ | $\alpha_G$ |
|---|---|---|
| $SU(N)$ | $\mathbb{Z}_N$ | $\frac{N-1}{2N}$ |
| $Sp(N)$ | $\mathbb{Z}_2$ | $\frac{N}{4}$ |
| $\mathrm{Spin}(2N+1)$ | $\mathbb{Z}_2$ | $\frac{1}{2}$ |
| $\mathrm{Spin}(4N+2)$ | $\mathbb{Z}_4$ | $\frac{2N+1}{8}$ |
| $\mathrm{Spin}(4N)$ | $\mathbb{Z}_2 \times \mathbb{Z}_2$ | $\left(\frac{N}{4}, \frac{1}{2}\right)$ |
| $E_6$ | $\mathbb{Z}_3$ | $\frac{2}{3}$ |
| $E_7$ | $\mathbb{Z}_2$ | $\frac{3}{4}$ |

If $B$ is trivial, then we sum over $G$ bundles which have the property that the instanton number

$$\oint_{M_4 \subset M_5} I_4 \in \mathbb{Z}, \tag{279}$$

where $M_4$ is any 4-cycle inside $M_5$. If instead we activate a non-trivial $B$, then we sum over $G/Z(G)$ bundles instead of $G$ bundles so that

$$B = w_2(G/Z(G)) \in H^2(M_5, Z(G)), \tag{280}$$

where $w_2 \in H^2(G/Z(G), Z(G))$ is the characteristic class capturing the obstruction of lifting a $G/Z(G)$ bundle to a $G$ bundle. In this case, the instanton number can take fractional values with the fractional part captured by the following expression for $G \neq \mathrm{Spin}(4N)$

$$I_4 = \alpha_G \mathfrak{P}(B) \mod \mathbb{Z}, \tag{281}$$

where $\mathfrak{P}(B)$ is the Pontryagin square[35] of $B$ and $\alpha_G$ is a coefficient tabulated in table 2 [56]. For $G = \mathrm{Spin}(4N)$, the center 1-form symmetry $Z(G) = \mathbb{Z}_2^L \times \mathbb{Z}_2^R$ has two $\mathbb{Z}_2$ factors. Let us choose a basis for $Z(G)$ such that under $\mathbb{Z}_2^L$, spinor has charge 1 and cospinor has charge 0, and under $\mathbb{Z}_2^R$, spinor has charge 0 and cospinor has charge 1. Let us denote the background fields for the two $\mathbb{Z}_2$ factors as $B_L$ and $B_R$. Then, we have

$$I_4 = \frac{N}{4} \mathfrak{P}(B_L + B_R) + \frac{1}{2} B_L \cup B_R \mod \mathbb{Z}. \tag{282}$$

Now let us consider the case when the 1-form symmetry is broken to a subgroup $\mathcal{O} = \mathbb{Z}_k$ of $Z(G)$. For $G \neq \mathrm{Spin}(4N)$, we can write $Z(G) = \mathbb{Z}_n$, and then $k$ must divide $n$. A background $B'$ for $G/\mathcal{O}$ can be related to a background $B$ for $G/Z(G)$ as

$$B = \frac{n}{k} B' \tag{283}$$

modifying the fractionalization (281) of the instanton number to

$$I_4 = \frac{n^2 \alpha_G}{k^2} \mathfrak{P}(B') \mod \mathbb{Z}. \tag{284}$$

---

[35]When $B$ is valued in $\mathbb{Z}_n$ for odd $n$, we define $\mathfrak{P}(B) = B \cup B$ and its periods take integer values modulo $n$. For even $n$, $\mathfrak{P}(B)$ has periods which (on a spin manifold) take even integer values modulo $2n$.

For $G = \mathrm{Spin}(4N)$, the possible subgroups are $\mathbb{Z}_2^L$, $\mathbb{Z}_2^R$, and the diagonal $\mathbb{Z}_2 \hookrightarrow \mathbb{Z}_2^L \times \mathbb{Z}_2^R$. Then

$$
\begin{aligned}
\mathcal{O} = \mathbb{Z}_2^L: \quad & I_4 = \frac{N}{4}\mathfrak{P}(B_L) \quad \mathrm{mod} \quad \mathbb{Z} \\
\mathcal{O} = \mathbb{Z}_2^R: \quad & I_4 = \frac{N}{4}\mathfrak{P}(B_R) \quad \mathrm{mod} \quad \mathbb{Z} \\
\mathcal{O} = \mathbb{Z}_2: \quad & I_4 = \frac{1}{2}B' \cup B' = \frac{1}{2}\mathfrak{P}(B') \quad \mathrm{mod} \quad \mathbb{Z},
\end{aligned}
\tag{285}
$$

where in the final case we consider the diagonal $\mathbb{Z}_2$, where $B_L = B_R = B'$.

The Pontryagin square can also be presented in a continuous version, [1, 57–60]. For $SU(N)$ this is done by extending it to $U(N)$, whose connection is $A'$. The 1-form symmetry background $\mathbb{Z}_N$ is specified by a pair $(B, C)$, where $C$ is the $U(1)$ gauge field of $U(N)$, and the pair satisfies $NB = dC$. The fields transform under the 1-form symmetry as follows

$$
\begin{aligned}
A' &\to A' + \lambda \mathbb{I}_N, \\
C &\to C + N\lambda, \\
B &\to B + d\lambda,
\end{aligned}
\tag{286}
$$

where $\mathbb{I}_N$ is $N \times N$ identity matrix, $\lambda$ is a $U(1)$ connection. The invariant field strength reads

$$
F \to F' - B \mathbb{I}_N.
\tag{287}
$$

The instanton density[36], which is invariant under (286), is

$$
\tfrac{1}{2}\mathrm{Tr}(F \wedge F) \to \tfrac{1}{2}\mathrm{Tr}(F' \wedge F') - \tfrac{1}{N}\mathrm{Tr}(F') \wedge dC + \tfrac{1}{2N}dC \wedge dC.
\tag{288}
$$

Taking into account the constraint $\mathrm{Tr}(F') = dC$, and expressing everything in terms of the second Chern class,

$$
\tfrac{1}{2}\mathrm{Tr}(F' \wedge F') = \tfrac{1}{2}c_1(F')^2 - c_2(F'),
\tag{289}
$$

we have that

$$
\tfrac{1}{2}\mathrm{Tr}(F \wedge F) \to c_2(F') - \tfrac{N-1}{2N}dC \wedge dC,
\tag{290}
$$

where $dC$ have integer periods. The second term on the right hand side is the continuum version of the Pontryagin square term previously introduced. Matter fields can break the 1-form symmetry to a subgroup thereof. In the $SU(N)$ case this can be broken to $\mathbb{Z}_k$, where $\frac{N}{k} = q \in \mathbb{Z}$. Expression (283) with $n = N$ implies that

$$
NB' = dC',
\tag{291}
$$

where $dC'$ has also integer period. Finally $dC'$ is related to $dC$ by using $NB = dC$ and (283) with $n = N$,

$$
\frac{N}{k}dC' = dC.
\tag{292}
$$

We can now see that with (290) is modified for $\mathcal{O} = \mathbb{Z}_k \subset \mathbb{Z}_N$ as follows,

$$
\tfrac{1}{2}\mathrm{Tr}(F \wedge F) \to c_2(F') - \frac{N-1}{2N}\left(\frac{N}{k}\right)^2 dC' \wedge dC',
\tag{293}
$$

which is consistent with the general expression (284). For the other simply connected groups, $G$, a similar procedure can be implemented by embedding a maximal subgroup $\prod_i SU(n_i) \subset G$, and by activating certain 1-form symmetry backgrounds for the subgroup factors, as described in [56].

---

[36]In order to avoid carrying around factors of $2\pi$, we implement the following redefinition $\frac{F}{2\pi} \to F$. This will allow to us to avoid the factors of $2\pi$ in the periodicities as well.

### B.1 Anomalies of 5d Gauge Theories

There are two type of 't Hooft anomaly of 5d gauge theories. The first one is a mixed 't Hooft anomaly between the instanton symmetry and the 1-form symmetry. The second is a 1-form symmetry cubic 't Hooft anomaly, which is present only in the case $G = SU(N)$, with bare Chern-Simons coupling $\kappa \neq 0$.

It was proposed in [18] that the fractionalization of the instanton number in the presence of a non-trivial 1-form symmetry background $B$ leads to an anomaly under large gauge transformations of $U(1)_I$. Correspondingly, the large gauge transformations take the form

$$A_I \to A_I + a_I \tag{294}$$

with

$$\oint_{\gamma \subset M_5} a_I \in \mathbb{Z} \tag{295}$$

for $\gamma$ an arbitrary 1-cycle in $M_5$. Then the coupling

$$\mathcal{S}_{\text{IR}}[A_I] \supset A_I \wedge I_4 \,, \tag{296}$$

leads to the following phase ambiguity of the partition function,

$$Z[A_I, B] \to Z[A_I, B] \exp\left( 2\pi i \frac{n^2 \alpha_G}{k^2} \int_{M_5} a_I \cup \mathfrak{P}(B) \right) \tag{297}$$

for $G \neq \text{Spin}(4N)$. The above phase ambiguity is valued in $\mathbb{Z}_q$ if

$$\frac{n^2 \alpha_G}{k^2} = \frac{p}{q} \,. \tag{298}$$

The anomaly theory $A_6[A_I, B]$ associated to the above phase ambiguity is then

$$\mathcal{A}_6^{\text{mix}}[A_I, B]_{\text{IR}} = 2\pi i \frac{n^2 \alpha_G}{k^2} \int_{Y_6} F_I \cup \mathfrak{P}(B) \,, \tag{299}$$

where $Y_6$ is a 6-manifold such that $\partial Y_6 = M_5$. The connection $A_I$ lifts to a connection on $Y_6$ and $F_I$ is its field strength.

For $G = \text{Spin}(4N)$ and $\mathcal{O} = Z(G) = \mathbb{Z}_2^L \times \mathbb{Z}_2^R$, the phase ambiguity is

$$Z[A_I, B] \to Z[A_I, B] \exp\left( 2\pi i \int_{M_5} a_I \cup \left( \frac{N}{4} \mathfrak{P}(B_L + B_R) + \frac{1}{2} B_L \cup B_R \right) \right), \tag{300}$$

which is $\mathbb{Z}_2$ valued. For the case of $G = \text{Spin}(4N)$ and $\mathcal{O} = \mathbb{Z}_2^L \times \mathbb{Z}_2^R$, the anomaly theory can be written as

$$\mathcal{A}_6^{\text{mix}}[A_I, B]_{\text{IR}} = 2\pi i \int_{Y_6} F_I \cup \left( \frac{N}{4} \mathfrak{P}(B_L + B_R) + \frac{1}{2} B_L \cup B_R \right). \tag{301}$$

The second 't Hooft anomaly was proposed in [24], and it involves only the one for symmetry. This anomaly appears by activating the 1-form symmetry, $\mathcal{O} = \mathbb{Z}_k$, background $B$ and substituting (287) into the bare Chern-Simons coupling, by also using the fact that

$$\kappa CS_5(A) = 2\pi \frac{\kappa}{6} \int_{Y_6} \text{Tr}(F^3) \,, \tag{302}$$

where $M_6$ is a six-dimensional space which bounds $M_5$ where the 5d theory lives.

For even $\kappa$ the anomaly theory is

$$\mathcal{A}_6[B]_{\text{IR}} = 2\pi i \frac{\kappa N(N-1)(N-2)}{6} \int_{Y_6} B^3 \,, \tag{303}$$

where $\oint B \in \frac{\mathbb{Z}}{k}$. For odd $\kappa$ the anomaly theory is

$$\mathcal{A}_6[B, A_c]_{\text{IR}} = 2\pi i \frac{\kappa N(N-1)(N-2)}{6} \int_{Y_6} B^3 + 2\pi i \frac{N(N-1)}{2} \int_{M_6} B^2 dA_c \,, \tag{304}$$

where $A_c$ is a spin$^c$ connection suche that $dA = \frac{1}{2} w_2(TM_5)$. This is due to the fact that for $\kappa$ odd the $\kappa CS_5$ is not an integer on general $M_5$. To better understand this anomaly we should actually analyse the $B^3$ in the integral. In fact, in the continuum limit we have that $kB = dC$, for an auxiliary pure gauge $U(1)$ connection, $C$, as introduced in (286). The anomaly then reads,

$$\mathcal{A}_6[B]_{\text{IR}} = 2\pi i \frac{\kappa N(N-1)(N-2)}{6k^2} \int_{M_6} dC^3 \,, \tag{305}$$

where $dC$ is the frst Chern class of a complex line bundle associated with the auxiliary $U(1)$. The integral $\int_{M_6} dC^3 \in 6\mathbb{Z}$ on spin manifold and $\int_{M_6} dC^3 \in 3\mathbb{Z}$ on non-spin manifold [46,49].

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
