# Peer review of "The Global Form of Flavor Symmetries and 2-Group Symmetries in 5d SCFTs"

_SciPost Physics, doi:SciPost Phys. 13, 024 (2022)_

## Round 3 · Referee Report · Anonymous (Referee 1) · 2022-3-29

Report

The manuscript studies the 2-group symmetries in various 5d SCFTs, and in particular the global form of the 0-form symmetry. The symmetries are determined from the low energy gauge theory on the Coulomb branch or after mass deformations. The symmetry is also verified using the known superconformal and ray indices.

The results are interesting and I recommend the manuscript for publication after following minor points are taken care of.
-p35. For option 1, is it clear that the anomaly (3.48) is well-defined for 2-group bundles where $B$ is not closed?
-p36. In (3.54), the cup1 product is not associative so the equation needs a bracket.

---

## Round 3 · Referee Report · Anonymous (Referee 2) · 2022-6-27

Report

This paper studies 2-group global symmetries in 5D SCFT’s. The authors demonstrate that there is a general phenomenon that leads to 2-group global symmetries in 5D theories where there is a redundancy between the global and gauge symmetry of the theory. In this paper, the authors work in detail through an explicit example, demonstrating the existence of 2-group symmetries through multiple methods. The authors use this analysis to argue that the existence of 2-group global symmetries on the Coulomb branch of a SCFT must be matched by the UV theory. I highly recommend this paper for publication.

---

## Editorial Decision

published